# Bayesian Learning of Optimal Policies in Markov Decision Processes with Countably Infinite State-Space

**Saghar Adler**
University of Michigan

Vijay Subramanian
University of Michigan

## Abstract

Models of many real-life applications, such as queueing models of communication networks or computing systems, have a countably infinite state-space. Algorithmic and learning procedures that have been developed to produce optimal policies mainly focus on finite state settings, and do not directly apply to these models. To overcome this lacuna, in this work we study the problem of optimal control of a family of discrete-time countable state-space Markov Decision Processes (MDPs) governed by an unknown parameter $\theta \in \Theta$, and defined on a countably-infinite state-space $\mathcal{X} = \mathbb{Z}_+^d$, with finite action space $\mathcal{A}$, and an unbounded cost function. We take a Bayesian perspective with the random unknown parameter $\boldsymbol{\theta}^*$ generated via a given fixed prior distribution on $\Theta$. To optimally control the unknown MDP, we propose an algorithm based on Thompson sampling with dynamically-sized episodes: at the beginning of each episode, the posterior distribution formed via Bayes' rule is used to produce a parameter estimate, which then decides the policy applied during the episode. To ensure the stability of the Markov chain obtained by following the policy chosen for each parameter, we impose ergodicity assumptions. From this condition and using the solution of the average cost Bellman equation, we establish an $\tilde{O}(dh^d\sqrt{|\mathcal{A}|T})$ upper bound on the Bayesian regret of our algorithm, where $T$ is the time-horizon. Finally, to elucidate the applicability of our algorithm, we consider two different queueing models with unknown dynamics, and show that our algorithm can be applied to develop approximately optimal control algorithms.

## 1 Introduction

Many real-life applications, such as communication networks, supply chains, and computing systems, are modeled using queueing models with countably infinite state-space. In the existing analysis of these systems, the models are assumed to be known, but despite this, developing optimal control schemes is hard, with only a few examples worked out [35, 9, 54]. However, knowing the model, algorithmic procedures exist to produce approximately optimal policies [35] (such as value iteration and linear programming). Given the success of data-driven optimal control design, in particular Reinforcement Learning (RL), we explore the use of such methods for the countable state-space controlled Markov processes. However, current RL methods that focus on finite-state settings do not apply to the mentioned queueing models. With the model unknown, our goal is to develop a meta-learning scheme that is RL-based but obtains good performance by utilizing algorithms developed when models are known. Specifically, we study the problem of optimal control of a family of discrete-time countable state-space MDPs governed by an unknown parameter $\theta$ from a general space $\Theta$ with each MDP evolving on the countable state-space $\mathcal{X} = \mathbb{Z}_+^d$ and finite action space $\mathcal{A}$. The cost function is unbounded and polynomially dependent on the state, following the examples of minimizing waiting times in queueing systems. Taking a Bayesian view, we assume the model is governed by an unknown parameter $\theta^* \in \Theta$ generated from a fixed and known prior distribution. We aim to learn a policy $\pi$ that minimizes the optimal infinite-horizon average cost over a given class of policies $\Pi$ with low Bayesian regret with respect to the (parameter-dependent) optimal policy in $\Pi$.

37th Conference on Neural Information Processing Systems (NeurIPS 2023).

To avoid many technical difficulties in countably infinite state-space settings, it is crucial to establish certain assumptions regarding the class of models from which the unknown system is drawn; some examples are: i) the number of deterministic stationary policies is not finite; and ii) in average cost optimal control problems, without stability/ergodicity assumptions, an optimal policy may not exist [40], and when it exists, it may not be stationary or deterministic [20]. With these in mind, we assume that for any state-action pair, the transition kernels in the model class are categorical and skip-free to the right, i.e., with finite support with a bound depending on the state only in an additive manner; both are common features of queueing models where an increase in state is due to arrivals. A second set of assumptions ensure stability by assuming that the Markov chains obtained by using different policies in $\Pi$ are geometrically ergodic with uniformity across $\Theta$. From these assumptions, moments on hitting times are derived in terms of Lyapunov functions for polynomial ergodicity. These assumptions also yield a solution to the average cost optimality equation (ACOE) [9].

**Contributions:** To optimally control the unknown MDP, we propose an algorithm based on Thompson sampling with dynamically-sized episodes; posterior sampling is used based on its broad applicability and computational efficiency [46, 47]. At the beginning of each episode, a posterior distribution is formed using Bayes' rule, and an estimate is realized from this distribution which then decides the policy used throughout the episode. To evaluate the performance of our proposed algorithm, we use the metric of Bayesian regret, which compares the expected total cost achieved by a learning policy $\pi_L$ until time horizon $T$ with the policy achieving the optimal infinite-horizon average cost in the policy class $\Pi$. We consider regret guarantees in three different settings as follows:
1. In Theorem 1, for $\Pi$ being the set of all policies and assuming that we have oracle access to the optimal policy for each parameter, we establish an $\tilde{O}(dh^d\sqrt{|\mathcal{A}|T})$ upper bound on the Bayesian regret of this algorithm compared to the optimal policy.
2. In Corollary 1, where class $\Pi$ is a subset of all stationary policies and where we know the best policy within this subset for each parameter via an oracle, we prove an $\tilde{O}(dh^d\sqrt{|\mathcal{A}|T})$ upper bound on the Bayesian regret of our proposed algorithm, relative to the best-in-class policy.
3. In Theorem 2, we explore a scenario where we have access to an approximately optimal policy, rather than the optimal policy in set $\Pi$ (which are all assumed to be stationary policies). When the approximately optimal policies satisfy Assumptions 3-4, we prove an $\tilde{O}(dh^d\sqrt{|\mathcal{A}|T})$ regret bound, relative to the optimal policy in set $\Pi$.

Finally, to provide examples of our framework for developing approximately optimal control algorithms for stochastic systems with unknown dynamics, we study two different queueing models that meet our technical conditions. The first example is a continuous-time queueing system with two heterogeneous servers with unknown service rates and a common infinite buffer with the decision being the use of the slower server. Here, the optimal policy that minimizes the average waiting time is a threshold policy [38] which yields a queue-length after which the slower server is always used. The second model is a two-server queueing system, each with separate infinite buffers, to one of which a dispatcher routes an incoming arrival. Here, the optimal policy minimizing the waiting time is a switching-curve [26] with the specifics unknown for general parameter values, so we find the best policy within a commonly used set of switching-curve policies (Max-Weight policies [58, 59]), and assign the arrival to the queue with minimum weighted queue-length. For both models, we verify our assumptions for the class of optimal/best-in-class policies corresponding to different service rates and conclude that our proposed algorithm can be used to learn the optimal/best-in-class policy.

**Related Work:** Thompson sampling [62], or posterior sampling, has been applied to RL in many contexts of unknown MDPs [55, 45] and partially observed MDPs [28]; see tutorials [22, 50] for a comprehensive survey. It has been used in the parametric learning context [6] to minimize either Bayesian [46, 47, 49, 1, 60, 61] or frequentist [5, 23] regret. The bulk of the literature, including [5, 23, 49], analyzes finite-state and finite-action models but with different parameterizations such that a general dependence of the models on the parameters is allowed. The work in [61] studies general state-space MDPs but with a scalar parameterization with a Lipschitz dependence of the underlying models. Our problem formulation specifically considers countable state-space models with the models related via ergodicity, which we believe is a natural choice. Our focus on parametric learning is also connected to older work in adaptive control [3, 24] which studies asymptotically optimal learning for general parameter settings but with either a finite or countably infinite number of policies. Learning-based asymptotically optimal control in queues has a long history [36, 35] but recently there is increased work that also characterizes finite-time regret performance with respect to a well-known good policy or the optimal policy; see [63] for a survey. A series of work has studied

learning with Max-Weight policies to get stability and linear regret [44, 30] or just stability [65]. A recent related work [18] considers learning optimal paramterized policies in queueing networks when the MDP is known. In a finite or countable state-space setting of specific queueing models where the parameters can be estimated, many works [2, 17, 53, 32, 31, 14, 21, 16] have used forced exploration type schemes to obtain either regret that is constant or scaling logarithmically in the time-horizon.

Another line of work studies the problem of learning the optimal policy in an undiscounted finite-horizon MDP with a bounded reward function. Reference [66] uses a Thompson sampling-based learning algorithm with linear value function approximation to study an MDP with a bounded reward function in a finite-horizon setting. Reference [15] considers an episodic finite-horizon MDP with known bounded rewards but unknown transition kernels modeled using linearly parameterized exponential families. A maximum likelihood (ML) based algorithm coupled with exploration done by constructing high probability confidence sets around the ML estimate is used to learn the unknown parameters. In another work, [48] extends the problem setting of [15] to an episodic finite-horizon MDP with unknown rewards and transitions modeled using parametric bilinear exponential families. To learn the unknown parameters, they use a ML based algorithm with exploration done with explicit perturbation. We note that all mentioned works consider a finite-horizon problem. In contrast, our work considers an average cost problem, an infinite-horizon setting, and provides finite-time performance guarantees. In addition, these works focus on an MDP with a bounded reward function. Our focus, however, is learning in MDPs with unbounded rewards with the goal of covering practical queueing examples. We note that the parameterization of transitions used in [48, 15] can be used within our framework. However, similar to our work, additional stability assumptions are necessary to guarantee asymptotic learning and sub-linear regret. Another issue with exponential transition families is that they do not allow for $0$ entries, which limits their applicability in queueing models.

In another work, [51] studies discounted MDPs with unknown dynamics, and unbounded state-space, but with bounded rewards, and learns an online policy that satisfies a specific notion of stability. It is also assumed that a Lyapunov function ensuring stability for the optimal policy exists. We note that [51] ignores optimality and focuses on finding a stable policy, which contrasts with our work that evaluates performance relative to the optimal policy. Secondly, [51] considers a discounted reward problem, essentially a finite-time horizon problem. Average cost problems, such as ours, are infinite-time horizon problems, so connections to discounted problems can only be made in the limit of the discount parameter going to $1$. Moreover, [51] considers a bounded reward function, simplifying their analysis but not practical for many queueing examples. Further, the assumption of a stable optimal policy with a Lyapunov function (as in [51]) is highly restrictive for bounded reward settings with discounting. Additionally, average cost problems with bounded costs need strong state-independent recurrence conditions for the existence of (stationary) optimal solutions, which many queueing examples don't satisfy; see [12]. Further complications can also arise with bounded costs: e.g., [20] shows that a stationary average cost optimal policy may not exist.

## 2 Problem formulation

We consider a family of discrete-time Markov Decision Processes (MDPs) governed by parameter $\theta \in \Theta$ with the MDP for parameter $\theta$ described by $(\mathcal{X}, \mathcal{A}, c, P_\theta)$. For exposition purposes, we assume that all the MDPs are on (a common) countably infinite state-space $\mathcal{X} = \mathbb{Z}_+^d$. We denote the finite action space by $\mathcal{A}$, the transition kernel by $P_\theta : \mathcal{X} \times \mathcal{A} \to \Delta(\mathcal{X})$, and the cost function by $c : \mathcal{X} \times \mathcal{A} \to \mathbb{R}_+$. As mentioned earlier, we will take a Bayesian view of the problem and assume that the model is generated using an unknown parameter $\boldsymbol{\theta}^* \in \Theta$, which is generated from a given fixed prior distribution $\nu(\cdot)$ on $\Theta$. Our goal is to find a policy $\pi : \mathcal{X} \to \mathcal{A}$ that tries to achieve Bayesian optimal performance in policy class $\Pi$, i.e., minimizes the expected regret with $\boldsymbol{\theta}^*$ chosen from the prior distribution $\nu(\cdot)$. For each value $\theta \in \Theta$, the minimum infinite-horizon average cost is defined as

$$J(\theta) = \inf_{\pi \in \Pi} \limsup_{T \to \infty} \frac{1}{T} \mathbb{E} \big[ \sum_{t=1}^{T} c(\boldsymbol{X}(t), A(t)) \big], \tag{1}$$

where we optimize over a given class of policies $\Pi$ and $\boldsymbol{X}(t) = (X_1(t), \dots, X_d(t)) \in \mathcal{X}$ and $A(t) \in \mathcal{A}$ are the state and action at $t \in \mathbb{N}$. Typically, we set this class to be all (causal) policies, but it is also possible to consider $\Pi$ to be a proper subset of all policies as we will explore in our results. For a learning policy $\pi_L$ that aims to select the optimal control without model knowledge but with

knowledge of $\Theta$ and the prior $\nu$, the Bayesian regret until time horizon $T \geq 2$ is defined as

$$R(T, \pi_L) = \mathbb{E}\Big[\sum_{t=1}^{T}\big[c(\boldsymbol{X}(t), A(t)) - J(\boldsymbol{\theta}^*)\big]\Big], \qquad (2)$$

where the expectation is taken over $\boldsymbol{\theta}^* \sim \nu$ and the dynamics induced by $\pi_L$. Owing to underlying challenges in countable state-space MDPs, we require the below assumptions on the cost function.

**Assumption 1.** *The cost function $c : \mathcal{X} \times \mathcal{A} \to \mathbb{R}_+$ is assumed to satisfy the following two conditions:*

1. *For every number $z \geq 0$ and action $a \in \mathcal{A}$, $c(\boldsymbol{x}, a) \geq z$ outside a finite subset of $\mathcal{X}$.*

2. *The cost function is upper-bounded by a multivariate polynomial $f_c : \mathbb{Z}_+^d \to \mathbb{R}_+$ which is increasing in every component on $\boldsymbol{x} \in \mathbb{Z}_+^d$ and has maximum degree of $r$ $(\geq 1)$ in any dimension. We can assume that $f_c(\boldsymbol{x}) = K \sum_{i=1}^{d}(x_i)^r$ for some $K > 0$, where $\boldsymbol{x} = (x_1, \ldots, x_d)$.*

Thus, the cost function increases without bound (in the state) at a polynomial rate. This assumption is common in practice—holding costs in queueing models are polynomial in the state components. To avoid technical issues the infinite state-space setting also necessitates some assumptions on the class from which the unknown model is drawn. For instance, irreducibility of Markov chains on such state-spaces does not ensure positive recurrence (and ergodicity). Moreover, for average cost optimal control problems, without stability even the existence of an optimal policy is not guaranteed, and we need more conditions. The following assumption ensures a skip-free behaviour for transitions, which holds in many queueing models, where an increase in state corresponds to (new) arrivals.

**Assumption 2.** *From any state-action pair $(\boldsymbol{x}, a)$, the transition is to a finite number of states. We also assume that all transition kernels are skip-free to the right: for some $h \geq 1$ which is independent of $\theta \in \Theta$ and $(\boldsymbol{x}, a) \in \mathcal{X} \times \mathcal{A}$, we have $P_\theta(\boldsymbol{x}'; \boldsymbol{x}, a) = 0$ for all $\boldsymbol{x}' \in \{\tilde{\boldsymbol{x}} \in \mathbb{Z}_+^d : \|\tilde{\boldsymbol{x}}\|_1 > \|\boldsymbol{x}\|_1 + h\}$.*

Learning necessitates some commonalities within the class of models so that using a policy well-suited to one model provides information on other models too. For us, these are in the form of constraints on the transition kernels of the models and stability assumptions. As simple union bound arguments don't work in the countably infinite state-space setting, we will use the stability assumptions instead. In our setting, we consider a class of models, each with a policy being well-suited to at least one model in the class, and use the set of policies to search within. Using a reduced set of policies is necessary as the number of deterministic stationary policies is infinite. To learn correctly while restricting attention to this subset policy class, requires some regularity assumptions when a policy well-suited to one model is tried on a different model. Our ergodicity assumptions are one convenient choice; see Appendix A.1 for details. These assumptions let us characterize the distributions of the first passage times of the Markov processes via stability conditions; see Lemmas 10 and 11.

**Assumption 3.** *For any MDP $(\mathcal{X}, \mathcal{A}, c, P_\theta)$ with parameter $\theta \in \Theta$, there exists a unique optimal policy $\pi_\theta^*$ that minimizes the infinite-horizon average cost within the class of policies $\Pi$. Furthermore, for any $\theta_1, \theta_2 \in \Theta$, the Markov process with transition kernel $P_{\theta_1}^{\pi_{\theta_2}^*}$ obtained from the MDP $(\mathcal{X}, \mathcal{A}, c, P_{\theta_1})$ by following policy $\pi_{\theta_2}^*$ is irreducible, aperiodic, and geometrically ergodic with geometric ergodicity coefficient $\gamma_{\theta_1, \theta_2}^g \in (0, 1)$ and stationary distribution $\mu_{\theta_1, \theta_2}$. This is equivalent to the existence of finite set $C_{\theta_1, \theta_2}^g$ and Lyapunov function $V_{\theta_1, \theta_2}^g : \mathcal{X} \to [1, +\infty)$ satisfying*

$$\Delta V_{\theta_1, \theta_2}^g(\boldsymbol{x}) \leq -\big(1 - \gamma_{\theta_1, \theta_2}^g\big) V_{\theta_1, \theta_2}^g(\boldsymbol{x}), \ \boldsymbol{x} \in \mathcal{X} \setminus C_{\theta_1, \theta_2}^g \ and \ P_{\theta_1}^{\pi_{\theta_2}^*} V_{\theta_1, \theta_2}^g(\boldsymbol{x}) < +\infty, \ \boldsymbol{x} \in C_{\theta_1, \theta_2}^g,$$

*where $\Delta V_{\theta_1, \theta_2}^g(\boldsymbol{x}) := P_{\theta_1}^{\pi_{\theta_2}^*} V_{\theta_1, \theta_2}^g(\boldsymbol{x}) - V_{\theta_1, \theta_2}^g(\boldsymbol{x})$. Setting $b_{\theta_1, \theta_2}^g := \max_{\boldsymbol{x} \in C_{\theta_1, \theta_2}^g} P_{\theta_1}^{\pi_{\theta_2}^*} V_{\theta_1, \theta_2}^g(\boldsymbol{x}) + V_{\theta_1, \theta_2}^g(\boldsymbol{x})$ yields*

$$\Delta V_{\theta_1, \theta_2}^g(\boldsymbol{x}) \leq -\big(1 - \gamma_{\theta_1, \theta_2}^g\big) V_{\theta_1, \theta_2}^g(\boldsymbol{x}) + b_{\theta_1, \theta_2}^g \mathbb{I}_{C_{\theta_1, \theta_2}^g}(\boldsymbol{x}), \quad \boldsymbol{x} \in \mathcal{X}. \qquad (3)$$

*Then, we have the following assumptions relating all the models in $\Theta$:*

1. *The geometric ergodicity coefficient is uniformly bounded below 1: $\gamma_*^g := \sup_{\theta_1, \theta_2 \in \Theta} \gamma_{\theta_1, \theta_2}^g < 1$.*

2. *We assume that $\{0^d\} \subseteq \cap_{\theta_1, \theta_2 \in \Theta} C_{\theta_1, \theta_2}^g$ and $C_*^g = \cup_{\theta_1, \theta_2 \in \Theta} C_{\theta_1, \theta_2}^g$ is a finite set. We further assume that $b_*^g := \sup_{\theta_1, \theta_2} b_{\theta_1, \theta_2}^g < +\infty$.*

**Remark 1.** *The uniqueness of the optimal policy is not essential for the validity of our results, provided that all optimal policies satisfy our assumptions. When this condition is not met, we need to select an optimal policy that is geometrically ergodic for all $\theta \in \Theta$. This issue can be avoided by using a smaller subset of policies for which ergodicity can be shown, such as Max-Weight policies.*

Geometric ergodicity implies that all moments of the hitting time of state $0^d$, say $\tau_{0^d}$, from any initial state $\boldsymbol{x} \neq 0^d$ are finite as $\mathbb{E}_{\boldsymbol{x}}[\kappa^{\tau_{0^d}}] \leq c_1 V^g(\boldsymbol{x})$ (for specific $\kappa > 1$ and $c_1$), and so, $\mathbb{E}_{\boldsymbol{x}}[\tau_{0^d}^k] \leq c_1 V^g(\boldsymbol{x})k!/\log^k(\kappa) < +\infty$ for all $k \in \mathbb{N}$; see Appendix A.2. Function $V^g$ is typically exponential in some norm of the state and yields an exponential bound for moments of hitting times, and a poor regret bound. To improve the regret bound, we need a different drift equation with function $V^p$ with polynomial dependence on a norm of the state that bounds certain polynomial moments of $\tau_{0^d}$.

**Assumption 4.** *Given $\theta_1, \theta_2 \in \Theta$, Markov process obtained from MDP $(\mathcal{X}, \mathcal{A}, c, P_{\theta_1})$ by following policy $\pi_{\theta_2}^*$ is polynomially ergodic through the Foster-Lyapunov criteria: there exists a finite set $C_{\theta_1,\theta_2}^p$, constants $\beta_{\theta_1,\theta_2}^p, b_{\theta_1,\theta_2}^p > 0$, $\alpha_{\theta_1,\theta_2}^p \in [\frac{r}{r+1}, 1)$, and function $V_{\theta_1,\theta_2}^p : \mathcal{X} \to [1, +\infty)$ satisfying*

$$\Delta V_{\theta_1,\theta_2}^p(\boldsymbol{x}) \leq -\beta_{\theta_1,\theta_2}^p \left(V_{\theta_1,\theta_2}^p(\boldsymbol{x})\right)^{\alpha_{\theta_1,\theta_2}^p} + b_{\theta_1,\theta_2}^p \mathbb{I}_{C_{\theta_1,\theta_2}^p}(\boldsymbol{x}), \quad \boldsymbol{x} \in \mathcal{X}. \tag{4}$$

*Then, we have the following assumptions relating all the models in $\Theta$:*

1. *$V_{\theta_1,\theta_2}^p$ is a polynomial with positive coefficients, maximum degree (in any dimension) $r_{\theta_1,\theta_2}^p$, and sum of coefficients $s_{\theta_1,\theta_2}^p$. We assume $r_*^p = \sup_{\theta_1,\theta_2} r_{\theta_1,\theta_2}^p < \infty$ and $s_*^p = \sup_{\theta_1,\theta_2} s_{\theta_1,\theta_2}^p < \infty$.*

2. *We assume that $\{0^d\} \subseteq \cap_{\theta_1,\theta_2 \in \Theta} C_{\theta_1,\theta_2}^p$ and $C_*^p = \cup_{\theta_1,\theta_2 \in \Theta} C_{\theta_1,\theta_2}^p$ is a finite set. We further assume that $\beta_*^p := \inf_{\theta_1,\theta_2} \beta_{\theta_1,\theta_2}^p > 0$ and $b_*^p := \sup_{\theta_1,\theta_2} b_{\theta_1,\theta_2}^p < \infty$.*

3. *Let $K_{\theta_1,\theta_2}(\boldsymbol{x}) := \sum_{n=0}^{\infty} 2^{-n-2}\left(P_{\theta_1}^{\pi_{\theta_2}^*}\right)^n(\boldsymbol{x}, 0^d)$, which is positive for any pair $\theta_1, \theta_2 \in \Theta$ by irreducibility. We assume that it is strictly positive in $\Theta$: $K_* := \inf_{\theta_1,\theta_2} \min_{\boldsymbol{x} \in C_*^p} K_{\theta_1,\theta_2}(\boldsymbol{x}) > 0$.*

Assumptions 3-4 hold in many models of interest; see Appendix E. As average cost optimality is our design criterion, we need to ensure the existence of solutions to ACOE when $\Pi$ is the set of all policies, or Poisson equation when $\Pi$ is a subset of all policies. We discuss these two cases separately. *Case 1: $\Pi$ is the set of all policies.* For any parameter $\theta \in \Theta$, the MDP $(\mathcal{X}, \mathcal{A}, c, P_\theta)$ is said to satisfy the ACOE if there exists a constant $J(\theta)$ and a unique function $v(\cdot; \theta) : \mathcal{X} \to \mathbb{R}$ such that

$$J(\theta) + v(\boldsymbol{x}; \theta) = \min_{a \in \mathcal{A}} \left\{c(\boldsymbol{x}, a) + \sum_{\boldsymbol{y} \in \mathcal{X}} P_\theta(\boldsymbol{y}|\boldsymbol{x}, a)v(\boldsymbol{y}; \theta)\right\} \text{ with } v(0^d; \theta) = 0.$$

From [13] if the following conditions hold, ACOE has a solution, $J_\theta$ is the optimal infinite-horizon average cost, and there is an optimal stationary policy with ACOE becoming (5): (i) for every $(\boldsymbol{x}, a)$ and $z \geq 0$, cost function $c(\boldsymbol{x}, a) \geq z$ outside a finite subset of $\mathcal{X}$; (ii) there is a stationary policy with an irreducible and aperiodic Markov process with finite average cost; and (iii) from every $(\boldsymbol{x}, a)$ transition to a finite number of states is possible. From Assumptions 1-3, the above conditions hold. *Case 2: $\Pi$ is a proper subset of all policies.* Here, we posit that for every $\theta \in \Theta$ and its best in-class policy $\pi_\theta^*$, there exists a constant $J(\theta)$, the average cost of $\pi_\theta^*$, and a function $v(\cdot; \theta) : \mathcal{X} \to \mathbb{R}$ with

$$J(\theta) + v(\boldsymbol{x}; \theta) = c(\boldsymbol{x}, \pi_\theta^*(\boldsymbol{x})) + \sum_{\boldsymbol{y} \in \mathcal{X}} P_\theta(\boldsymbol{y}|\boldsymbol{x}, \pi_\theta^*(\boldsymbol{x}))v(\boldsymbol{y}; \theta). \tag{5}$$

This holds by the solution of the Poisson equation with the appropriate forcing function. For a Markov process $\boldsymbol{X}$ on the space $\mathcal{X}$ with transition kernel $P$ and cost function $\bar{c}(\cdot)$, a solution to the Poisson equation [41] is a scalar $J$ and function $v(\cdot) : \mathcal{X} \mapsto \mathbb{R}$ such that $J + v = \bar{c} + Pv$, where $v(\boldsymbol{z}) = 0$ for some $\boldsymbol{z} \in \mathcal{X}$. In our setting using [41, Sections 9.6-9.8], for a model governed by $\theta \in \Theta$ following policy $\pi_\theta^*$, we show a solution to the Poisson equation exists and is given by $v^{\pi_\theta^*}(0^d) = 0$, and

$$J(\theta) = \bar{C}^{\pi_\theta^*}(0^d)/\mathbb{E}_{0^d}^{\pi_\theta^*}[\tau_{0^d}] \text{ and } v^{\pi_\theta^*}(\boldsymbol{x}) = \bar{C}^{\pi_\theta^*}(\boldsymbol{x}) - J(\theta)\mathbb{E}_{\boldsymbol{x}}^{\pi_\theta^*}[\tau_{0^d}], \quad \forall \boldsymbol{x} \in \mathcal{X}, \tag{6}$$

where $\bar{C}^{\pi_\theta^*}(\boldsymbol{x}) = \mathbb{E}_{\boldsymbol{x}}^{\pi_\theta^*}\left[\sum_{i=0}^{\tau_{0^d}-1} c(\boldsymbol{X}(i), \pi_\theta^*(\boldsymbol{X}(i)))\right]$, and expectation is over trajectories of Markov chain $\boldsymbol{X}$ with transition kernel $P_\theta^{\pi_\theta^*}$ starting in state $\boldsymbol{x}$. In Appendix A.3, we present related definitions and show that from Assumptions 3-4, the requirements for the existence and finiteness of the solutions to Poisson equation are satisfied. Finally, we assume $\sup_{\theta \in \Theta} J(\theta)$ is finite, which typically holds as a result of the boundedness assumptions stated in Assumptions 3 or 4, along with Assumption 1.

**Algorithm 1** Thompson Sampling with Dynamically-sized Episodes (TSDE)
─────────────────────────────────────────────────────────────
1: Input: $\nu_0$
2: Initialization: $\boldsymbol{X}(1) = 0^d$, $t \leftarrow 1$
3: **for** episodes $k = 1, 2, ...$ **do**
4:     $t_k \leftarrow t$
5:     Generate $\theta_k \sim \nu_{t_k}$
6:     **while** $t \leq t_k + \tilde{T}_{k-1}$ and $N_t(\boldsymbol{x}, a) \leq 2N_{t_k}(\boldsymbol{x}, a)$ for all $(\boldsymbol{x}, a) \in \mathcal{X} \times \mathcal{A}$ **do**
7:         Apply action $A(t) = \pi_{\theta_k}^*(\boldsymbol{X}(t))$
8:         $N_t(\boldsymbol{X}(t), A(t)) \leftarrow N_t(\boldsymbol{X}(t), A(t)) + 1$
9:         Observe new state $\boldsymbol{X}(t+1)$
10:        Update $\nu_{t+1}$ according to (7)
11:        $t \leftarrow t + 1$
12:     **end while**
13:     $\tilde{T}_k \leftarrow t - t_k$
14:     **while** $\boldsymbol{X}(t) \neq 0^d$ **do**
15:         Apply action $A(t) = \pi_{\theta_k}^*(\boldsymbol{X}(t))$
16:         Observe new state $\boldsymbol{X}(t+1)$
17:     **end while**
18:     $T_k \leftarrow t - t_k$
19: **end for**
─────────────────────────────────────────────────────────────

**Remark 2.** *In Assumption 4 we can use any other policy $\pi_{\theta_2}$ such that the Markov process obtained from MDP $(\mathcal{X}, \mathcal{A}, c, P_{\theta_1})$ by following policy $\pi_{\theta_2}$ is irreducible and polynomially ergodic via the Foster-Lyapunov criteria with the uniformity discussed. Irreducibility is important as the policy will be used at times when the state is not known in advance, specifically at Steps 14-17 in Algorithm 1.*

**Assumption 5.** *We assume that $J^* := \sup_{\theta \in \Theta} J(\theta) < +\infty$.*

## 3 Thompson sampling based learning algorithm

We will use the learning algorithm Thompson sampling with dynamically-sized episodes from [49] to learn the unknown parameter $\boldsymbol{\theta}^* \in \Theta$ and the corresponding policy, $\pi_{\boldsymbol{\theta}^*}^*$, but suitably modify it for our countable state-space setting. Consider the prior distribution $\nu_0 = \nu$ defined on $\Theta$ from which $\boldsymbol{\theta}^*$ is sampled. At each time $t \in \mathbb{N}$, the posterior distribution $\nu_t$ is updated according to Bayes' rule as

$$\nu_{t+1}(d\theta) = \frac{\mathbb{P}_\theta(\boldsymbol{X}(t+1) | \boldsymbol{X}(t), A(t)) \nu_t(d\theta)}{\int_{\theta' \in \Theta} \mathbb{P}_{\theta'}(\boldsymbol{X}(t+1) | \boldsymbol{X}(t), A(t)) \nu_t(d\theta')}, \tag{7}$$

and the posterior estimate $\theta_{t+1}$, if generated, is from the posterior distribution $\nu_{t+1}$. The modified Thompson-sampling with dynamically-sized episodes algorithm (TSDE) is presented in Algorithm 1. The TSDE algorithm operates in episodes: at the beginning of episode $k$, parameter $\theta_k$ is sampled from the posterior distribution $\nu_{t_k}$ and during episode $k$, actions are generated from the stationary policy according to $\theta_k$, i.e., $\pi_{\theta_k}^*$. Let $t_k$ be the time the $k$-th episode begins. Define $\tilde{t}_{k+1}$ as the first time after $t_k$ that the conditions of Line 6 of Algorithm 1 is triggered and $t_{k+1}$ as the first time at or after $\tilde{t}_{k+1}$ where state $0^d$ is visited; for the last episode started before or at $T$, we ensure that $t_k$ and $\tilde{t}_k$ are less than or equal $T + 1$. Explicitly, $t_1 = 1$ and for $k > 1$, $t_k = \min\{t \geq \tilde{t}_k : \boldsymbol{X}(t) = 0^d \text{ or } t > T\}$. Let $T_k = t_{k+1} - t_k$ be the length of the $k$-th episode and set $\tilde{T}_k = \tilde{t}_{k+1} - t_k$ with the convention $\tilde{T}_0 = 1$. For any state-action pair $(\boldsymbol{x}, a)$, we define $N_1(\boldsymbol{x}, a) = 0$ and for $t > 1$,

$$N_t(\boldsymbol{x}, a) = \left| \{t_k \leq i < \tilde{t}_{k+1} \leq t \text{ for some } k \geq 1 : (\boldsymbol{X}(i), A(i)) = (\boldsymbol{x}, a)\} \right|.$$

Notice that for all state-action pairs $(\boldsymbol{x}, a)$ and $\tilde{t}_{k+1} \leq t \leq t_{k+1}$, we have $N_t(\boldsymbol{x}, a) = N_{\tilde{t}_{k+1}}(\boldsymbol{x}, a)$. We denote $K_T$ as the number of episodes started by or at time $T$, or $K_T = \max\{k : t_k \leq T\}$. The length of episode $k < K_T$ is not fixed and is determined according to two stopping criteria: (1) $t > t_k + \tilde{T}_{k-1}$, (2) $N_t(\boldsymbol{x}, a) > 2N_{t_k}(\boldsymbol{x}, a)$ for some state-action pair $(\boldsymbol{x}, a)$. After either criterion is met, the system will still follow policy $\pi_{\theta_k}^*$ until the first time at which state $0^d$ is visited; see Line 14 and Figure 1. We use this settling period to $0^d$ because the system state can be arbitrary when the

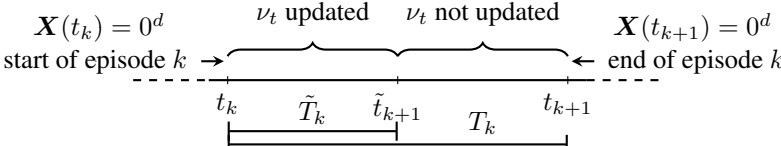

Figure 1: MDP evolution in episode $k < K_T$.

first stopping criterion is met. As the countable state-space setting precludes a simple union-bound argument to overcome this uncertainty (as in the literature for finite state settings), we let the system reach the special state $0^d$. Another (essentially equivalent) option is to wait until the state hits the finite set $C_*^g$ or $C_*^p$ and then use a union bound argument for all states in either set. For analytical convenience, we only use the state samples observed before arrival $\tilde{t}_{k+1}$ to update the posterior distribution. The posterior update is halted during the settling period to $0^d$ as we have no control on the states visited during it, despite it being finite in duration (by our assumptions).

## 4   Regret analysis of Algorithm 1

The performance of any learning policy $\pi_L$ is evaluated using the metric of expected regret compared to the optimal expected average cost of true parameter $\boldsymbol{\theta}^*$, namely, $J(\boldsymbol{\theta}^*)$. In this section, we evaluate the performance of Algorithm 1 and derive an upper bound for $R(T, \pi_{TSDE})$, its expected regret up to time $T$. In Section 2, we argued that at time $t$ in episode $k$ ($t_k \le t < t_{k+1}$), there exist a constant $J(\theta_k)$ and a unique function $v(\cdot; \theta_k) : \mathcal{X} \to \mathbb{R}$ such that $v\left(0^d; \theta_k\right) = 0$ and

$$J(\theta_k) + v(\boldsymbol{X}(t); \theta_k) = c(\boldsymbol{X}(t), \pi_{\theta_k}^*(\boldsymbol{X}(t))) + \sum_{\boldsymbol{y} \in \mathcal{X}} P_{\theta_k}(\boldsymbol{y}|\boldsymbol{X}(t), \pi_{\theta_k}^*(\boldsymbol{X}(t)))v(\boldsymbol{y}; \theta_k), \quad (8)$$

in which $\pi_{\theta_k}^*$ is the optimal or best-in-class policy (depending on the context) according to parameter $\theta_k$ and $J(\theta_k)$ is the average cost for the Markov process obtained from MDP $(\mathcal{X}, \mathcal{A}, c, P_{\theta_k})$ by following $\pi_{\theta_k}^*$. We derive a bound for the expected regret $R(T, \pi_{TSDE})$ following the proof steps of [49] while extending it to the countable state-space setting of our problem. Using (8), the regret is decomposed into three terms and each term is bounded separately:

$$R(T, \pi_{TSDE}) = \mathbb{E}\Big[\sum_{k=1}^{K_T} \sum_{t=t_k}^{t_{k+1}-1} c(\boldsymbol{X}(t), \pi_{\theta_k}^*(\boldsymbol{X}(t)))\Big] - T\,\mathbb{E}\left[J\left(\boldsymbol{\theta}^*\right)\right] = R_0 + R_1 + R_2, \quad (9)$$

$$\text{with } R_0 = \mathbb{E}\Big[\sum_{k=1}^{K_T} T_k J(\theta_k)\Big] - T\,\mathbb{E}[J(\boldsymbol{\theta}^*)], \quad (10)$$

$$R_1 = \mathbb{E}\Big[\sum_{k=1}^{K_T} \sum_{t=t_k}^{t_{k+1}-1} \big[v(\boldsymbol{X}(t); \theta_k) - v(\boldsymbol{X}(t+1); \theta_k)\big]\Big], \quad (11)$$

$$R_2 = \mathbb{E}\Big[\sum_{k=1}^{K_T} \sum_{t=t_k}^{t_{k+1}-1} \big[v(\boldsymbol{X}(t+1); \theta_k) - \sum_{\boldsymbol{y} \in \mathcal{X}} P_{\theta_k}(\boldsymbol{y}|\boldsymbol{X}(t), \pi_{\theta_k}^*(\boldsymbol{X}(t)))v(\boldsymbol{y}; \theta_k)\big]\Big]. \quad (12)$$

Before bounding the above regret terms, we address the complexities arising from the countable state-space setting. Firstly, we need to study the maximum state (with respect to the $\ell_\infty$-norm) visited up to time $T$ in the MDP $(\mathcal{X}, \mathcal{A}, c, P_{\boldsymbol{\theta}^*})$ following Algorithm 1; we denote this maximum state by $M_{\boldsymbol{\theta}^*}^T$. In Appendix C, we derive upper bounds on the moments of hitting times of state $0^d$ and utilize this to bound the moments of random variable $M_{\boldsymbol{\theta}^*}^T$, which then lets us study the number of episodes $K_T$ by time $T$. Another challenge in analyzing the regret is that the relative value function $v(\boldsymbol{x}; \theta)$ is unlikely to be bounded in the countable state-space setting. Hence, in (13) and (14), we find bounds for the relative value function in terms of hitting time $\tau_{0^d}$ from the initial state $\boldsymbol{x}$. Based on these results, we provide an upper bound for the regret of Algorithm 1 in Theorem 1.

***Maximum state norm under polynomial and geometric ergodicity.*** Here we state the results that characterize the maximum $l_\infty$-norm of the state vector achieved up until and including time $T$, and

the resulting bounds on the number of episodes executed until time $T$. Owing to space constraints the details (including formal statements) are presented in Appendix B. The results are listed as below:

(i) In Lemma 6, we bound the moments of the maximum length of recurrence times of state $0^d$, using the ergodicity assumptions 3 and 4. This, along with the skip-free property, allows us to prove that the $p$-th moment of $\max_{1 \le i \le T} \tau_{0^d}^{(i)}$ and $M_{\boldsymbol{\theta}^*}^T$ are both of order $O(\log^p T)$.

(ii) In Lemma 7, we find an upper bound for the number of episodes in which the second stopping criterion is met or there exists a state-action pair for which $N_t(\boldsymbol{x}, a)$ has increased more than twice.

(iii) In Lemma 8, we bound the total number of episodes $K_T$ by time $T$ by bounding the number of episodes triggered by the first stopping criterion, using the fact that in such episodes, $\tilde{T}_k = \tilde{T}_{k-1} + 1$. Moreover, to account for the settling time of each episode, we use geometric ergodicity and Lemma 6. It follows that the expected value of the number of episodes $K_T$ is of the order $\tilde{O}(\sqrt{h^d |\mathcal{A}| T})$.

*Regret analysis.* Next, we bound regret terms $R_0$, $R_1$ and $R_2$ using the approach of [49] along with additional arguments to extend their result to a countably infinite state-space. We consider the relative value function $v(\boldsymbol{x}; \theta)$ of policy $\pi_\theta^*$ introduced for the optimal policy in ACOE or for the best in-class policy in the Poisson equation. In either of these cases, policy $\pi_\theta^*$ satisfies (5), which is the corresponding Poisson equation with forcing function $c(\boldsymbol{x}, \pi_\theta^*(\boldsymbol{x}))$ in a Markov chain with transition matrix $P_\theta^{\pi_\theta^*}$. In (6), we presented the solution $(J, v)$ to the Poisson equation, which yields the following upper bound for the relative value function, as argued in Appendix A.3:

$$v(\boldsymbol{x}; \theta) \le \bar{C}^{\pi_\theta^*}(\boldsymbol{x}) \le \mathbb{E}_{\boldsymbol{x}}^{\pi_\theta^*} \left[ Kd \left( \|\boldsymbol{x}\|_\infty + h\tau_{0^d} \right)^r \tau_{0^d} \right]. \tag{13}$$

We can similarly lower bound the relative value function using Assumption 5 as

$$v(\boldsymbol{x}; \theta) \ge -J(\theta) \mathbb{E}_{\boldsymbol{x}}^{\pi_\theta^*}[\tau_{0^d}] \ge -J^* \mathbb{E}_{\boldsymbol{x}}^{\pi_\theta^*}[\tau_{0^d}]. \tag{14}$$

From Assumption 3, all moments of $\tau_{0^d}$ and thus, the derived bounds are finite. Also, in Lemma 10 we bound the moments of $\tau_{0^d}$ of order $i \le r + 1$ using the polynomial Lyapunov function $V_{\theta_1, \theta_2}^p$, which is then used to bound the expected regret. We next bound the first regret term $R_0$ from the first stopping criterion in terms of the number of episodes $K_T$ and the settling time of each episode $k$.

**Lemma 1.** *The first regret term $R_0$ satisfies $R_0 \le J^* \mathbb{E}[K_T (\max_{1 \le i \le T} \tau_{0^d}^{(i)} + 1)]$.*

Proof of Lemma 1 is given in Appendix B.4. From Lemma 6, all moments of $\max_{1 \le i \le T} \tau_{0^d}^{(i)}$ are bounded by a polylogarithmic function. Futhermore, as a result of Lemma 8, expected value of the number of episodes $K_T$ is of the order $\tilde{O}(\sqrt{h^d |\mathcal{A}| T})$, which leads to a $\tilde{O}(\sqrt{h^d |\mathcal{A}| T})$ regret term $R_0$. Next, an upper bound on $R_1$ defined in (11) is derived. In the proof of Lemma 2 we argue that as the relative value function is equal to 0 at all time instances $t_k$ for $k \le K_T$, the only term that contributes to the regret is the value function at the end of time horizon $T$. We use the lower bound derived in (14) to show that the second regret term $R_1$ is $\tilde{O}(1)$; the proof is given in Appendix B.5.

**Lemma 2.** *The second regret term $R_1$ satisfies $R_1 \le c_2 \mathbb{E}[(M_{\boldsymbol{\theta}^*}^T)^{r_*^p}] + c_3$, where $c_2 = J^* 2^{r_*^p} s_*^p (\beta_*^p)^{-1}$ and $c_3 = J^* (\beta_*^p)^{-1} \left( s_*^p (2h)^{r_*^p} + b_*^p (K_*)^{-1} \right)$.*

From Lemma 6, $\mathbb{E}[(M_{\boldsymbol{\theta}^*}^T)^{r_*^p}]$ is $O(\log^{r_*^p} T)$; hence, $R_1$ is upper bounded by a polylogarithmic function of the order $r_*^p$. Finally, in Lemma 3, we derive an upper bound for the third regret term $R_2$ defined in (12) using the bound derived for the relative value function in (13). To bound $R_2$, we characterize it in terms of the difference between the empirical and true unknown transition kernel and following the concentration method used in [64, 10, 49, 7, 8], we argue that with high probability the total variation distance between the two distributions is small; for proof, see Appendix B.6.

**Lemma 3.** *For problem-dependent constant $c_{p_3}$ and polynomial $Q(T) = c_{p_3} (Th)^{r + r_*^p} / 48$, we have*

$$R_2 \le (\log(hT + h) + 1)^d + c_{p_3} \sqrt{|\mathcal{A}| T} \log_2 \left( 2|\mathcal{A}| T^2 Q(T) \right) \mathbb{E} \left[ (M_{\boldsymbol{\theta}^*}^T + h)^{d + r + r_*^p} \left( \max_{1 \le i \le T} \tau_{0^d}^{(i)} \right) \right].$$

The above Lemma results in a $\tilde{O}(Krd J^* h^{d + 2r + r_*^p} \sqrt{|\mathcal{A}| T})$ regret term as a result of Lemma 6, where $h$ is the skip-free parameter defined in Assumption 2, $d$ is the dimension of the state-space, $K$ and $r$ are the cost function parameters defined in Assumption 1, $J^*$ is the supremum on the optimal cost, $r_*^p$ is defined in Assumption 4, and where $\tilde{O}$ hides logarithmic factors in problem parameters one of which is $\log^{d + r + r_*^p + 2}(T)$. For simplicity, we have not included the Lyapunov

functions related parameters in the regret. Finally, from Lemmas 1, 2, 3, along with the Cauchy-Schwarz inequality, we conclude that the regret of Algorithm 1 $R(T, \pi_{TSDE})(= R_0 + R_1 + R_2)$ is $\tilde{O}(KrdJ^*h^{d+2r+r_*^p}\sqrt{|\mathcal{A}|T})$; for brevity, we will state that regret is of the order $\tilde{O}(dh^d\sqrt{|\mathcal{A}|T})$.

**Theorem 1.** *Under Assumptions 1-5, the regret of Algorithm 1, $R(T, \pi_{TSDE})$, is $\tilde{O}(dh^d\sqrt{|\mathcal{A}|T})$.*

Theorem 1 can be extended to the problem of finding the best policy within a sub-class of policies in set $\Pi$, which may or may not contain the optimal policy. In Section 2, we stated that Assumptions 3 and 4 hold for policies in $\Pi$ and we used this to argue that the Poisson equation has a solution given in (6). As a result, repeating the same arguments as in Theorem 1 with the modification that $\pi_\theta^*$ is the best in-class policy of the MDP governed by parameter $\theta$, yields the following corollary.

**Corollary 1.** *Under Assumptions 1 through 5, the regret of Algorithm 1 when using the best in-class policy is $\tilde{O}(dh^d\sqrt{|\mathcal{A}|T})$.*

**Requirement of an optimal policy oracle.** To implement our algorithm, we need to find the optimal policy for each model sampled by the algorithm—optimal policy for Theorem 1 and optimal policy within policy class $\Pi$ for Corollary 1. In the finite state-space setting, [49] provides a schedule of $\epsilon$ values and selects $\epsilon$-optimal policies to obtain $\tilde{O}(\sqrt{T})$ regret guarantees. The issue with extending the analysis of [49] to the countable state-space setting is that we need to ensure (uniform) ergodicity for the chosen $\epsilon$-optimal policies. Another issue is that, to the best of our knowledge, there isn't a general structural characterization of all $\epsilon$-optimal stationary policies for countable state-space MDPs or even a characterization of the policy within this set that is selected by any computational procedure in the literature; current results only discuss characterization of the stationary optimal policy. In the absence of such results, stability assumptions with the same uniformity across models as in our submission will be needed, which are likely too strong to be useful. However, if we could verify the stability requirements of Assumptions 3 and 4 for a subset of policies, the optimal oracle is not needed, and instead, by choosing approximately optimal policies within this subset, we can follow the same proof steps as [49] to guarantee regret performance similar to Corollary 1 (without knowledge of model parameters). Thus, in Theorem 2 we extend the previous regret guarantees to the algorithm employing $\epsilon$-optimal policy; proof is given in Appendix B.8.

**Theorem 2.** *Consider a non-negative sequence $\{\epsilon_k\}_{k=1}^\infty$ such that for every $k \in \mathbb{N}$, $\epsilon_k$ is bounded above by $\frac{1}{k+1}$ and an $\epsilon_k$-optimal policy satisfying Assumptions 3 and 4 is given. The regret incurred by Algorithm 1 while using the $\epsilon_k$-optimal policy during any episode $k$ is $\tilde{O}(dh^d\sqrt{|\mathcal{A}|T})$.*

## 5 Evaluation and Conclusion: Application of Algorithm 1 to queueing models

Next, we present an evaluation of our algorithm. We study two different queueing models shown in Figure 2, each with Poisson arrivals at rate $\lambda$, and two heterogeneous servers with exponentially distributed services times with unknown service rate vector $\boldsymbol{\theta}^* = (\theta_1^*, \theta_2^*)$. Vector $\boldsymbol{\theta}^*$ is sampled from the prior distribution $\nu$ defined on the space $\Theta$ given as $\Theta = \{(\theta_1, \theta_2) \in \mathbb{R}_+^2 : \frac{\lambda}{\theta_1+\theta_2} \leq \frac{1-\delta}{1+\delta}, 1 \leq \frac{\theta_1}{\theta_2} \leq R\}$, for fixed $R \geq 1$ and $\delta \in (0, 0.5)$. The first condition ensures the stability of the queueing models, while the second guarantees the compactness of the parameter space of the parameterized policies. In both systems, the goal of the dispatcher is to minimize the expected sojourn time of jobs, which by Little's law [52] is equivalent to minimizing the average number of jobs in the system. After verifying Assumptions 1-5 in Appendix E for the cost function $c(\boldsymbol{x}) = \|\boldsymbol{x}\|_1$, Theorem 1 yields a Bayesian regret of order $\tilde{O}(\sqrt{|\mathcal{A}|T})$ for Algorithm 1.

***Model 1. Two-server queueing system with a common buffer.*** We consider the continuous-time queueing system of Figure 2a, where the countable state space is $\mathcal{X} = \{\boldsymbol{x} = (x_0, x_1, x_2) \in \mathbb{Z}_+ \times \{0, 1\}^2\}$, where $x_0$ is the queue length, and $x_i$, $i = 1, 2$ equal 1 if server $i$ is busy. The action space is $\mathcal{A} = \{h, b, 1, 2\}$, where $h$ means no action, $b$ sends a job to both servers, and $i = 1, 2$ assigns a job to server $i$. In [38], it is shown that by uniformization [39] and sampling the continuous-time Markov process at rate $\lambda + \theta_1^* + \theta_2^*$, a discrete-time Markov chain is obtained, which converts the original continuous-time problem to an equivalent discrete-time problem where we need to minimize $\limsup_{T\to\infty} T^{-1} \sum_{t=0}^{T-1} \|\boldsymbol{X}(t)\|_1$. Further, [38] shows that the optimal policy is a threshold policy $\pi_{t(\boldsymbol{\theta}^*)}$ with optimal finite threshold $t(\boldsymbol{\theta}^*) \in \mathbb{N}$: always assign a job to the faster (first) server when free, and to the second server if it is free and $\|\boldsymbol{x}\|_1 > t(\boldsymbol{\theta}^*)$, and take no action otherwise. In

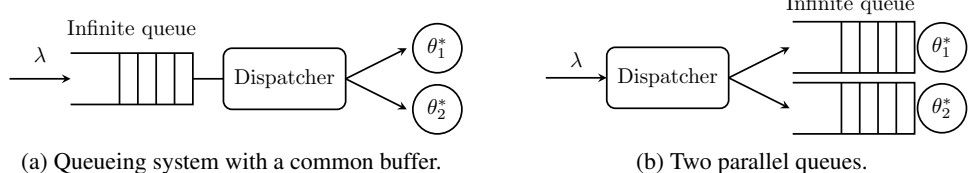

(a) Queueing system with a common buffer.     (b) Two parallel queues.

Figure 2: Two-server queueing systems with heterogeneous service rates.

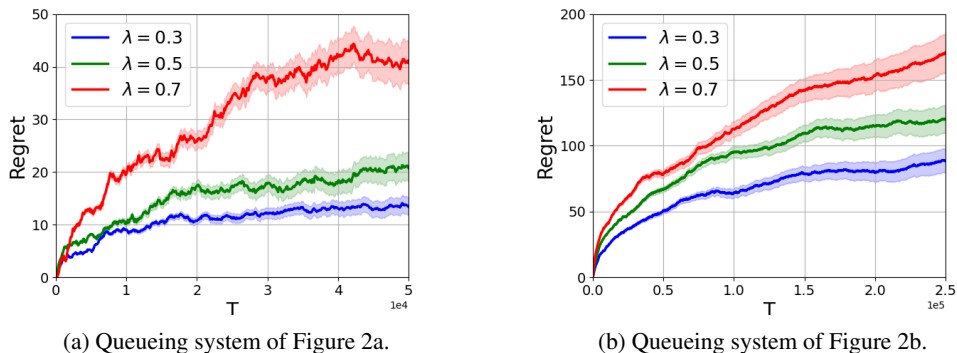

(a) Queueing system of Figure 2a.     (b) Queueing system of Figure 2b.

Figure 3: Regret performance for $\lambda = 0.3, 0.5, 0.7$. Shaded region shows the $\pm\sigma$ area of mean regret.

Appendix E.1, we argue that the discrete-time Markov process governed by $\theta \in \Theta$ and following threshold policy $\pi_t$ for any threshold $t$ belonging to a compact set satisfies Assumptions 1-5.

***Model 2. Two heterogeneous parallel queues.*** We consider the continuous-time queueing system of Figure 2b with countable state space $\mathcal{X} = \{\boldsymbol{x} = (x_1, x_2) \in \mathbb{Z}_+^2\}$, where $x_i$ is the number of jobs in the server-queue pair $i$. The action space is $\mathcal{A} = \{1, 2\}$, where action $i$ sends the arrival to queue $i$. We obtain the discrete-time MDP by sampling the queueing system at the arrivals, and then aim to find the average cost minimizing policy within the class $\Pi = \{\pi_\omega; \omega \in [(c_R R)^{-1}, c_R R]\}$, $c_R \geq 1$. Policy $\pi_\omega : \mathcal{X} \to \mathcal{A}$ routes arrivals based on the weighted queue lengths: $\pi_\omega(\boldsymbol{x}) = \arg\min(1 + x_1, \omega(1 + x_2))$ with ties broken for 1. Even with the transition kernel fully specified (by the values of arrival and service rates), the optimal policy in $\Pi$ is not known except when $\theta_1 = \theta_2$ where the optimal value is $\omega = 1$, and so, to learn it, we will use Proximal Policy Optimization for countable state-space MDPs [18]. Note that [18] requires full model knowledge, which holds in our scheme as we use parameters sampled from the posterior for choosing the policy at the beginning of each episode. In Appendix E.2, we argue that the discrete-time Markov process governed by parameter $\theta \in \Theta$ and following policy $\pi_\omega$ for $\omega \in [(c_R R)^{-1}, c_R R]$ satisfies Assumptions 1-5.

Next, we report the numerical results of Algorithm 1 in the two queueing models of Figure 2 and calculate regret using (2). The regret is averaged over 2000 simulation runs and plotted against the number of transitions in the sampled discrete-time Markov process. Figure 3 shows the behavior of the regret of the two queuing models for three different arrival rates and service rates distributed according to a Dirichlet prior over $[0.5, 1.9]^2$. We observe that the regret is sub-linear in time and grows as the arrival rate increases. For the queueing model of Figure 2a, the minimum average cost $J(\theta)$ and optimal policy $\pi_\theta^*$ are known explicitly [38] for every $\theta \in \Theta$, which are used in Algorithm 1 and for regret calculation. Conversely, for the second queueing model, $J(\theta)$ and $\pi_\theta^*$ are not known. The PPO algorithm [18] is used to empirically find both the optimal weight and the policy's average cost. Additional details of the simulations and more plots are presented in Appendix G.

**Conclusions and future work.** We studied the problem of learning optimal policies in countable state-space MDPs governed by unknown parameters. We proposed a learning policy based on Thompson sampling and established finite-time performance guarantees on the Bayesian regret. We highlighted the practicality of our proposed algorithm by considering two different queuing models and showing that our algorithm can be applied to develop optimal control policies. For future work we plan two directions to explore: to generalize our algorithm to consider polices that might not all be stabilizing, and also to simplify the algorithm using ideas from [61, 57].

## Disclosure of Funding

SA's research was supported by NSF via grants ECCS2038416, CCF2008130, and CNS1955777, and a grant from General Dynamics via MIDAS at the University of Michigan, Ann Arbor. VS's research is supported in part by NSF via grants ECCS2038416, CCF2008130, CNS1955777, and CMMI2240981.

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

# A  Proofs related to problem formulation

## A.1  Ergodicity definitions

Suppose that Markov process $\boldsymbol{X}$ on $\mathcal{X}$ with transition kernel $P$ is irreducible, aperiodic and positive recurrent with stationary distribution $\mu$ and let $f : \mathcal{X} \mapsto [1, \infty)$ be a measurable function such that $\mu(f) := \mathbb{E}_\mu[f(Y)] < +\infty$ with $Y \sim \mu$. We are interested in conditions under which for a sequence of positive numbers $\rho := (\rho(n))_{n \geq 0}$,

$$\lim_{n \to \infty} \rho(n) \| P^n(\boldsymbol{x}, \cdot) - \mu(\cdot) \|_f = 0, \qquad \forall \boldsymbol{x} \in \mathcal{X}, \tag{15}$$

where for a signed measure $\tilde{\mu}$ on $\mathcal{X}$, $\| \tilde{\mu} \|_f := \sup_{|g| \leq f} | \tilde{\mu}(g) |$. The sequence $\rho$ is interpreted as the rate function, and three different notions of ergodicity are distinguished based on the following rate functions: $\rho(n) \equiv 1$, $\rho(n) = \zeta^n$ for $\zeta > 1$, and $\rho(n) = n^{\zeta - 1}$ for $\zeta \geq 1$. Further, for each rate function $\rho$, we state the Foster-Lyapunov characterization of ergodicity of the Markov process $\boldsymbol{X}$, which provides sufficient conditions for (15) to hold.

1. If $\rho(n) \equiv 1$ for all $n \geq 0$, the Markov process $\boldsymbol{X}$ satisfying (15) is said to be $f$-**ergodic**. From [43], for an irreducible and aperiodic chain, $f$-ergodicity is *equivalent* to the existence of a function $V : \mathcal{X} \mapsto [0, \infty)$, a finite set $C$, and positive constant $b$ such that

   $$\Delta V \leq -f + b \mathbb{I}_C, \tag{16}$$

   where $\Delta V := PV - V$ with $PV(\boldsymbol{x}) := \sum_{\boldsymbol{x}' \in \mathcal{X}} P(\boldsymbol{x}, \boldsymbol{x}') V(\boldsymbol{x}')$. The drift condition (16) implies positive recurrence of the Markov process, existence of a unique stationary distribution $\mu$, and $\mu(f) \leq b < +\infty$ ([43], Theorem 14.3.7).

2. If $\rho(n) = \zeta^n$ for some $\zeta > 1$, the Markov process $\boldsymbol{X}$ satisfying (15) is said to be $f$-**geometrically ergodic**. From [43], for an irreducible and aperiodic chain, $f$-geometric ergodicity is *equivalent* to the existence of a function $V : \mathcal{X} \mapsto [1, \infty)$, a finite set $C$, a constant $\gamma \in (0, 1)$ and positive constant $b$ such that

   $$\Delta V \leq -(1 - \gamma)V + b \mathbb{I}_C. \tag{17}$$

   The drift condition (17) implies positive recurrence of the Markov process, existence of a unique stationary distribution $\mu$, and $\mu(V) \leq \frac{b}{1-\gamma} < +\infty$ ([43], Theorem 14.3.7). Moreover, if $f(\cdot) \equiv 1$ in (15), then the Markov process $\boldsymbol{X}$ is called **geometrically ergodic**.

3. If $\rho(n) = n^{\zeta - 1}$ for some $\zeta \geq 1$, the Markov process $\boldsymbol{X}$ satisfying (15) is said to be $f$-**polynomially ergodic**. From [43, 29], for an irreducible and aperiodic chain, the existence of a function $V : \mathcal{X} \mapsto [1, \infty)$, a finite set $C$, a constant $\alpha \in [0, 1)$, and positive constants $c$ and $b$ such that

   $$\Delta V \leq -c V^\alpha + b \mathbb{I}_C \tag{18}$$

   *implies* $V_\zeta$-polynomial ergodicity of $\boldsymbol{X}$ at rate $\rho(n) = n^{\zeta - 1}$ for all $\zeta \in [1, 1/(1 - \alpha)]$ with $V_\zeta = V^{1 - \zeta(1 - \alpha)}$. The drift condition (18) implies positive recurrence of the Markov process, existence of a unique stationary distribution $\mu$, and $\mu(V^\alpha) \leq \frac{b}{c} < +\infty$.

## A.2  Lemma 4

**Lemma 4.** *For any state $\boldsymbol{x} \neq 0^d$, there exists constants $\kappa > 1$ and $c_1$ such that the following holds for the hitting time of state $0^d$, $\tau_{0^d}$,*

$$\mathbb{E}_{\boldsymbol{x}}[\kappa^{\tau_{0^d}}] \leq c_1 V^g(\boldsymbol{x}).$$

*Proof.* We define $\tilde{V} := \sum_{n=0}^\infty {}_{0^d}P^n V^g$ where ${}_{0^d}P^n$ is the $n$-step taboo probability [43] defined as

$$_A P^n_{\boldsymbol{x}B} = \mathbb{P}_{\boldsymbol{x}} \left( \boldsymbol{X}_n \in B, \tau_A > n \right),$$

for $A, B \subseteq \mathcal{X}$, and $\tau_A$ is the first hitting time of set $A$. We also let ${}_A P^0_{\boldsymbol{x}B} = \mathbb{I}_B(\boldsymbol{x})$. We have

$$_{0^d}P\tilde{V}(\boldsymbol{x}) = \sum_{\boldsymbol{y} \neq 0^d} P_{\boldsymbol{xy}} \tilde{V}(\boldsymbol{y}) = \sum_{n=0}^\infty \sum_{\boldsymbol{y}, \boldsymbol{z} \neq 0^d} P_{\boldsymbol{xy}} \, {}_{0^d}P^n_{\boldsymbol{yz}} V^g(\boldsymbol{z})$$

$$= \sum_{n=0}^\infty \sum_{\boldsymbol{z} \neq 0^d} {}_{0^d}P^{n+1}_{\boldsymbol{xz}} V^g(\boldsymbol{z}) = \tilde{V}(\boldsymbol{x}) - V^g(\boldsymbol{x}).$$

In Appendix D.3, we argue that there exists $\tilde{b}^g > 1$ such that $\tilde{V}(\boldsymbol{y}) \leq \tilde{b}^g V^g(\boldsymbol{y})$ for all $\boldsymbol{y} \in \mathcal{X}$, which leads to

$$_{0^d} P\tilde{V} = \tilde{V} - V^g \leq \tilde{V} - \frac{1}{\tilde{b}^g}\tilde{V} = \left(1 - \frac{1}{\tilde{b}^g}\right)\tilde{V}. \tag{19}$$

Define Lyapunov function

$$\tilde{V}^g(\boldsymbol{x}) = \begin{cases} (1 + 2\tilde{b}^g)\tilde{V}(\boldsymbol{x}), & \text{if } \boldsymbol{x} \neq 0^d, \\ 1 + \left(2\tilde{b}^g\right)^{-1}, & \text{if } \boldsymbol{x} = 0^d. \end{cases}$$

From the above equation and (19), we get

$$\begin{aligned}
P\tilde{V}^g(\boldsymbol{x}) &= \sum_{\boldsymbol{y} \neq 0^d} P_{\boldsymbol{xy}}\tilde{V}^g(\boldsymbol{y}) + P_{\boldsymbol{x}0^d}\tilde{V}^g(0^d) \\
&= \sum_{\boldsymbol{y} \neq 0^d} P_{\boldsymbol{xy}}(1 + 2\tilde{b}^g)\tilde{V}(\boldsymbol{y}) + P_{\boldsymbol{x}0^d}\left(1 + \frac{1}{2\tilde{b}^g}\right) \\
&\leq \left(1 - \frac{1}{\tilde{b}^g}\right)(1 + 2\tilde{b}^g)\tilde{V}(\boldsymbol{x}) + 1 + \frac{1}{2\tilde{b}^g} \\
&\leq \left(1 - \frac{1}{\tilde{b}^g}\right)(1 + 2\tilde{b}^g)\tilde{V}(\boldsymbol{x}) + \left(1 + \frac{1}{2\tilde{b}^g}\right)\tilde{V}(\boldsymbol{x}) \\
&= \left(1 - \frac{1}{2\tilde{b}^g}\right)(1 + 2\tilde{b}^g)\tilde{V}(\boldsymbol{x}).
\end{aligned}$$

Thus,

$$P\tilde{V}^g(\boldsymbol{x}) \leq \left(1 - \frac{1}{2\tilde{b}^g}\right)\tilde{V}^g(\boldsymbol{x}) + \left(1 - \frac{1}{2\tilde{b}^g}\right)(1 + 2\tilde{b}^g)\tilde{V}(0^d)\mathbb{I}_{0^d}(\boldsymbol{x}), \quad \boldsymbol{x} \in \mathcal{X}.$$

To find an upper bound for $\mathbb{E}_{\boldsymbol{x}}[\kappa^{\tau_{0^d}}]$, we apply [43, Theorem 15.2.5], which is a generalization of Lemma 12. For any $1 \leq \kappa \leq \frac{2\tilde{b}^g}{2\tilde{b}^g - 1}$, there exists $\epsilon > 0$ such that

$$\mathbb{E}_{\boldsymbol{x}}\left[\sum_{i=0}^{\tau_{0^d}-1} \tilde{V}^g(\boldsymbol{X}_i)\kappa^i\right] \leq \epsilon^{-1}\kappa^{-1}\tilde{V}^g(\boldsymbol{x}).$$

As $\tilde{V}^g(\boldsymbol{y}) \geq 1$ for all $\boldsymbol{y} \in \mathcal{X}$, we have

$$\begin{aligned}
\mathbb{E}_{\boldsymbol{x}}[\kappa^{\tau_{0^d}}] &\leq \kappa \mathbb{E}_{\boldsymbol{x}}\left[\sum_{i=0}^{\tau_{0^d}-1} \tilde{V}^g(\boldsymbol{X}_i)\kappa^i\right] \leq \epsilon^{-1}\tilde{V}^g(\boldsymbol{x}) \\
&= \epsilon^{-1}\left(1 + 2\tilde{b}^g\right)\tilde{V}(\boldsymbol{x}) \leq \tilde{b}^g\epsilon^{-1}\left(1 + 2\tilde{b}^g\right)V^g(\boldsymbol{x}),
\end{aligned}$$

and the claim holds for any $\kappa \in [1, \frac{2\tilde{b}^g}{2\tilde{b}^g - 1}]$ and $c_1 = \tilde{b}^g\epsilon^{-1}\left(1 + 2\tilde{b}^g\right)$. $\qquad\square$

### A.3 Poisson equation

For an irreducible Markov process on the countably-infinite space $\mathcal{X}$ with time-homogeneous transition kernel $P$ and cost function $\bar{c}(\cdot)$, a solution pair to the Poisson equation [41] is a scalar $J$ and function $v(\cdot) : \mathcal{X} \mapsto \mathbb{R}$ such that $J + v = \bar{c} + Pv$, where $v(\boldsymbol{z}) = 0$ for some $\boldsymbol{z} \in \mathcal{X}$. If the Markov process is also positive recurrent and $\mathbb{E}_{\boldsymbol{x}}\left[\sum_{i=0}^{\tau_{\boldsymbol{y}}-1} |\bar{c}(\boldsymbol{X}(i))|\right] < \infty$, where $\tau_{\boldsymbol{y}}$ is the first hitting time of some state $\boldsymbol{y} \in \mathcal{X}$, then solution pair $(J, v)$ given as

$$J = \frac{\mathbb{E}_{\boldsymbol{y}}\left[\sum_{i=0}^{\tau_{\boldsymbol{y}}-1} |\bar{c}(\boldsymbol{X}(i))|\right]}{\mathbb{E}_{\boldsymbol{y}}[\tau_{\boldsymbol{y}}]} \text{ and } v(\boldsymbol{x}) = \mathbb{E}_{\boldsymbol{y}}\left[\sum_{i=0}^{\tau_{\boldsymbol{x}}-1} |\bar{c}(\boldsymbol{X}(i))|\right] - J\mathbb{E}_{\boldsymbol{x}}[\tau_{\boldsymbol{y}}], \quad \forall \boldsymbol{x} \in \mathcal{X},$$

is a solution to the Poisson equation $J + v = \bar{c} + Pv$ with $v(\boldsymbol{z}) = 0$ [41, Theorem 9.5].

**Lemma 5.** *Consider Markov Decision Processes $(\mathcal{X}, \mathcal{A}, c, P_\theta)$ governed by parameter $\theta \in \Theta$ following the best-in-class policy $\pi_\theta^*$. Then the pair $\left(J(\theta), v^{\pi_\theta^*}\right)$ given as*

$$J(\theta) := \frac{\bar{C}^{\pi_\theta^*}(0^d)}{\mathbb{E}_{0^d}^{\pi_\theta^*}[\tau_{0^d}]} \text{ and } v^{\pi_\theta^*}(\boldsymbol{x}) = \bar{C}^{\pi_\theta^*}(\boldsymbol{x}) - J(\theta)\mathbb{E}_x^{\pi_\theta^*}[\tau_{0^d}], \quad \forall \boldsymbol{x} \in \mathcal{X},$$

*is a solution to the Poisson equation $v + J = c + P_\theta^{\pi_\theta^*} v$, where $v^{\pi_\theta^*}(0^d) = 0$ and $\bar{C}^{\pi_\theta^*}(\boldsymbol{x}) = \mathbb{E}_{\boldsymbol{x}}^{\pi_\theta^*}\left[\sum_{i=0}^{\tau_{0^d}-1} c(\boldsymbol{X}(i), \pi_\theta^*(\boldsymbol{X}(i)))\right]$.*

*Proof.* From [41, Theorem 9.5], a solution pair to the Poisson equation exists if $\mathbb{E}_{\boldsymbol{x}}^{\pi_\theta^*}[\tau_{0^d}]$ and $\bar{C}^{\pi_\theta^*}(\boldsymbol{x})$ are finite for all $\boldsymbol{x} \in \mathcal{X}$. The former follows from positive recurrence assumed in Assumption 3 and for the latter, from Assumptions 1 and 2,

$$\bar{C}^{\pi_\theta^*}(\boldsymbol{x}) = \mathbb{E}_{\boldsymbol{x}}^{\pi_\theta^*}\left[\sum_{i=0}^{\tau_{0^d}-1} c(\boldsymbol{X}(i), \pi_\theta^*(\boldsymbol{X}(i)))\right] \leq \mathbb{E}_{\boldsymbol{x}}^{\pi_\theta^*}\left[\sum_{i=0}^{\tau_{0^d}-1}\sum_{j=1}^{d} K(X_j(i))^r\right]$$

$$\leq \mathbb{E}_{\boldsymbol{x}}^{\pi_\theta^*}\left[\sum_{i=0}^{\tau_{0^d}-1} Kd(\|\boldsymbol{x}\|_\infty + hi)^r\right] \leq \mathbb{E}_{\boldsymbol{x}}^{\pi_\theta^*}\left[Kd(\|\boldsymbol{x}\|_\infty + h\tau_{0^d})^r \tau_{0^d}\right],$$

which is finite from geometric ergodicity (Assumption 3) and the discussion following that. $\quad\square$

## B  Proofs of regret analysis

In this section, we state the proofs related to regret analysis of Section 4. We first note a key property of Thompson sampling from [49], which states that for any episode $k$, measurable function $f$, and $\mathcal{H}_{t_k}$−measurable random variable $Y$, we have

$$\mathbb{E}\left[f(\theta_k, Y)\right] = \mathbb{E}\left[f(\boldsymbol{\theta}^*, Y)\right], \tag{20}$$

where $\mathcal{H}_t := \sigma\left(\boldsymbol{X}(1), \ldots, \boldsymbol{X}(t), A(1), \ldots, A(t-1)\right)$ for all $t \in \mathbb{N}$. We start with deriving upper bounds on the hitting times of state $0^d$ using the ergodicity conditions of Assumptions 3 and 4. Previous works [25, 27, 29] have already established bounds on hitting times in geometrically and polynomially ergodic chains in terms of their corresponding Lyapunov function. However, our objective is to provide a precise characterization of all constants included in these bounds in terms of the constants of the drift equations 3 and 4. This characterization allows us to derive uniform bounds across the model class. In Appendix C.1, using the polynomial Lyapunov function provided in Assumption 4, we establish upper bounds on the $i$-th moment of hitting time of state $0^d$ from any state $\boldsymbol{x} \in \mathcal{X}$ and for $1 \leq i \leq r + 1$. Importantly, the derived bound is polynomial in terms of any component of the state $x_i$. Additionally, in Appendix C.2, we characterize the tail probabilities of the return time to state $0^d$ starting from $0^d$ in terms of the geometric Lyapunov function of Assumption 3. The derived tail bounds will be used in Lemma 6 to derive upper bounds for all moments of hitting times in the model class. These bounds, along with the skip-free behavior of the model, allow us to study the maximum state (with respect to $\ell_\infty$-norm) achieved up to time $T$ in MDP $(\mathcal{X}, \mathcal{A}, c, P_{\boldsymbol{\theta}^*})$ following Algorithm 1 as follows.

**Lemma 6.** *For $p \in \mathbb{N}$, the p-th moment of $\max_{1 \leq i \leq T} \tau_{0^d}^{(i)}$ and $M_{\boldsymbol{\theta}^*}^T$, that is the maximum $\ell_\infty$-norm of the state vector achieved up until and including time $T$ is $O(\log^p T)$.*

In the proof of Lemma 6 given in Appendix B.1, we make use of geometric ergodicity of the chain and the fact that hitting times have geometric tails to find an upper bound for moments of $M_{\boldsymbol{\theta}^*}^T$. Using this, we aim to bound the number of episodes started before or at $T$, denoted by $K_T$. We first find an upper bound for the number of episodes in which the second stopping criterion is met or there exists a state-action pair for which $N_t(\boldsymbol{x}, a)$ has increased more than twice. In the following lemma, we bound the number of such episodes, which we denote by $K_M$, in terms of random variable $M_{\boldsymbol{\theta}^*}^T$ and other problem-dependent constants. Proof of Lemma 7 is given in Appendix B.2.

**Lemma 7.** *The number of episodes triggered by the second stopping criterion and started before or at time $T$, denoted by $K_M$, satisfies $K_M \leq 2|\mathcal{A}|(M_{\boldsymbol{\theta}^*}^T + 1)^d \log_2 T$ a.s.*

We next bound the total number of episodes $K_T$ by bounding the number of episodes triggered by the first stopping criterion, using the fact that in such episodes, $\tilde{T}_k = \tilde{T}_{k-1} + 1$. Moreover, to address the settling time of each episode $k$, shown by $E_k = T_k - \tilde{T}_k$, we use the geometric ergodicity property and Lemma 6. Finally, the proof of Lemma 8 is given in Appendix B.3.

**Lemma 8.** *The number of episodes started by $T$ satisfies $K_T \leq 2\sqrt{|\mathcal{A}|(M_{\boldsymbol{\theta}^*}^T + 1)^d T \log_2 T}$ a.s.*

From Lemma 8, the upper bound given in Lemma 6 for moments of $M_{\boldsymbol{\theta}^*}^T$, and Cauchy–Schwarz inequality, it follows that the expected value of the number of episodes $K_T$ is of the order $\tilde{O}(\sqrt{h^d |\mathcal{A}| T})$. This term has a crucial role in determining the overall order of the total regret up to time $T$. In the rest of this section, we present a detailed proof of the lemmas and other results used to prove Theorem 1.

**Remark 3.** *The skip-free to the right property in Assumption 2 yields a polynomially-sized subset of the underlying state-space that can be explored as a function of $T$. This polynomially-sized subset can be viewed as the effective finite-size of the system in the worst-case, and then, directly applying finite-state problem bounds [49] would result in a regret of order $\tilde{O}(T^{d+0.5})$; since $d \geq 1$, such a coarse bound is not helpful even for asserting asymptotic optimality! However, to achieve a regret of $\tilde{O}(\sqrt{T})$, it is essential to carefully understand and characterize the distribution of $M_{\boldsymbol{\theta}^*}^T$ and then its moments, as demonstrated in Lemma 6.*

**Remark 4.** *The derived regret bound can be extended to a larger class of MDPs which consist of transient states in addition to the single irreducible class. Specifically, for any $\theta_1, \theta_2 \in \Theta$, the Markov process with transition kernel $P_{\theta_1}^{\pi_{\theta_2}^*}$ obtained from the MDP $(\mathcal{X}, \mathcal{A}, c, P_{\theta_1})$ by following policy $\pi_{\theta_2}^*$ has a single irreducible class $I_{\theta_1, \theta_2}$ and a set of transient states $T_{\theta_1, \theta_2}$. Furthermore, Assumptions 3 and 4 hold for the single irreducible class. The reasoning behind the proof remains true in this case using the following argument: each episode $k$ starts at $0^d$ which is in the irreducible set for the chosen policy $\pi_{\theta_k}^*$, hence, throughout the episode the algorithm remains in the irreducible set that is positive recurrent and never visits any transient states. In other words, episodes starting and ending at $0^d$ with a fixed episode dependent policy implies that reachable set of $0^d$ is all that can be explored, which is positive recurrent by our assumptions. As a result, we can restrict our proof derivations to the subset that is reachable from $0^d$ in each episode and follow the same analysis. The Lyapunov function based bounds apply to the positive recurrent states, and hence, restricting attention to states reachable from $0^d$ within each episode, we can use these bounds for our assessment of regret using norms of the state. Thereafter, the coarse bounds on the norms of the state can be applied as carried out in our proof.*

**Remark 5.** *By problem-dependent parameters, we refer to the parameters that characterize the complexity or size of the model class $\Theta$. These parameters are not just a function of the size of the state-space and diameter of the MDP (as mentioned in the literature on finite-size problems[5, 23, 49]), as stability needs to be accounted for in the countable state-space setting. The dependence is, thus, more complex and requires the inclusion of stability parameters, such as Lyapunov functions, petite sets, and ergodicity coefficients that are discussed in Assumptions 1-4.*

**Remark 6.** *In the subsequent sections, several equalities and inequalities in the proofs are between random variables and hold almost surely (a.s.). Throughout the remainder, we will omit the explicit mention of a.s., but any such statement should be interpreted in this context.*

### B.1   Proof of Lemma 6

*Proof.* Let $\{\alpha_i\}_{i \geq 0}$ be the sequence of hitting times of state $0^d$ starting from $0^d$ (set $\alpha_0 = 0$). Define $\tau_{0^d}^{(i)}$ as the length of the $i$-th recurrence time of state $0^d$ for $i \in \mathbb{N}$, i.e., $\tau_{0^d}^{(i)} = \alpha_i - \alpha_{i-1}$. For simplicity, we take $\tau_{0^d} = \tau_{0^d}^{(1)}$. Each such recurrence time is generated using policy $\pi_{\theta_i}^*$ that is determined using the algorithm in operation in an MDP governed by parameter $\boldsymbol{\theta}^*$. Furthermore, $\{\tau_{0^d}^{(i)}\}_{i \in \mathbb{N}}$ are independent with length at least 1, but they need not be identically distributed. The time $T$ can be in the middle of one of these recurrence times, hence the current recurrence interval count is $N(T) = \inf\{n : \sum_{i=1}^n \tau_{0^d}^{(i)} \geq T\}$. Note that the lower bound of 1 on every $\tau_{0^d}^{(i)}$ says that $N(T) \leq T$ a.s. Further, from the skip-free to the right property, the most any component of state can increase in during recurrence time $\tau_{0^d}^{(i)}$ is $h\tau_{0^d}^{(i)}$. Hence, the most any component of the state (and also the $\|\cdot\|_\infty$ norm of the state) can increase is given by $h \max_{i=1,\ldots,T} \tau_{0^d}^{(i)}$ where the random variables

are independent with geometrically decaying tails with a worst case rate of

$$\sup_{\theta_1,\theta_2\in\Theta} \tilde{\gamma}^g_{\theta_1,\theta_2} = 1 - \left( \sup_{\theta_1,\theta_2\in\Theta} \tilde{b}^g_{\theta_1,\theta_2} \right)^{-1};$$

see Lemma 11. From Lemma 10, we have

$$\tilde{b}^g_{\theta_1,\theta_2} = \frac{3b^g_{\theta_1,\theta_2}+1}{1-\gamma^g_{\theta_1,\theta_2}} \left( |C^g_{\theta_1,\theta_2}|^2 \max\left(1, \max_{\boldsymbol{u}\in C^g_{\theta_1,\theta_2}\setminus\{0^d\}} \mathbb{E}^{\pi^*_{\theta_2}}_{\boldsymbol{u}}[\tau_{0^d}]\right)\right)$$

$$\leq \frac{3b^g_*+1}{1-\gamma^g_*} \left( |C^g_*|^2 \max\left(1, \sup_{\substack{\boldsymbol{u}\in C^g_*\setminus\{0^d\}\\ \theta_1,\theta_2\in\Theta}} \phi^p_{\theta_1,\theta_2}(1)\left(V^p_{\theta_1,\theta_2}(\boldsymbol{u}) + b^p_{\theta_1,\theta_2}\alpha_{C^p_{\theta_1,\theta_2}}\right)\right)\right)$$

$$\leq \frac{3b^g_*+1}{1-\gamma^g_*} \left( |C^g_*|^2 \max\left(1, \sup_{\substack{\boldsymbol{u}\in C^g_*\setminus\{0^d\}\\ \theta_1,\theta_2\in\Theta}} \frac{1}{\beta^p_{\theta_1,\theta_2}}\left(s^p_{\theta_1,\theta_2}\|\boldsymbol{u}\|_\infty^{r^p_{\theta_1,\theta_2}} + \frac{b^p_{\theta_1,\theta_2}}{\min_{\boldsymbol{y}\in C^p_{\theta_1,\theta_2}} K_{\theta_1,\theta_2}(\boldsymbol{y})}\right)\right)\right)$$

$$\leq \frac{3b^g_*+1}{1-\gamma^g_*} \left( |C^g_*|^2 \max\left(1, \sup_{\boldsymbol{u}\in C^g_*\setminus\{0^d\}} \frac{1}{\beta^p_*}\left(s^p_*\|\boldsymbol{u}\|_\infty^{r^p_*} + \frac{b^p_*}{K_*}\right)\right)\right) \tag{21}$$

$$:= \tilde{b}^g_*,$$

and we define $\tilde{\gamma}^g_* := 1 - (\tilde{b}^g_*)^{-1}$. From the definition of $b^g_{\theta_1,\theta_2}$ in Assumption 3, $b^g_{\theta_1,\theta_2}$ is greater than or equal to 2. Thus, $\tilde{b}^g_{\theta_1,\theta_2} \geq 7$ and we have

$$\sup_{\theta_1,\theta_2\in\Theta} c^g_{\theta_1,\theta_2} = \sup_{\theta_1,\theta_2\in\Theta} \frac{b^g_{\theta_1,\theta_2}\left(\tilde{b}^g_{\theta_1,\theta_2}\right)^2}{\tilde{b}^g_{\theta_1,\theta_2}-1} \leq \frac{b^g_*\left(\tilde{b}^g_*\right)^2}{6} := c^g_*,$$

and as a result of Lemma 11,

$$\mathbb{P}_{0^d}(\tau^{(i)}_{0^d} > n) \leq c^g_* \left(\gamma^g_*\right)^n, \qquad 1 \leq i \leq T. \tag{22}$$

We upper bound $\mathbb{E}\left[M^T_{\boldsymbol{\theta}^*}\right]$ using the independence of $\{\tau^{(i)}_{0^d}\}_{i\in\mathbb{N}}$ and the above equation,

$$\mathbb{E}\left[M^T_{\boldsymbol{\theta}^*}\right] \leq h\,\mathbb{E}[\max_{1\leq i\leq T}\tau^{(i)}_{0^d}] = h\sum_{n=0}^\infty \mathbb{P}(\max_{1\leq i\leq T}\tau^{(i)}_{0^d} > n)$$

$$= h\sum_{n=0}^\infty \left(1 - \mathbb{P}(\max_{1\leq i\leq T}\tau^{(i)}_{0^d}\leq n)\right) = h\sum_{n=0}^\infty \left(1 - \prod_{i=1}^T \mathbb{P}\left(\tau^{(i)}_{0^d}\leq n\right)\right)$$

$$\leq hn_0 + h\sum_{n=n_0}^\infty 1 - \left(1 - c^g_*(\gamma^g_*)^{n_0}(\gamma^g_*)^{n-n_0}\right)^T$$

$$\leq h(n_0+1) + h\sum_{n=n_0+1}^\infty 1 - \left(1 - (\gamma^g_*)^{n-n_0}\right)^T,$$

where $n_0$ is the smallest $n \geq 0$ such that $c^g_*\left(\gamma^g_*\right)^n < 1$. By Reimann sum approximation, we get

$$\mathbb{E}\left[M^T_{\boldsymbol{\theta}^*}\right] \leq h(n_0+1) + h\sum_{n=1}^\infty 1 - (1-(\gamma^g_*)^n)^T$$

$$< h(n_0+1) + h\int_0^\infty 1 - (1-(\gamma^g_*)^u)^T \, du$$

$$= h(n_0+1) + \frac{h}{\log\gamma^g_*}\int_0^1 \frac{1-u^T}{1-u} \, du$$

$$\leq h(n_0+1) + \frac{h}{\log\gamma^g_*}(\log T + 1),$$

where the last inequality follows from $\sum_{n=1}^T n^{-1} \le \log T + 1$ and thus $\mathbb{E}\left[M_{\boldsymbol{\theta}^*}^T\right]$ is $O(h \log T)$. We now extend the result to moments of order greater than one. From (22), for $1 \le i \le T$,

$$\mathbb{P}_{0^d}(\tau_{0^d}^{(i)} > n) \le c_*^g (\gamma_*^g)^n = c_*^g (\gamma_*^g)^{n_0} (\gamma_*^g)^{n-n_0} < (\gamma_*^g)^{n-n_0}.$$

For $n \ge n_0$, let $t = n - n_0 \ge 0$ and $Y_i = \max(\tau_{0^d}^{(i)} - n_0, 0)$ to get

$$\mathbb{P}_{0^d}(Y_i > t) = \mathbb{P}_{0^d}(\tau_{0^d}^{(i)} - n_0 > t) < (\gamma_*^g)^t,$$

which means random variables $\{Y_i\}_{i=1}^T$ are stochastically dominated by independent and identically distributed geometric random variables with parameter $1 - \gamma_*^g$. Furthermore, [56] argues that the $p$-th moment of the maximum of $T$ independent and identically distributed geometric random variables is $O(\log^p T)$. Thus, the $p$-th moment of $\max_{1 \le i \le T} Y_i$ is $O(\log^p T)$ and

$$\max_{1 \le i \le T} Y_i = \max(\tau_{0^d}^{(1)} - n_0, \dots, \tau_{0^d}^{(T)} - n_0, 0) = \max(\tau_{0^d}^{(1)}, \dots, \tau_{0^d}^{(T)}, n_0) - n_0$$

$$\ge \max(\tau_{0^d}^{(1)}, \dots, \tau_{0^d}^{(T)}) - n_0 \ge h^{-1} M_{\boldsymbol{\theta}^*}^T - n_0,$$

which gives

$$\mathbb{E}\left[\left(M_{\boldsymbol{\theta}^*}^T\right)^p\right] \le h^p \, \mathbb{E}\left[\left(\max_{1 \le i \le T} \tau_{0^d}^{(i)}\right)^p\right] \le h^p \, \mathbb{E}\left[\left(\max_{1 \le i \le T} Y_i + n_0\right)^p\right].$$

Since the right-hand side of the above equation is $O(h^p \log^p T)$, the claim is proved. $\quad\square$

## B.2 Proof of Lemma 7

*Proof.* Let $K_M(\boldsymbol{x}, a)$ be the number of episodes $k$ such that $1 \le k \le K_T$ and in which the number of visits to the state-action pair $(\boldsymbol{x}, a)$ is increased more than twice at episode $k$, or

$$K_M(\boldsymbol{x}, a) = |\{k \le K_T : N_{\tilde{t}_{k+1}}(\boldsymbol{x}, a) > 2N_{t_k}(\boldsymbol{x}, a)\}|.$$

As for every episode in the above set the number of visits to $(x, a)$ doubles,

$$K_M(\boldsymbol{x}, a) \le \log_2(N_{T+1}(\boldsymbol{x}, a)) + 1,$$

and we can upper bound $K_M$ as follows

$$K_M = \sum_{\boldsymbol{x} \in \mathcal{X}, a \in \mathcal{A}} K_M(\boldsymbol{x}, a) = \sum_{\substack{\|\boldsymbol{x}\|_\infty \le M_{\boldsymbol{\theta}^*}^T \\ a \in \mathcal{A}}} K_M(\boldsymbol{x}, a)$$

$$\le \sum_{\substack{\|\boldsymbol{x}\|_\infty \le M_{\boldsymbol{\theta}^*}^T \\ a \in \mathcal{A}}} (1 + \log_2 N_{T+1}(\boldsymbol{x}, a)) \le |\mathcal{A}| \left(M_{\boldsymbol{\theta}^*}^T + 1\right)^d (1 + \log_2 T).$$

This completes the proof. $\quad\square$

## B.3 Proof of Lemma 8

*Proof.* We define macro episodes with start times $t_{n_k}$, $k = 1, 2, \dots, K_M + 1$ where $t_{n_1} = t_1$, $t_{n_{K_M+1}} = T + 1$ (which is equivalent to $n_{K_M+1} = K_T + 1$), and for $1 < k < K_M + 1$

$$t_{n_{k+1}} = \min\{t_j > t_{n_k} : \quad N_{t_j}(\boldsymbol{x}, a) > 2N_{t_{j-1}}(\boldsymbol{x}, a) \text{ for some } (\boldsymbol{x}, a)\},$$

which are episodes wherein the second stopping criterion is triggered. Any episode (except for the last episode) in a macro episode must be triggered by the first stopping criterion; equivalently, $\tilde{T}_j = \tilde{T}_{j-1} + 1$ for all $j = n_k, n_k + 1, \dots, n_{k+1} - 2$. For $1 \le k \le K_M$, let $T_k^M = \sum_{j=n_k}^{n_{k+1}-1} T_j$ be the length of the $k$-th macro episode. We have

$$T_k^M = \sum_{j=n_k}^{n_{k+1}-1} T_j \ge \sum_{j=n_k}^{n_{k+1}-1} \tilde{T}_j \ge 1 + \sum_{j=n_k}^{n_{k+1}-2} (j - n_k + 2) = 0.5(n_{k+1} - n_k)(n_{k+1} - n_k + 1).$$

Consequently, $n_{k+1} - n_k \leq \sqrt{2T_k^M}$ for all $1 \leq k \leq K_M$. From this, we obtain

$$K_T = n_{K_M+1} - 1 = \sum_{k=1}^{K_M}(n_{k+1} - n_k) \leq \sum_{k=1}^{K_M}\sqrt{2T_k^M}.$$

Using the above equation and the fact that $\sum_{k=1}^{K_M}T_k^M = T$ we get

$$K_T \leq \sum_{k=1}^{K_M}\sqrt{2T_k^M} \leq \sqrt{K_M\sum_{k=1}^{K_M}2T_k^M} = \sqrt{2K_MT}.$$

Finally, from Lemma 7 we get

$$K_T \leq \sqrt{2K_MT} \leq 2\sqrt{|\mathcal{A}|\left(M_{\boldsymbol{\theta}^*}^T + 1\right)^d T\log_2 T}.$$

This completes the proof. $\qquad\square$

## B.4 Proof of Lemma 1

*Proof.* Let $E_k = T_k - \tilde{T}_k \geq 0$ be the settling time needed to return to state $0^d$ after a stopping criterion is realized in episode $k$. We have

$$R_0 = \mathbb{E}\left[\sum_{k=1}^{K_T}T_kJ(\theta_k)\right] - T\,\mathbb{E}\left[J(\boldsymbol{\theta}^*)\right]$$

$$= \mathbb{E}\left[\sum_{k=1}^{K_T}\tilde{T}_kJ(\theta_k)\right] + \mathbb{E}\left[\sum_{k=1}^{K_T}E_kJ(\theta_k)\right] - T\,\mathbb{E}\left[J(\boldsymbol{\theta}^*)\right]. \tag{23}$$

We first simplify the first term in the above summation. From the monotone convergence theorem,

$$\mathbb{E}\left[\sum_{k=1}^{K_T}\tilde{T}_kJ(\theta_k)\right] = \sum_{k=1}^{\infty}\mathbb{E}\left[\mathbb{I}_{\{t_k \leq T\}}\tilde{T}_kJ(\theta_k)\right].$$

Note that the first stopping criterion of Algorithm 1 ensures that $\tilde{T}_k \leq \tilde{T}_{k-1} + 1$ at all episodes $k \geq 1$. Hence

$$\mathbb{E}\left[\mathbb{I}_{\{t_k \leq T\}}\tilde{T}_kJ(\theta_k)\right] \leq \mathbb{E}\left[\mathbb{I}_{\{t_k \leq T\}}(\tilde{T}_{k-1} + 1)J(\theta_k)\right].$$

Since $\mathbb{I}_{\{t_k \leq T\}}(\tilde{T}_{k-1} + 1)$ is measurable with respect to $\mathcal{H}_{t_k}$, by (20) we get

$$\mathbb{E}\left[\mathbb{I}_{\{t_k \leq T\}}(\tilde{T}_{k-1} + 1)J(\theta_k)\right] = \mathbb{E}\left[\mathbb{I}_{\{t_k \leq T\}}(\tilde{T}_{k-1} + 1)J(\boldsymbol{\theta}^*)\right].$$

Therefore,

$$\mathbb{E}\left[\sum_{k=1}^{K_T}\tilde{T}_kJ(\theta_k)\right] \leq \sum_{k=1}^{\infty}\mathbb{E}\left[\mathbb{I}_{\{t_k \leq T\}}(\tilde{T}_{k-1} + 1)J(\boldsymbol{\theta}^*)\right] = \mathbb{E}\left[\sum_{k=1}^{K_T}(\tilde{T}_{k-1} + 1)J(\boldsymbol{\theta}^*)\right].$$

Thus,

$$\mathbb{E}\left[\sum_{k=1}^{K_T}\tilde{T}_kJ(\theta_k)\right] - T\,\mathbb{E}\left[J(\boldsymbol{\theta}^*)\right] \leq \mathbb{E}\left[J(\boldsymbol{\theta}^*)\sum_{k=1}^{K_T}(\tilde{T}_{k-1} + 1)\right] - \mathbb{E}\left[J(\boldsymbol{\theta}^*)\sum_{k=1}^{K_T}T_k\right]$$

$$= \mathbb{E}\left[J(\boldsymbol{\theta}^*)\left(K_T + 1 - T_{K_T} - \sum_{k=1}^{K_T-1}E_k\right)\right]$$

$$\leq \mathbb{E}\left[J(\boldsymbol{\theta}^*)K_T\right]. \tag{24}$$

For the second term in (23), from Assumption 5

$$\mathbb{E}\Big[\sum_{k=1}^{K_T} E_k J(\theta_k)\Big] \le J^* \mathbb{E}\Big[\sum_{k=1}^{K_T} E_k\Big] \le J^* \mathbb{E}[K_T \max_{1\le i\le T} \tau_{0^d}^{(i)}]. \tag{25}$$

Substitutinh (24) and (25) in (23), we get

$$R_0 \le \mathbb{E}\left[K_T J(\boldsymbol{\theta}^*)\right] + J^* \mathbb{E}[K_T \max_{1\le i\le T} \tau_{0^d}^{(i)}]$$

$$\le J^* \mathbb{E}\left[K_T\right] + J^* \mathbb{E}[K_T \max_{1\le i\le T} \tau_{0^d}^{(i)}]$$

$$= J^* \mathbb{E}\left[K_T\Big(\max_{1\le i\le T} \tau_{0^d}^{(i)} + 1\Big)\right].$$

$\square$

### B.5   Proof of Lemma 2

*Proof.* We note that the state of the MDP is equal to $0^d$ at the beginning of all episodes and the relative value function $v(\boldsymbol{x};\theta)$ is equal to 0 at $\boldsymbol{x} = 0^d$ for all $\theta$. Thus,

$$R_1 = \mathbb{E}\Big[\sum_{k=1}^{K_T}\sum_{t=t_k}^{t_{k+1}-1}\Big[v\left(\boldsymbol{X}\left(t\right);\theta_k\right) - v\left(\boldsymbol{X}\left(t+1\right);\theta_k\right)\Big]\Big]$$

$$= \mathbb{E}\Big[\sum_{k=1}^{K_T}\Big[v\left(\boldsymbol{X}\left(t_k\right);\theta_k\right) - v\left(\boldsymbol{X}\left(t_{k+1}\right);\theta_k\right)\Big]\Big]$$

$$= \mathbb{E}\Big[\sum_{k=1}^{K_T-1}\Big[v\left(0^d;\theta_k\right) - v\left(0^d;\theta_k\right)\Big] + v\left(0^d;\theta_{K_T}\right) - v\left(\boldsymbol{X}(T+1);\theta_{K_T}\right)\Big]$$

$$= -\mathbb{E}[v\left(\boldsymbol{X}(T+1);\theta_{K_T}\right)].$$

From the lower bound derived for the relative value function in (14),

$$-v(\boldsymbol{x};\theta) \le J^* \mathbb{E}_{\boldsymbol{x}}^{\pi_\theta^*}[\tau_{0^d}] \le \frac{J^*}{\beta_*^p}\Big(s_*^p \|\boldsymbol{x}\|_\infty^{r_*^p} + \frac{b_*^p}{K_*}\Big),$$

where the second inequality follows from (21) in the proof of Lemma 6. We also note that $\|\boldsymbol{X}(T+1)\|_\infty \le M_{\boldsymbol{\theta}^*}^T + h$. Thus,

$$R_1 = -\mathbb{E}[v\left(\boldsymbol{X}(T+1);\theta_{K_T}\right)] \le \mathbb{E}\Big[\frac{J^*}{\beta_*^p}\Big(s_*^p(M_{\boldsymbol{\theta}^*}^T + h)^{r_*^p} + \frac{b_*^p}{K_*}\Big)\Big].$$

From the inequality $(a+b)^r \le 2^r(a^r + b^r)$, we have

$$R_1 \le \frac{J^* 2^{r_*^p} s_*^p}{\beta_*^p}\mathbb{E}\Big[\left(M_{\boldsymbol{\theta}^*}^T\right)^{r_*^p}\Big] + \frac{J^*}{\beta_*^p}\Big(s_*^p(2h)^{r_*^p} + \frac{b_*^p}{K_*}\Big).$$

$\square$

### B.6   Proof of Lemma 3

*Proof.* Let $\boldsymbol{Z}\left(t\right) = \big(\boldsymbol{X}\left(t\right), \pi_{\theta_k}^*\left(\boldsymbol{X}\left(t\right)\right)\big)$ be the state-action pair at $t_k \le t < t_{k+1}$. $R_2$ can be upper bounded as

$$R_2 = \mathbb{E}\Big[\sum_{k=1}^{K_T}\sum_{t=t_k}^{t_{k+1}-1}\Big[v\left(\boldsymbol{X}\left(t+1\right);\theta_k\right) - \sum_{\boldsymbol{y}\in\mathcal{X}} P_{\theta_k}\Big(\boldsymbol{y}\,\Big|\,\boldsymbol{X}\left(t\right), \pi_{\theta_k}^*\left(\boldsymbol{X}\left(t\right)\right)\Big) v\left(\boldsymbol{y};\theta_k\right)\Big]\Big]$$

$$\le \mathbb{E}\Big[\sum_{k=1}^{K_T}\sum_{t=t_k}^{t_{k+1}-1}\Big[\sum_{\boldsymbol{y}\in\mathcal{X}} |P_{\boldsymbol{\theta}^*}(\boldsymbol{y}|\boldsymbol{Z}\left(t\right)) - P_{\theta_k}(\boldsymbol{y}|\boldsymbol{Z}\left(t\right))|\,|v(\boldsymbol{y};\theta_k)|\Big]\Big]$$

$$\le \sum_{t=1}^{T}\mathbb{E}\Big[\Big(\max_{\substack{1\le k\le K_T \\ \|\boldsymbol{x}\|_\infty\le M_{\boldsymbol{\theta}^*}^T}} |v(\boldsymbol{x};\theta_k)|\Big)\|P_{\boldsymbol{\theta}^*}(\cdot|\boldsymbol{Z}\left(t\right)) - P_{\theta_k}(\cdot|\boldsymbol{Z}\left(t\right))\|_1\Big]. \tag{26}$$

We have
$$\|P_{\boldsymbol{\theta}^*}(\cdot|\boldsymbol{Z}(t)) - P_{\theta_k}(\cdot|\boldsymbol{Z}(t))\|_1 \le \|P_{\boldsymbol{\theta}^*}(\cdot|\boldsymbol{Z}(t)) - P_{\hat{\theta}_k}(\cdot|\boldsymbol{Z}(t))\|_1 + \|P_{\theta_k}(\cdot|\boldsymbol{Z}(t)) - P_{\hat{\theta}_k}(\cdot|\boldsymbol{Z}(t))\|_1,$$
where $P_{\hat{\theta}_k}(\boldsymbol{y}|\boldsymbol{Z}(t))$ is the empirical transition probability defined as
$$P_{\hat{\theta}_k}(\boldsymbol{y}|\boldsymbol{Z}(t)) = \frac{N_{t_k}(\boldsymbol{Z}(t),\boldsymbol{y})}{\max(1, N_{t_k}(\boldsymbol{Z}(t)))},$$
and for any tuple $(\boldsymbol{x}, a, \boldsymbol{y})$, we define $N_1(\boldsymbol{x}, a, \boldsymbol{y}) = 0$ and for $t > 1$,
$$N_t(\boldsymbol{x}, a, \boldsymbol{y}) = |\{t_k \le i < \tilde{t}_{k+1} \le t \text{ for some } k \ge 1 : (\boldsymbol{X}(i), A(i), \boldsymbol{X}(i+1)) = (\boldsymbol{x}, a, \boldsymbol{y})\}|.$$
Thus, from (26) and defining random variable $v_M = \max_{\substack{1 \le k \le K_T \\ \|\boldsymbol{x}\|_\infty \le M_{\boldsymbol{\theta}^*}^T}} |v(\boldsymbol{x}; \theta_k)|$,

$$R_2 \le \sum_{t=1}^T \mathbb{E}\left[v_M \|P_{\boldsymbol{\theta}^*}(\cdot|\boldsymbol{Z}(t)) - P_{\hat{\theta}_k}(\cdot|\boldsymbol{Z}(t))\|_1\right] + \sum_{t=1}^T \mathbb{E}\left[v_M \|P_{\theta_k}(\cdot|\boldsymbol{Z}(t)) - P_{\hat{\theta}_k}(\cdot|\boldsymbol{Z}(t))\|_1\right]. \tag{27}$$

We define set $B_k$ as the set of parameters $\theta$ for which the transition kernel $P_\theta(\cdot|\boldsymbol{z})$ is close to the empirical transition kernel $P_{\hat{\theta}_k}(\cdot|\boldsymbol{z})$ at episode $k$ for every state-action pair $\boldsymbol{z} = (\boldsymbol{x}, a) \in \mathcal{X} \times \mathcal{A}$, or

$$B_k = \left\{\theta : \|P_\theta(\cdot|\boldsymbol{z}) - P_{\hat{\theta}_k}(\cdot|\boldsymbol{z})\|_1 \le \beta_k(\boldsymbol{z}), \ \boldsymbol{z} = (\boldsymbol{x}, a) \in \{0, 1, \cdots, hT\}^d \times \mathcal{A}\right\},$$

where $\beta_k(\boldsymbol{z}) = \sqrt{\frac{14 \prod_{i=1}^d (x_i + h)}{\max(1, N_{t_k}(\boldsymbol{z}))} \log\left(\frac{2|\mathcal{A}|T}{\tilde{\delta}}\right)}$ for $\boldsymbol{x} = (x_1, \ldots, x_d)$ and some $0 < \tilde{\delta} < 1$, which will be determined later. We simplify the $\ell_1$-difference of the real and empirical transition kernels as follows

$$\|P_{\boldsymbol{\theta}^*}(\cdot|\boldsymbol{Z}(t)) - P_{\hat{\theta}_k}(\cdot|\boldsymbol{Z}(t))\|_1$$
$$= \mathbb{I}_{\{\boldsymbol{\theta}^* \notin B_k\}}\|P_{\theta_*}(\cdot|\boldsymbol{Z}(t)) - P_{\hat{\theta}_k}(\cdot|\boldsymbol{Z}(t))\|_1 + \mathbb{I}_{\{\boldsymbol{\theta}^* \in B_k\}}\|P_{\theta_*}(\cdot|\boldsymbol{Z}(t)) - P_{\hat{\theta}_k}(\cdot|\boldsymbol{Z}(t))\|_1$$
$$\le 2\mathbb{I}_{\{\boldsymbol{\theta}^* \notin B_k\}} + \beta_k(\boldsymbol{Z}(t)).$$

Similarly, we have
$$\|P_{\theta_k}(\cdot|\boldsymbol{Z}(t)) - P_{\hat{\theta}_k}(\cdot|\boldsymbol{Z}(t))\|_1 \le 2\mathbb{I}_{\{\theta_k \notin B_k\}} + \beta_k(\boldsymbol{Z}(t)).$$

Substituting in (27), we get

$$R_2 \le \mathbb{E}\left[\sum_{k=1}^{K_T}\sum_{t=t_k}^{t_{k+1}-1} 2v_M\left[\mathbb{I}_{\{\boldsymbol{\theta}^* \notin B_k\}} + \mathbb{I}_{\{\theta_k \notin B_k\}}\right]\right] + \mathbb{E}\left[\sum_{k=1}^{K_T}\sum_{t=t_k}^{t_{k+1}-1} 2v_M \beta_k(\boldsymbol{Z}(t))\right]. \tag{28}$$

We first find an upper bound for $v_M = \max_{\substack{1 \le k \le K_T \\ \|\boldsymbol{x}\|_\infty \le M_{\boldsymbol{\theta}^*}^T}} |v(\boldsymbol{x}; \theta_k)|$ using the bounds derived in (13) and (14). From (13),

$$\begin{aligned}
v(\boldsymbol{x}; \theta_k) &\le \mathbb{E}_{\boldsymbol{x}}^{\pi_{\theta_k}^*}\left[Kd(\|\boldsymbol{x}\|_\infty + h\tau_{0^d})^r \tau_{0^d}\right]\\
&\le \mathbb{E}_{\boldsymbol{x}}^{\pi_{\theta_k}^*}\left[2^r Kd(\|\boldsymbol{x}\|_\infty^r + h^r(\tau_{0^d})^r)\tau_{0^d}\right]\\
&= Kd(2\|\boldsymbol{x}\|_\infty)^r \mathbb{E}_{\boldsymbol{x}}^{\pi_{\theta_k}^*}[\tau_{0^d}] + Kd(2h)^r \mathbb{E}_{\boldsymbol{x}}^{\pi_{\theta_k}^*}\left[(\tau_{0^d})^{r+1}\right]\\
&\le Kd2^r(\|\boldsymbol{x}\|_\infty^r + h^r)\mathbb{E}_{\boldsymbol{x}}^{\pi_{\theta_k}^*}\left[(\tau_{0^d})^{r+1}\right]\\
&\le Kd(r+1)2^r(\|\boldsymbol{x}\|_\infty^r + h^r)\phi_{\theta_k}^p(r+1)\left(V_{\theta_k}^p(\boldsymbol{x}) + b_{\theta_k}^p \alpha_{C_{\theta_k}^p}\right)\\
&\le Kd(r+1)2^r(\|\boldsymbol{x}\|_\infty^r + h^r)\phi_{\theta_k}^p(r+1)\left(s_*^p\|\boldsymbol{x}\|_\infty^{rp} + b_*^p(K_*)^{-1}\right), \tag{29}
\end{aligned}$$

where the second line follows from the inequality $(a+b)^r \le 2^r(a^r + b^r)$, the fifth line from Lemma 10, and the last line from Assumption 4 and (21). We further have

$$\begin{aligned}
\phi_{\theta_1, \theta_2}^p(r+1) &= \prod_{j=1}^{r+1} \frac{1}{\beta_{\theta_1, \theta_2}^{\eta_j}}\left(2^{j-1} + (j-1)\alpha_{C_{\theta_1, \theta_2}^p} b_{\theta_1, \theta_2}^{\eta_j}\right)\\
&\le \prod_{j=1}^{r+1} \frac{r+1}{\min(1, \beta_*^p)}\left(2^{j-1} + (j-1)(K_*)^{-1} b_{\theta_1, \theta_2}^{\eta_j}\right),
\end{aligned}$$

where using the definition of $b^{\eta_j}_{\theta_1,\theta_2}$ in (38),

$$b^{\eta_j}_{\theta_1,\theta_2} = \left(b^p_{\theta_1,\theta_2}\right)^{\eta_j} + \eta_j \tilde{\beta}^p_{\theta_1,\theta_2} \max\left(1, \left(\tilde{\beta}^p_{\theta_1,\theta_2}\right)^{(\alpha^p_{\theta_1,\theta_2}+\eta_j-1)/(1-\alpha^p_{\theta_1,\theta_2})}\right) \leq 1 + b^p_* + \beta^p_*.$$

We also define

$$\phi^p_*(r+1) := \prod_{j=1}^{r+1} \frac{r+1}{\min(1,\beta^p_*)} \left(2^{j-1} + (j-1)(K_*)^{-1}(1 + b^p_* + \beta^p_*)\right).$$

We next find a lower bound for $v(\boldsymbol{x};\theta_k)$ using (14) as follows:

$$v(\boldsymbol{x};\theta_k) \geq -J^* \mathbb{E}^{\pi^*_{\theta_k}}_{\boldsymbol{x}}[\tau_{0^d}] \geq -\frac{J^*}{\beta^p_*}\left(s^p_* \|\boldsymbol{x}\|^{r^p_*}_\infty + \frac{b^p_*}{K_*}\right).$$

Combining (29) and the above equation, we get a uniform upper bound for $|v(\boldsymbol{x};\theta_k)|$ over $\Theta$, which we use to upper bound $v_M = \max_{\substack{1 \leq k \leq K_T \\ \|\boldsymbol{x}\|_\infty \leq M^T_{\boldsymbol{\theta}*}}} |v(\boldsymbol{x};\theta_k)|$ as below

$$\begin{aligned}
v_M &\leq (J^* + Kd(r+1)2^r) \phi^p_*(r+1) \left(\left(M^T_{\boldsymbol{\theta}*}\right)^r + h^r\right)\left(s^p_* \left(M^T_{\boldsymbol{\theta}*}\right)^{r^p_*} + b^p_*(K_*)^{-1}\right) \\
&= c_{p_1}\left(\left(M^T_{\boldsymbol{\theta}*}\right)^r + h^r\right)\left(s^p_* \left(M^T_{\boldsymbol{\theta}*}\right)^{r^p_*} + b^p_*(K_*)^{-1}\right) \\
&\leq c_{p_2}\left(M^T_{\boldsymbol{\theta}*}\right)^{r+r^p_*},
\end{aligned} \tag{30}$$

where the constant terms are defined as

$$c_{p_1} := (J^* + Kd(r+1)2^r)\phi^p_*(r+1), \quad c_{p_2} := \max\left(1, c_{p_1}(h^r+1)(s^p_* + b^p_*(K_*)^{-1})\right).$$

A deterministic upper bound on $v_M$ can also be found from the above equation. Noting that from Assumption 2, until time $T$ only states with each component less than or equal to $hT$ are visited, we have

$$v_M \leq c_{p_2}\left(M^T_{\boldsymbol{\theta}*}\right)^{r+r^p_*} \leq c_{p_2}(Th)^{r+r^p_*} := Q(T),$$

where $Q(T)$ is a polynomial defined as above. Using the bounds derived for $v_M$, we bound $R_2$ starting with the first term on the right-hand side of (28). We have

$$\begin{aligned}
\mathbb{E}\left[\sum_{k=1}^{K_T}\sum_{t=t_k}^{t_{k+1}-1} 2v_M\left[\mathbb{I}_{\{\boldsymbol{\theta}^* \notin B_k\}} + \mathbb{I}_{\{\theta_k \notin B_k\}}\right]\right] &\leq 2Q(T)\mathbb{E}\left[\sum_{k=1}^{K_T}\sum_{t=t_k}^{t_{k+1}-1} \mathbb{I}_{\{\boldsymbol{\theta}^* \notin B_k\}} + \mathbb{I}_{\{\theta_k \notin B_k\}}\right] \\
&\leq 2TQ(T)\mathbb{E}\left[\sum_{k=1}^{K_T} \mathbb{I}_{\{\boldsymbol{\theta}^* \notin B_k\}} + \mathbb{I}_{\{\theta_k \notin B_k\}}\right] \\
&\leq 2TQ(T)\sum_{k=1}^{T}\mathbb{E}\left[\mathbb{I}_{\{\boldsymbol{\theta}^* \notin B_k\}} + \mathbb{I}_{\{\theta_k \notin B_k\}}\right] \\
&\leq 4TQ(T)\sum_{k=1}^{T}\mathbb{P}\{\boldsymbol{\theta}^* \notin B_k\},
\end{aligned} \tag{31}$$

where the last inequality follows from (20) and the fact that set $B_k$ is $\mathcal{H}_{t_k}$−measurable. To further simplify the first term in (28), we find an upper bound for $\mathbb{P}\{\boldsymbol{\theta}^* \notin B_k\}$ using [64]. For a fixed $\boldsymbol{z} = (\boldsymbol{x},a)$ and $n$ independent samples of the distribution $P_{\boldsymbol{\theta}^*}(.|\boldsymbol{z})$, the $L^1$-deviation of the true distribution $P_{\boldsymbol{\theta}^*}(.|\boldsymbol{z})$ and empirical distribution at the end of episode $k$, $P_{\hat{\theta}_k}(.|\boldsymbol{z})$, is bounded in [10] as

$$\mathbb{P}\left\{\|P_{\boldsymbol{\theta}^*}(\cdot|\boldsymbol{z}) - P_{\hat{\theta}_k}(\cdot|\boldsymbol{z})\|_1 \geq \sqrt{\frac{14\prod_{i=1}^d (x_i+h)}{n}\log\left(\frac{2|\mathcal{A}|T}{\tilde{\delta}}\right)}\right\} \leq \frac{\tilde{\delta}}{20|\mathcal{A}|T^7\prod_{i=1}^d(x_i+h)}.$$

Therefore,

$$\mathbb{P}\left\{\|P_{\boldsymbol{\theta}^*}(\cdot|\boldsymbol{z}) - P_{\hat{\theta}_k}(\cdot|\boldsymbol{z})\|_1 \geq \beta_k(\boldsymbol{z}) \,\Big|\, N_{t_k}(\boldsymbol{z}) = n\right\} \leq \frac{\tilde{\delta}}{20|\mathcal{A}|T^7\prod_{i=1}^d(x_i+h)},$$

and

$$\mathbb{P}\left\{\|P_{\boldsymbol{\theta}^*}(\cdot|\boldsymbol{z}) - P_{\hat{\theta}_k}(\cdot|\boldsymbol{z})\|_1 \geq \beta_k(\boldsymbol{z})\right\}$$

$$= \sum_{n=1}^{T} \mathbb{P}\left\{\|P_{\boldsymbol{\theta}^*}(\cdot|\boldsymbol{z}) - P_{\hat{\theta}_k}(\cdot|\boldsymbol{z})\|_1 \geq \beta_k(\boldsymbol{z}) \,\Big|\, N_{t_k}(\boldsymbol{z}) = n\right\} \mathbb{P}\left\{N_{t_k}(\boldsymbol{z}) = n\right\}$$

$$\leq \frac{\tilde{\delta}}{20|\mathcal{A}|T^6 \prod_{i=1}^{d}(x_i + h)}.$$

The probability that at episode $k \leq T$, the true parameter $\boldsymbol{\theta}^*$ does not belong to the confidence set $B_k$ can be bounded using the above and union bound as

$$\mathbb{P}\{\boldsymbol{\theta}^* \notin B_k\} \leq \sum_{\boldsymbol{z} \in \{0,1,\cdots,hT\}^d \times \mathcal{A}} \mathbb{P}\left\{\|P_{\boldsymbol{\theta}^*}(\cdot|\boldsymbol{z}) - P_{\hat{\theta}_k}(\cdot|\boldsymbol{z})\|_1 \geq \beta_k(\boldsymbol{z})\right\}$$

$$\leq \sum_{\boldsymbol{z} \in \{0,1,\cdots,hT\}^d \times \mathcal{A}} \frac{\tilde{\delta}}{20|\mathcal{A}|T^6 \prod_{i=1}^{d}(x_i + h)}$$

$$= \sum_{\boldsymbol{x} \in \{0,1,\cdots,hT\}^d} \frac{\tilde{\delta}}{20T^6 \prod_{i=1}^{d}(x_i + h)}$$

$$\leq \frac{\tilde{\delta}}{20T^6} \left(\log\left(h(T+1)\right) + 1\right)^d$$

$$\leq \frac{\tilde{\delta}}{20k^6} \left(\log\left(h(T+1)\right) + 1\right)^d.$$

In the summation in the above equation, we have simplified the expression by summing over $x_i \leq hT$ instead of considering the more detailed summation over $x_i \leq M_{\boldsymbol{\theta}^*}^T$. However, this simplification does not affect the final evaluation of regret, as this term is not dominant and only contributes to a logarithmic term in the regret bound. Substituting in (31),

$$\mathbb{E}\left[\sum_{k=1}^{K_T} \sum_{t=t_k}^{t_{k+1}-1} 2v_M \left[\mathbb{I}_{\{\boldsymbol{\theta}^* \notin B_k\}} + \mathbb{I}_{\{\theta_k \notin B_k\}}\right]\right] \leq 4TQ(T) \sum_{k=1}^{T} \mathbb{P}\{\boldsymbol{\theta}^* \notin B_k\}$$

$$\leq \frac{\tilde{\delta}\left(\log\left(h(T+1)\right) + 1\right)^d TQ(T)}{5} \sum_{k=1}^{\infty} \frac{1}{k^6}$$

$$< \tilde{\delta}\left(\log\left(h(T+1)\right) + 1\right)^d TQ(T). \qquad (32)$$

We now upper bound the second term in (28). From (30),

$$\mathbb{E}\left[\sum_{k=1}^{K_T} \sum_{t=t_k}^{t_{k+1}-1} 2v_M \beta_k\left(\boldsymbol{Z}(t)\right)\right] \leq 2c_{p_2} \mathbb{E}\left[\left(M_{\boldsymbol{\theta}^*}^T\right)^{r+r_*^p} \sum_{k=1}^{K_T} \sum_{t=t_k}^{t_{k+1}-1} \beta_k\left(\boldsymbol{Z}(t)\right)\right]. \qquad (33)$$

To bound the regret term resulting from the summation of $\beta_k\left(\boldsymbol{Z}(t)\right)$, we note that from the second stopping criterion, $N_t\left(\boldsymbol{Z}(t)\right) \leq 2N_{t_k}\left(\boldsymbol{Z}(t)\right)$ for all $t_k \leq t < t_{k+1}$ and

$$\sum_{k=1}^{K_T} \sum_{t=t_k}^{t_{k+1}-1} \beta_k\left(\boldsymbol{Z}(t)\right)$$

$$= \sum_{k=1}^{K_T} \sum_{t=t_k}^{t_{k+1}-1} \sqrt{\frac{14 \prod_{i=1}^{d}(\boldsymbol{X}_i(t) + h)}{\max(1, N_{t_k}(\boldsymbol{Z}(t)))} \log\left(\frac{2|\mathcal{A}|T}{\tilde{\delta}}\right)}$$

$$\leq \sqrt{14 \log\left(\frac{2|\mathcal{A}|T}{\tilde{\delta}}\right)} \left[\sum_{k=1}^{K_T} \sum_{t=t_k}^{\tilde{t}_{k+1}-1} \sqrt{\frac{2 \prod_{i=1}^{d}(\boldsymbol{X}_i(t) + h)}{\max(1, N_t(\boldsymbol{Z}(t)))}} + \sum_{k=1}^{K_T} \sum_{t=\tilde{t}_{k+1}}^{t_{k+1}-1} \sqrt{\prod_{i=1}^{d}(\boldsymbol{X}_i(t) + h)}\right].$$

$$(34)$$

The first summation can be simplified as

$$\sum_{k=1}^{K_T} \sum_{t=t_k}^{\tilde{t}_{k+1}-1} \sqrt{\frac{2\prod_{i=1}^d(\boldsymbol{X}_i(t)+h)}{\max(1,N_t(\boldsymbol{Z}(t)))}} \leq \sqrt{2(M_{\boldsymbol{\theta}^*}^T+h)^d} \sum_{k=1}^{K_T} \sum_{t=t_k}^{\tilde{t}_{k+1}-1} \frac{1}{\sqrt{\max(1,N_t(\boldsymbol{Z}(t)))}}$$

$$\leq 3\sqrt{2(M_{\boldsymbol{\theta}^*}^T+h)^d} \sum_{\boldsymbol{z}\in\{0,1,\cdots,M_{\boldsymbol{\theta}^*}^T\}^d\times\mathcal{A}} \sqrt{N_{T+1}(\boldsymbol{z})}$$

$$\leq 3\sqrt{2|\mathcal{A}|(M_{\boldsymbol{\theta}^*}^T+h)^d} \sqrt{\sum_{\boldsymbol{z}\in\{0,1,\cdots,M_{\boldsymbol{\theta}^*}^T\}^d\times\mathcal{A}} N_{T+1}(\boldsymbol{z})}$$

$$\leq 3\sqrt{2|\mathcal{A}|T}(M_{\boldsymbol{\theta}^*}^T+h)^d,$$

where the second inequality is due to the following arguments,

$$\sum_{k=1}^{K_T} \sum_{t=t_k}^{\tilde{t}_{k+1}-1} \frac{1}{\sqrt{\max(1,N_t(\boldsymbol{Z}(t)))}} = \sum_{\boldsymbol{z}\in\{0,1,\cdots,M_{\boldsymbol{\theta}^*}^T\}^d\times\mathcal{A}} \left(\mathbb{I}_{\{N_{T+1}(\boldsymbol{z})>0\}} + \sum_{i=1}^{N_{T+1}(\boldsymbol{z})-1} \frac{1}{\sqrt{i}}\right)$$

$$\leq 3 \sum_{\boldsymbol{z}\in\{0,1,\cdots,M_{\boldsymbol{\theta}^*}^T\}^d\times\mathcal{A}} \sqrt{N_{T+1}(\boldsymbol{z})}.$$

For the second term in (34), we get

$$\sum_{k=1}^{K_T} \sum_{t=\tilde{t}_{k+1}}^{t_{k+1}-1} \sqrt{\prod_{i=1}^d(\boldsymbol{X}_i(t)+h)} = \sqrt{(M_{\boldsymbol{\theta}^*}^T+h)^d} \sum_{k=1}^{K_T} E_k$$

$$\leq K_T \left(\max_{1\leq i\leq T} \tau_{0^d}^{(i)}\right) \sqrt{(M_{\boldsymbol{\theta}^*}^T+h)^d}$$

$$\leq 2\sqrt{|\mathcal{A}|T\log_2 T} \left(\max_{1\leq i\leq T} \tau_{0^d}^{(i)}\right) (M_{\boldsymbol{\theta}^*}^T+h)^d,$$

where $E_k = T_k - \tilde{T}_k$, and $K_T$ is bounded from Lemma 8. Thus $\sum_{k=1}^{K_T}\sum_{t=t_k}^{t_{k+1}-1} \beta_k(\boldsymbol{Z}(t))$ is bounded as

$$\sum_{k=1}^{K_T} \sum_{t=t_k}^{t_{k+1}-1} \beta_k(\boldsymbol{Z}(t)) \leq 24\sqrt{|\mathcal{A}|T\log_2 T\log\left(\frac{2|\mathcal{A}|T}{\tilde{\delta}}\right)} \left(\max_{1\leq i\leq T} \tau_{0^d}^{(i)}\right) (M_{\boldsymbol{\theta}^*}^T+h)^d.$$

Substituting the above bound in (33),

$$\mathbb{E}\left[\sum_{k=1}^{K_T} \sum_{t=t_k}^{t_{k+1}-1} 2v_M\beta_k(\boldsymbol{Z}(t))\right]$$

$$\leq 48c_{p_2}\sqrt{|\mathcal{A}|T\log_2 T\log\left(\frac{2|\mathcal{A}|T}{\tilde{\delta}}\right)} \mathbb{E}\left[(M_{\boldsymbol{\theta}^*}^T)^{r+r_*^p}(M_{\boldsymbol{\theta}^*}^T+h)^d\left(\max_{1\leq i\leq T}\tau_{0^d}^{(i)}\right)\right]$$

$$\leq c_{p_3}\sqrt{|\mathcal{A}|T\log_2 T\log\left(\frac{2|\mathcal{A}|T}{\tilde{\delta}}\right)} \mathbb{E}\left[(M_{\boldsymbol{\theta}^*}^T+h)^{d+r+r_*^p}\left(\max_{1\leq i\leq T}\tau_{0^d}^{(i)}\right)\right],$$

where $c_{p_3} := 48c_{p_2}$. Finally, from the above equation, (32), and (28),

$$R_2 \leq \tilde{\delta}\left(\log(h(T+1))+1\right)^d TQ(T)$$

$$+ c_{p_3}\sqrt{|\mathcal{A}|T\log_2 T\log\left(\frac{2|\mathcal{A}|T}{\tilde{\delta}}\right)} \mathbb{E}\left[(M_{\boldsymbol{\theta}^*}^T+h)^{d+r+r_*^p}\left(\max_{1\leq i\leq T}\tau_{0^d}^{(i)}\right)\right].$$

By choosing $\tilde{\delta} = \frac{1}{TQ(T)}$, we get

$R_2$

$$\leq (\log(h(T+1))+1)^d + c_{p_3}\sqrt{|\mathcal{A}|T\log_2 T\log(2|\mathcal{A}|T^2Q(T))} \mathbb{E}\left[(M_{\boldsymbol{\theta}^*}^T+h)^{d+r+r_*^p}\left(\max_{1\leq i\leq T}\tau_{0^d}^{(i)}\right)\right],$$

$$\leq (\log(h(T+1))+1)^d + c_{p_3}\sqrt{|\mathcal{A}|T\log_2\left(2|\mathcal{A}|T^2Q(T)\right)} \mathbb{E}\left[(M_{\boldsymbol{\theta}^*}^T+h)^{d+r+r_*^p}\left(\max_{1\leq i\leq T}\tau_{0^d}^{(i)}\right)\right],$$

where $Q(T) = c_{p_2}(Th)^{r+r_*^p}$. $\qquad\square$

## B.7 Proof of Theorem 1

*Proof.* Lemmas 1, 2, and 3 along with Cauchy-Schwarz inequality showed that the regret terms $R_0$ and $R_2$ are of the order $\tilde{O}(KrdJ^*h^{d+2r+r_*^p}\sqrt{|\mathcal{A}|T})$ and the term $R_1$ is $\tilde{O}(J^*(h)^{r_*^p})$. Therefore, from $R(T, \pi_{TSDE}) = R_0 + R_1 + R_2$, the regret of Algorithm 1, $R(T, \pi_{TSDE})$, is $\tilde{O}(KrdJ^*h^{d+2r+r_*^p}\sqrt{|\mathcal{A}|T})$. $\qquad\square$

## B.8 Requirement of an optimal policy oracle.

To implement our algorithm, we need to find the optimal policy for each model sampled by the algorithm—optimal policy for Theorem 1 and optimal policy within policy class $\Pi$ for Corollary 1; this has also been used in past work [23, 24, 36]. In the finite state-space setting, [49] provides a schedule of $\epsilon$ values and selects $\epsilon$-optimal policies to obtain $\tilde{O}(\sqrt{T})$ regret guarantees. The issue with extending the analysis of [49] to the countable state-space setting is that we need to ensure (uniform) ergodicity for the chosen $\epsilon$-optimal policies; the $\limsup$ or $\liminf$ of the time-average expected reward (used to define the average cost problem) being finite doesn't imply ergodicity. In other words, we must formulate (and verify) ergodicity assumptions for a potentially large set of close-to-optimal algorithms whose structure is undetermined. Another issue is that, to the best of our knowledge, there isn't a general structural characterization of all $\epsilon$-optimal stationary policies for countable state-space MDPs or even a characterization of the policy within this set that is selected by any computational procedure in the literature; current results only discuss existence and characterization of the stationary optimal policy. In the absence of such results, stability assumptions with the same uniformity across models as in our submission will be needed, which are likely too strong to be useful.

If we could verify the stability requirements of Assumptions 3 and 4 for a subset of policies, the optimal oracle is not needed, and instead, by choosing approximately optimal policies within this subset, we can follow the same proof steps as [49] to guarantee regret performance similar to Corollary 1 (without knowledge of model parameters). To theoretically analyze the performance of the algorithm that follows an approximately optimal policy rather than the optimal one, we assume that for a specific sequence of $\{\epsilon_k\}_{k=1}^{\infty}$, an $\epsilon_k$-optimal policy is given, which is defined below.

**Definition 1.** *Policy $\pi \in \Pi$ is called an $\epsilon$-optimal policy if for every $\theta \in \Theta$,*

$$c(\boldsymbol{x}, \pi(\boldsymbol{x})) + \sum_{\boldsymbol{y} \in \mathcal{X}} P_\theta(\boldsymbol{y}|\boldsymbol{x}, \pi(\boldsymbol{x}))v(\boldsymbol{y}; \theta) \leq c(\boldsymbol{x}, \pi_\theta^*(\boldsymbol{x})) + \sum_{\boldsymbol{y} \in \mathcal{X}} P_\theta(\boldsymbol{y}|\boldsymbol{x}, \pi_\theta^*(\boldsymbol{x}))v(\boldsymbol{y}; \theta) + \epsilon,$$

*where $\pi_\theta^*$ is the optimal policy in the policy class $\Pi$ corresponding to parameter $\theta$ and $v(.; \theta)$ is the solution to Poisson equation* (5).

Given $\epsilon$-optimal policies that satisfy Assumptions 3 and 4, in Theorem 2 we extend the regret guarantees of Corollary 1 to the algorithm employing $\epsilon$-optimal policy, instead of the best-in-class policy, and show that the same regret upper bounds continue to apply.

**Theorem 3.** *Consider a non-negative sequence $\{\epsilon_k\}_{k=1}^{\infty}$ such that for every $k \in \mathbb{N}$, $\epsilon_k$ is bounded above by $\frac{1}{k+1}$ and an $\epsilon_k$-optimal policy satisfying Assumptions 3 and 4 is given. The regret incurred by Algorithm 1 while using the $\epsilon_k$-optimal policy during any episode $k$ is $\tilde{O}(dh^d\sqrt{|\mathcal{A}|T})$.*

*Proof.* For the $\epsilon_k$-optimal policy used in episode $k$, shown by $\pi^{\epsilon_k}$, we have

$$c(\boldsymbol{x}, \pi^{\epsilon_k}(\boldsymbol{x})) + \sum_{\boldsymbol{y} \in \mathcal{X}} P_{\theta_k}(\boldsymbol{y}|\boldsymbol{x}, \pi^{\epsilon_k}(\boldsymbol{x}))v(\boldsymbol{y}; \theta_k) \leq c(\boldsymbol{x}, \pi_{\theta_k}^*(\boldsymbol{x})) + \sum_{\boldsymbol{y} \in \mathcal{X}} P_{\theta_k}(\boldsymbol{y}|\boldsymbol{x}, \pi_{\theta_k}^*(\boldsymbol{x}))v(\boldsymbol{y}; \theta_k) + \epsilon_k$$

$$= J(\theta_k) + v(\boldsymbol{x}; \theta_k) + \epsilon_k.$$

Thus,

$$R(T, \pi_{TSDE}) = \mathbb{E}\Big[\sum_{k=1}^{K_T}\sum_{t=t_k}^{t_{k+1}-1} c(\boldsymbol{X}(t), \pi^{\epsilon_k}(\boldsymbol{X}(t)))\Big] - T\,\mathbb{E}\left[J\left(\boldsymbol{\theta}^*\right)\right] = R_0 + R_1 + R_2 + \mathbb{E}\Big[\sum_{k=1}^{K_T} T_k \epsilon_k\Big]$$

with $R_0 = \mathbb{E}\Big[\sum_{k=1}^{K_T} T_k J(\theta_k)\Big] - T\,\mathbb{E}\left[J(\boldsymbol{\theta}^*)\right],$

$$R_1 = \mathbb{E}\Big[\sum_{k=1}^{K_T}\sum_{t=t_k}^{t_{k+1}-1}\Big[v(\boldsymbol{X}(t);\theta_k) - v(\boldsymbol{X}(t+1);\theta_k)\Big]\Big],$$

$$R_2 = \mathbb{E}\Big[\sum_{k=1}^{K_T}\sum_{t=t_k}^{t_{k+1}-1}\Big[v(\boldsymbol{X}(t+1);\theta_k) - \sum_{\boldsymbol{y}\in\mathcal{X}} P_{\theta_k}(\boldsymbol{y}|\boldsymbol{X}(t), \pi^{\epsilon_k}(\boldsymbol{X}(t)))v(\boldsymbol{y};\theta_k)\Big]\Big].$$

We assumed that given $\epsilon$-optimal policies satisfy Assumptions 3 and 4. As a result, we can utilize the proof of Theorem 1 to deduce that the term $R_0 + R_1 + R_2$ is of the order $\tilde{O}(dh^d\sqrt{|\mathcal{A}|T})$. Moreover, we can simplify the term $\mathbb{E}\Big[\sum_{k=1}^{K_T} T_k \epsilon_k\Big]$ as below:

$$\mathbb{E}\Big[\sum_{k=1}^{K_T} T_k \epsilon_k\Big] = \mathbb{E}\Big[\sum_{k=1}^{K_T} \tilde{T}_k \epsilon_k\Big] + \mathbb{E}\Big[\sum_{k=1}^{K_T} E_k \epsilon_k\Big]. \tag{35}$$

From the second stopping condition of Algorithm 1, we have $\tilde{T}_k \leq \tilde{T}_{k-1} + 1 \leq \ldots \leq k + 1$ and

$$\mathbb{E}\Big[\sum_{k=1}^{K_T} T_k \epsilon_k\Big] \leq \mathbb{E}[K_T],$$

where we have used the assumption that $\epsilon_k \leq \frac{1}{k+1}$. For the second term of (35), from (25)

$$\mathbb{E}\Big[\sum_{k=1}^{K_T} E_k \epsilon_k\Big] \leq \mathbb{E}\Big[\sum_{k=1}^{K_T} \frac{E_k}{k+1}\Big] \leq \mathbb{E}\Big[\max_{1\leq i\leq T} \tau_{0^d}^{(i)} \sum_{k=1}^{K_T} \frac{1}{k+1}\Big] \leq \mathbb{E}\Big[\max_{1\leq i\leq T} \tau_{0^d}^{(i)} \log(K_T+1)\Big], \tag{36}$$

where in the last inequality we have used $\sum_{i=1}^{n} \frac{1}{n} \leq 1 + \log(n)$. Finally, as a result of Lemma 6 and Lemma 8, the result follows. □

## C  Bounds on hitting times under polynomial and geometric ergodicity

### C.1  Polynomial upper bounds for the moments of hitting time of state $0^d$

For any $\theta_1, \theta_2 \in \Theta$, consider the Markov process with transition kernel $P_{\theta_1}^{\pi_{\theta_2}^*}$ obtained from the MDP $(\mathcal{X}, \mathcal{A}, c, P_{\theta_1})$ by following policy $\pi_{\theta_2}^*$. [29, Lemma 3.5] establishes that if the process is polynomially ergodic, equivalently satisfies (4), then for every $0 < \eta \leq 1$, there exists constants $\beta_{\theta_1,\theta_2}^\eta, b_{\theta_1,\theta_2}^\eta > 0$ such that the following holds:

$$\Delta\left(V_{\theta_1,\theta_2}^p\right)^\eta(\boldsymbol{x}) \leq -\beta_{\theta_1,\theta_2}^\eta \left(V_{\theta_1,\theta_2}^p(\boldsymbol{x})\right)^{\alpha_{\theta_1,\theta_2}^p + \eta - 1} + b_{\theta_1,\theta_2}^\eta \mathbb{I}_{C_{\theta_1,\theta_2}^p}(\boldsymbol{x}), \quad \boldsymbol{x} \in \mathcal{X}, \tag{37}$$

where for $\eta \in (0,1)$, $\tilde{\beta}_{\theta_1,\theta_2}^p := \min(\beta_{\theta_1,\theta_2}^p, 1)$ and

$$\beta_{\theta_1,\theta_2}^\eta = \eta\tilde{\beta}_{\theta_1,\theta_2}^p, \quad b_{\theta_1,\theta_2}^\eta = \left(b_{\theta_1,\theta_2}^p\right)^\eta + \eta\tilde{\beta}_{\theta_1,\theta_2}^p \max\left(1, \left(\tilde{\beta}_{\theta_1,\theta_2}^p\right)^{(\alpha_{\theta_1,\theta_2}^p + \eta - 1)/(1 - \alpha_{\theta_1,\theta_2}^p)}\right), \tag{38}$$

and for $\eta = 1$, $\beta_{\theta_1,\theta_2}^\eta = \beta_{\theta_1,\theta_2}^p$ and $b_{\theta_1,\theta_2}^\eta = b_{\theta_1,\theta_2}^p$. Consequently, the following result is immediate from the proof of [29, Theorem 3.6]; for completeness, we provide the proof in Appendix D.1.

**Lemma 9.** *Suppose a finite set $C^p_{\theta_1,\theta_2}$, constants $\beta^p_{\theta_1,\theta_2}, b^p_{\theta_1,\theta_2} > 0$, $r/(r+1) \leq \alpha^p_{\theta_1,\theta_2} < 1$, and a function $V^p_{\theta_1,\theta_2} : \mathcal{X} \to [1,+\infty)$ exist such that (4) holds. Then, there exist a sequence of non-negative functions $V^i_{\theta_1,\theta_2} : \mathcal{X} \to [1,+\infty)$ for $i = 0,\dots,r+1$ that satisfy the following system of drift equations for finite sets $C^i_{\theta_1,\theta_2}$, constants $b^i_{\theta_1,\theta_2} \geq 0$ and $\beta^i_{\theta_1,\theta_2} > 0$:*

$$\Delta V^{i-1}_{\theta_1,\theta_2}(\boldsymbol{x}) \leq -\beta^i_{\theta_1,\theta_2} V^i_{\theta_1,\theta_2}(\boldsymbol{x}) + b^i_{\theta_1,\theta_2} \mathbb{I}_{C^i_{\theta_1,\theta_2}}(\boldsymbol{x}), \qquad \boldsymbol{x} \in \mathcal{X}, \ i = 1,\dots,r+1. \quad (39)$$

Notice that $r$ is the maximum degree of the cost function $c$ defined in Assumption 1. Following the proof and approach of [29] and using the set of equations (39), we can find an upper-bound for $\mathbb{E}_{\boldsymbol{x}}[\tau^i_{0^d}]$ for $i = 1,\dots,r+1$ in Lemma 10. In order to establish upper bounds for the first $r+1$ moments of $\tau_{0^d}$, it is crucial to choose the value of $\alpha^p_{\theta_1,\theta_2}$ greater than or equal to $\frac{r}{r+1}$, as demonstrated in the proof of Lemma 10 in Appendix D.2

**Lemma 10.** *For $i = 1,\dots,r+1$, and for all $\boldsymbol{x} \in \mathcal{X}$*

$$\mathbb{E}^{\pi^*_{\theta_2}}_{\boldsymbol{x}}[(\tau_{0^d})^i] \leq i\phi^p_{\theta_1,\theta_2}(i) \left( V^p_{\theta_1,\theta_2}(\boldsymbol{x}) + b^p_{\theta_1,\theta_2} \alpha_{C^p_{\theta_1,\theta_2}} \right),$$

*where $\phi^p_{\theta_1,\theta_2}(i) := \prod_{j=1}^{i} \frac{1}{\beta^{\eta_j}_{\theta_1,\theta_2}} \left( 2^{j-1} + (j-1)\alpha_{C^p_{\theta_1,\theta_2}} b^{\eta_j}_{\theta_1,\theta_2} \right)$, $\eta_i = 1 - (i-1)(1 - \alpha^p_{\theta_1,\theta_2})$, $b^{\eta_i}_{\theta_1,\theta_2}$ and $\beta^{\eta_i}_{\theta_1,\theta_2}$ defined in (38), and $\alpha_{C^p_{\theta_1,\theta_2}} = \left( \min_{\boldsymbol{y} \in C^p_{\theta_1,\theta_2}} K_{\theta_1,\theta_2}(\boldsymbol{y}) \right)^{-1}$.*

Based on Lemma 10, we impose the conditions of Assumption 4 to obtain uniform (over model class) and polynomial (in norm of the state) upper-bounds on the moments of hitting times to $0^d$. Moreover, these conditions lead to a uniform characterization of parameters of Lemma 10 over all models in our class.

## C.2 Distribution of return times to state $0^d$

For any $\theta_1, \theta_2 \in \Theta$, consider the Markov process with transition kernel $P^{\pi^*_{\theta_2}}_{\theta_1}$ obtained from the MDP $(\mathcal{X}, \mathcal{A}, c, P_{\theta_1})$ by following policy $\pi^*_{\theta_2}$. In the following lemma, we show that the tail probabilities of the return times to the common state $0^d$, again $\tau_{0^d}$, converge geometrically fast to 0, and characterize the convergence parameters in terms of the constants given in Assumption 3. Explicitly, we show

$$\mathbb{P}_{0^d}(\tau_{0^d} > n) \leq c^g_{\theta_1,\theta_2} \left( \tilde{\gamma}^g_{\theta_1,\theta_2} \right)^n,$$

for problem and policy dependent constants $c^g_{\theta_1,\theta_2}$ and $\tilde{\gamma}^g_{\theta_1,\theta_2}$. We will follow the method outlined in [27] with the goal to identify problem dependent parameters that will be relevant to our results. Proof of the following lemma is given in Appendix D.3 and follows the methodology of [27].

**Lemma 11.** *For every $\theta_1, \theta_2 \in \Theta$ in the Markov process obtained from the Markov decision process $(\mathcal{X}, \mathcal{A}, c, P_{\theta_1})$ following policy $\pi^*_{\theta_2}$, the return time to state 0 starting from state 0 satisfies the following:*

$$\mathbb{P}_{0^d}(\tau_{0^d} > n) \leq c^g_{\theta_1,\theta_2} \left( \tilde{\gamma}^g_{\theta_1,\theta_2} \right)^n,$$

*where*

$$c^g_{\theta_1,\theta_2} = \frac{b^g_{\theta_1,\theta_2} \left( \tilde{b}^g_{\theta_1,\theta_2} \right)^2}{\tilde{b}^g_{\theta_1,\theta_2} - 1} \quad and \quad \tilde{\gamma}^g_{\theta_1,\theta_2} = 1 - \frac{1}{\tilde{b}^g_{\theta_1,\theta_2}},$$

*with*

$$\tilde{b}^g_{\theta_1,\theta_2} = \frac{3b^g_{\theta_1,\theta_2} + 1}{1 - \gamma^g_{\theta_1,\theta_2}} \left( |C^g_{\theta_1,\theta_2}|^2 \max \left( 1, \max_{\boldsymbol{u} \in C^g_{\theta_1,\theta_2} \setminus \{0^d\}} \mathbb{E}^{\pi^*_{\theta_2}}_{\boldsymbol{u}}[\tau_{0^d}] \right) \right).$$

Based on Lemma 11, it is necessary to impose the conditions in Assumption 3 to obtain uniform tail probability bounds on $\tau_{0^d}$ for all model parameters and policy choices in $\Theta$. Moreover, these conditions lead to a uniform characterization of $c^g_{\theta_1,\theta_2}$ and $\tilde{\gamma}^g_{\theta_1,\theta_2}$ over $\Theta$. Furthermore, as a result of Lemma 10 and uniformity conditions of Assumption 4, $\mathbb{E}^{\pi^*_{\theta_2}}_{\boldsymbol{u}}[\tau_{0^d}]$ has a uniform bound over $\Theta$ and $C^g_{\theta_1,\theta_2} \setminus \{0^d\}$, which can be characterized in terms of the polynomial Lyapunov function.

# D  Proofs of hitting time bounds

## D.1  Proof of Lemma 9

*Proof.* In the proof, to avoid cumbersome notation we will drop the indices $\theta_1, \theta_2$. Following the proof of Theorem 3.6 in [29], we choose $\eta_i = 1 - (i-1)(1-\alpha^p)$ for $i = 1, \ldots, r+1$ and note that as $\alpha^p \in [\frac{r}{r+1}, 1)$, we have $\eta_i \in [\frac{1}{r+1}, 1]$. As a result, we can apply (37) to each $\eta_i$ to get

$$\Delta \left( V^p \right)^{\eta_i} (\boldsymbol{x}) \leq -\beta^{\eta_i} \left( V^p(\boldsymbol{x}) \right)^{i\alpha^p - i + 1} + b^{\eta_i} \mathbb{I}_{C^p}(\boldsymbol{x}), \quad i = 1, \ldots, r+1.$$

Thus, the system of drift equations (39) hold for

$$\begin{aligned}
V_i &= (V^p)^{1 - i(1 - \alpha^p)}, & i &= 0, \ldots, r+1, \\
\beta_i &= \beta^{\eta_i}, & i &= 1, \ldots, r+1, \\
b_i &= b^{\eta_i}, & i &= 1, \ldots, r+1, \\
C_i &= C^p, & i &= 1, \ldots, r+1,
\end{aligned}$$

where $\beta^{\eta_i}$ and $b^{\eta_i}$ are defined in (38). $\qquad\square$

## D.2  Proof of Lemma 10

The proof of Lemma 10 uses the following lemma.

**Lemma 12** (Proposition 11.3.2, [43]). *Suppose for nonnegative functions $f$, $g$, and $V$ on the state space $\mathcal{X}$ and every $k \in \mathbb{Z}_+$, the following holds:*

$$\mathbb{E}[V(X_{k+1})|\mathcal{F}_k] \leq V(X_k) - f(X_k) + g(X_k).$$

*Then, for any initial condition $x$ and stopping time $\tau$*

$$\mathbb{E}_x \left[ \sum_{k=0}^{\tau-1} f(X_k) \right] \leq V(x) + \mathbb{E}_x \left[ \sum_{k=0}^{\tau-1} g(X_k) \right].$$

*Proof of Lemma 10.* Following [29], the proof uses an induction argument. We will use the notation of Lemma 9 for simplicity. Similarly, in this proof we will also denote $\phi^p_{\theta_1, \theta_2}(i)$ as $\phi(i)$, $K_{\theta_1, \theta_2}(\cdot)$ as $K(\cdot)$, and $V^i_{\theta_1, \theta_2}, b^i_{\theta_1, \theta_2}, \beta^i_{\theta_1, \theta_2}, C^i_{\theta_1, \theta_2}$ as $V_i, b_i, \beta_i, C_i$.

From irreducibility, for all $\boldsymbol{x} \in \mathcal{X}$, $K(\boldsymbol{x})$ is positive and finite. Considering the system of drift equations found in Lemma 9, $C_i = C^p$ is a finite set for all $i = 1, \ldots, r+1$. Thus, $\min_{\boldsymbol{y} \in C_i} K(\boldsymbol{y})$ is strictly positive. For all $\boldsymbol{x} \in \mathcal{X}$ and $i = 1, \ldots, r+1$, we have

$$\mathbb{I}_{C_i}(\boldsymbol{x}) \leq \left( \min_{\boldsymbol{y} \in C_i} K(\boldsymbol{y}) \right)^{-1} K(\boldsymbol{x}). \tag{40}$$

We set $\alpha_{C^p} := (\min_{\boldsymbol{y} \in C_i} K(\boldsymbol{y}))^{-1} = (\min_{\boldsymbol{y} \in C^p} K(\boldsymbol{y}))^{-1}$. From Lemma 9, for $j = 1$ and $\boldsymbol{x} \in \mathcal{X}$

$$\Delta V_0(\boldsymbol{x}) \leq -\beta_1 V_1(\boldsymbol{x}) + b_1 \mathbb{I}_{C_1}(\boldsymbol{x}).$$

By applying Lemma 12, for all $\boldsymbol{x} \in \mathcal{X}$ we get

$$\beta_1 \mathbb{E}_{\boldsymbol{x}} \left[ \sum_{k=0}^{\tau_{0^d}-1} V_1(\boldsymbol{X}_k) \right] \leq V_0(\boldsymbol{x}) + b_1 \mathbb{E}_{\boldsymbol{x}} \left[ \sum_{k=0}^{\tau_{0^d}-1} \mathbb{I}_{C_1}(\boldsymbol{X}_k) \right]. \tag{41}$$

Using (40) and (41), followed by noting that

$$K(\boldsymbol{x}) = \sum_{n=0}^{\infty} 2^{-n-2} P^n(\boldsymbol{x}, 0^d) = \sum_{n=0}^{\infty} 2^{-n-2} \mathbb{E}_{\boldsymbol{x}}[\mathbb{I}_{0^d}(\boldsymbol{X}_n)],$$

we get

$$\mathbb{E}_{\boldsymbol{x}}\left[\sum_{k=0}^{\tau_{0^d}-1} V_1\left(\boldsymbol{X}_k\right)\right] \leq \frac{1}{\beta_1} V_0(\boldsymbol{x}) + \frac{b_1\alpha_{C^p}}{\beta_1}\mathbb{E}_{\boldsymbol{x}}\left[\sum_{n=0}^{\infty} 2^{-n-2}\sum_{k=0}^{\tau_{0^d}-1} \mathbb{I}_{0^d}\left(\boldsymbol{X}_{k+n}\right)\right]$$

$$= \frac{1}{\beta_1} V_0(\boldsymbol{x}) + \frac{b_1\alpha_{C^p}}{\beta_1}\mathbb{E}_{\boldsymbol{x}}\left[\sum_{n=0}^{\infty} 2^{-n-2}\sum_{k=n}^{\tau_{0^d}-1+n} \mathbb{I}_{0^d}\left(\boldsymbol{X}_k\right)\right]$$

$$\leq \frac{1}{\beta_1} V_0(\boldsymbol{x}) + \frac{b_1\alpha_{C^p}}{\beta_1}\mathbb{E}_{\boldsymbol{x}}\left[\sum_{n=0}^{\infty} 2^{-n-2}\sum_{k=n\vee\tau_{0^d}}^{\tau_{0^d}-1+n} \mathbb{I}_{0^d}\left(\boldsymbol{X}_k\right)\right]$$

$$\leq \frac{1}{\beta_1} V_0(\boldsymbol{x}) + \frac{b_1\alpha_{C^p}}{\beta_1}\sum_{n=0}^{\infty} 2^{-n-2}(n+1)$$

$$= \frac{1}{\beta_1} V_0(\boldsymbol{x}) + \frac{b_1\alpha_{C^p}}{\beta_1}.$$

As $V_1(\boldsymbol{x}) \geq 1$, this gives us a bound on $\mathbb{E}_{\boldsymbol{x}}[\tau_{0^d}]$ as follows:

$$\mathbb{E}_{\boldsymbol{x}}[\tau_{0^d}] \leq \frac{1}{\beta_1} V_0(\boldsymbol{x}) + \frac{b_1\alpha_{C^p}}{\beta_1}.$$

Assume for $i \geq 1$, by the induction assumption we have

$$\mathbb{E}_{\boldsymbol{x}}\left[\sum_{k=0}^{\tau_{0^d}-1} (k+1)^{i-1}V_i\left(\boldsymbol{X}_k\right)\right] \leq \phi(i)\left(V_0(\boldsymbol{x}) + b_1\alpha_{C^p}\right). \tag{42}$$

Set $j = i + 1$ in (39), which yields

$$\Delta V_i(\boldsymbol{x}) \leq -\beta_{i+1}V_{i+1}(\boldsymbol{x}) + b_{i+1}\mathbb{I}_{C^p}(\boldsymbol{x}).$$

Define $Z_k = k^i V_i(\boldsymbol{X}_k)$. From the above equation, we have

$$\mathbb{E}[Z_{k+1}|\boldsymbol{X}_k] \leq (k+1)^i\left(V_i\left(\boldsymbol{X}_k\right) - \beta_{i+1}V_{i+1}(\boldsymbol{X}_k) + b_{i+1}\mathbb{I}_{C^p}(\boldsymbol{X}_k)\right)$$

$$\leq Z_k + 2^i(k+1)^{i-1}V_i\left(\boldsymbol{X}_k\right) + (k+1)^i b_{i+1}\mathbb{I}_{C^p}(\boldsymbol{X}_k) - (k+1)^i\beta_{i+1}V_{i+1}(\boldsymbol{X}_k).$$

By applying Lemma 12 to the above equation, we get

$$\beta_{i+1}\mathbb{E}_{\boldsymbol{x}}\left[\sum_{k=0}^{\tau_{0^d}-1} (k+1)^i V_{i+1}\left(\boldsymbol{X}_k\right)\right]$$

$$\leq 2^i\mathbb{E}_{\boldsymbol{x}}\left[\sum_{k=0}^{\tau_{0^d}-1} (k+1)^{i-1}V_i\left(\boldsymbol{X}_k\right)\right] + b_{i+1}\mathbb{E}_{\boldsymbol{x}}\left[\sum_{k=0}^{\tau_{0^d}-1} (k+1)^i\mathbb{I}_{C^p}\left(\boldsymbol{X}_k\right)\right]$$

$$\leq 2^i\phi(i)\left(V_0(\boldsymbol{x}) + b_1\alpha_{C^p}\right) + \alpha_{C^p}b_{i+1}\mathbb{E}_{\boldsymbol{x}}[(\tau_{0^d})^i], \tag{43}$$

where the second inequality follows from (40) and the induction hypothesis (42). Thereafter, from (42) (by using integral lower bound after using $V_i \geq 1$), we have

$$\frac{1}{i}\mathbb{E}_{\boldsymbol{x}}[(\tau_{0^d})^i] \leq \mathbb{E}_{\boldsymbol{x}}\left[\sum_{k=0}^{\tau_{0^d}-1} (k+1)^{i-1}V_i\left(\boldsymbol{X}_k\right)\right] \leq \phi(i)\left(V_0(\boldsymbol{x}) + b_1\alpha_{C^p}\right).$$

Substituting in (43), we get

$$\beta_{i+1}\mathbb{E}_{\boldsymbol{x}}\left[\sum_{k=0}^{\tau_{0^d}-1} (k+1)^i V_{i+1}\left(\boldsymbol{X}_k\right)\right] \leq 2^i\phi(i)\left(V_0(\boldsymbol{x}) + b_1\alpha_{C^p}\right) + ib_{i+1}\alpha_{C^p}\phi(i)\left(V_0(\boldsymbol{x}) + b_1\alpha_{C^p}\right)$$

$$= \left(2^i + ib_{i+1}\alpha_{C^p}\right)\phi(i)\left(V_0(\boldsymbol{x}) + b_1\alpha_{C^p}\right)$$

$$= \beta_{i+1}\phi(i+1)\left(V_0(\boldsymbol{x}) + b_1\alpha_{C^p}\right).$$

This completes the proof. $\qquad\square$

## D.3 Proof of Lemma 11

*Proof.* In the proof, to avoid cumbersome notation we will drop the indices $\theta_1, \theta_2$. Based on Assumption 3, there exists a finite set $C^g$, constants $b^g, \gamma^g \in (0,1)$, and a function $V^g : \mathcal{X} \to [1, +\infty)$ satisfying

$$\Delta V^g(\boldsymbol{x}) \leq -\left(1 - \gamma^g\right) V^g(\boldsymbol{x}) + b^g \mathbb{I}_{C^g}(\boldsymbol{x}), \quad \boldsymbol{x} \in \mathcal{X}. \tag{44}$$

For $n \geq 1$, define the $n$-step taboo probabilities [43] as

$$_A P_{\boldsymbol{x} B}^n = \mathbb{P}_{\boldsymbol{x}}\left(\boldsymbol{X}_n \in B, \tau_A > n\right),$$

where $A, B \subseteq \mathcal{X}$, and $\tau_A$ is the first hitting time of set $A$. We also let $_A P_{\boldsymbol{x} B}^0 = \mathbb{I}_B(\boldsymbol{x})$ and $\tilde{V}^g = \sum_{n=0}^{\infty} {}_{0^d} P^n V^g$. Applying the last exit decomposition on $C^g \setminus \{0^d\}$ for all $x \in \mathcal{X}$, we obtain

$$\tilde{V}^g(\boldsymbol{x})$$

$$= \sum_{n=0}^{\infty} \sum_{\boldsymbol{y} \in \mathcal{X}} {}_{0^d} P_{\boldsymbol{x} \boldsymbol{y}}^n V^g(\boldsymbol{y})$$

$$= V^g(\boldsymbol{x}) + \sum_{n=1}^{\infty} \sum_{\boldsymbol{y} \in \mathcal{X}} {}_{C^g} P_{\boldsymbol{x} \boldsymbol{y}}^n V^g(\boldsymbol{y})$$

$$+ \sum_{n=1}^{\infty} \sum_{\boldsymbol{y} \in \mathcal{X}} \sum_{m=1}^{n-1} \sum_{\boldsymbol{z} \in C^g \setminus \{0^d\}} {}_{0^d} P_{\boldsymbol{x} \boldsymbol{z}}^m {}_{C^g} P_{\boldsymbol{z} \boldsymbol{y}}^{n-m} V^g(\boldsymbol{y}) + \sum_{n=1}^{\infty} \sum_{\boldsymbol{y} \in \mathcal{X}} \sum_{\boldsymbol{z} \in C^g \setminus \{0^d\}} {}_{0^d} P_{\boldsymbol{x} \boldsymbol{z}}^n {}_{C^g} P_{\boldsymbol{z} \boldsymbol{y}}^0 V^g(\boldsymbol{y})$$

$$= V^g(\boldsymbol{x}) + \sum_{n=1}^{\infty} \sum_{\boldsymbol{y} \in \mathcal{X}} {}_{C^g} P_{\boldsymbol{x} \boldsymbol{y}}^n V^g(\boldsymbol{y}) \tag{45}$$

$$+ \underbrace{\sum_{\boldsymbol{y} \in \mathcal{X}} \sum_{\boldsymbol{z} \in C^g \setminus \{0^d\}} \left(\sum_{m=1}^{\infty} {}_{0^d} P_{\boldsymbol{x} \boldsymbol{z}}^m\right) \left(\sum_{n=1}^{\infty} {}_{C^g} P_{\boldsymbol{z} \boldsymbol{y}}^n V^g(\boldsymbol{y})\right)}_{\text{Term 1}} + \underbrace{\sum_{n=1}^{\infty} \sum_{\boldsymbol{z} \in C^g \setminus \{0^d\}} {}_{0^d} P_{\boldsymbol{x} \boldsymbol{z}}^n V^g(\boldsymbol{z})}_{\text{Term 2}}, \tag{46}$$

where we break up the trajectories starting at state $\boldsymbol{x}$ and reaching state $\boldsymbol{y}$ while avoiding state $0^d$ into two: ones that never visit the set $C^g$, and the others that visit $C^g \setminus \{0^d\}$ up until time $m$ but not afterwards and exit $C^g \setminus \{0^d\}$ at time $m$.

We first bound Term 1 in (46) by finding an upper bound for the probability term $\sum_{m=1}^{\infty} {}_{0^d} P_{\boldsymbol{x} \boldsymbol{z}}^m$ using the first entrance decomposition on $C^g \setminus \{0^d\}$ while noting that $\boldsymbol{z} \in C^g \setminus \{0^d\}$:

$$\sum_{m=1}^{\infty} {}_{0^d} P_{\boldsymbol{x} \boldsymbol{z}}^m = \sum_{m=1}^{\infty} \sum_{l=1}^{m} \sum_{\substack{\boldsymbol{u} \in C^g \setminus \{0^d\} \\ \boldsymbol{v} \notin C^g}} {}_{C^g} P_{\boldsymbol{x} \boldsymbol{v}}^{l-1} P_{\boldsymbol{v} \boldsymbol{u}} {}_{0^d} P_{\boldsymbol{u} \boldsymbol{z}}^{m-l}$$

$$= \sum_{\boldsymbol{u} \in C^g \setminus \{0^d\}} \left(\sum_{l=0}^{\infty} \sum_{\boldsymbol{v} \notin C^g} {}_{C^g} P_{\boldsymbol{x} \boldsymbol{v}}^l P_{\boldsymbol{v} \boldsymbol{u}}\right) \left(\sum_{m=0}^{\infty} {}_{0^d} P_{\boldsymbol{u} \boldsymbol{z}}^m\right)$$

$$\leq \sum_{\boldsymbol{u} \in C^g \setminus \{0^d\}} \sum_{m=0}^{\infty} {}_{0^d} P_{\boldsymbol{u} \boldsymbol{z}}^m$$

$$\leq \sum_{\boldsymbol{u} \in C^g \setminus \{0^d\}} \sum_{m=0}^{\infty} \mathbb{P}_{\boldsymbol{u}}\left(\tau_{0^d} > m\right)$$

$$\leq |C^g| \max_{\boldsymbol{u} \in C^g \setminus \{0^d\}} \mathbb{E}_{\boldsymbol{u}}[\tau_{0^d}], \tag{47}$$

where the third line follows from the fact that $\sum_{l=0}^{\infty} \sum_{\boldsymbol{v} \notin C^g} {}_{C^g} P_{\boldsymbol{x} \boldsymbol{v}}^l P_{\boldsymbol{v} \boldsymbol{u}}$ is the probability of entrance to $C^g$ through $\boldsymbol{u} \in C^g \setminus \{0\}$, so it is less than 1. Irreducibility and positive recurrence combined with $|C^g| < \infty$ imply that $\max_{\boldsymbol{u} \in C^g \setminus \{0^d\}} \mathbb{E}_{\boldsymbol{u}}[\tau_{0^d}] < \infty$, which shows $\sum_{m=0}^{\infty} {}_{0^d} P_{\boldsymbol{x} \boldsymbol{z}}^m$ is finite. Next,

by induction we prove that for $n \geq 1$ and $\boldsymbol{z} \in C^g \setminus \{0^d\}$ we have

$$\sum_{\boldsymbol{y} \in \mathcal{X}} {}_{C^g} P_{\boldsymbol{zy}}^n V^g(\boldsymbol{y}) \leq (\gamma^g)^{n-1} b^g. \tag{48}$$

For $n = 1$, we have using Assumption 3 that

$$\sum_{\boldsymbol{y} \in \mathcal{X}} {}_{C^g} P_{\boldsymbol{zy}} V^g(\boldsymbol{y}) \leq \sum_{\boldsymbol{y} \in \mathcal{X}} P_{\boldsymbol{zy}} V^g(\boldsymbol{y}) \leq b^g.$$

Assuming that (48) holds for $n$, for $n + 1$ we have

$$\sum_{\boldsymbol{y} \in \mathcal{X}} {}_{C^g} P_{\boldsymbol{zy}}^{n+1} V^g(\boldsymbol{y}) \leq \sum_{\substack{\boldsymbol{y} \in \mathcal{X} \\ \boldsymbol{v} \notin C^g}} {}_{C^g} P_{\boldsymbol{zv}}^n P_{\boldsymbol{vy}} V^g(\boldsymbol{y}) \leq \gamma^g \sum_{\boldsymbol{v} \notin C^g} {}_{C^g} P_{\boldsymbol{zv}}^n V^g(\boldsymbol{v}) \quad \text{(Using (44))}$$

$$\leq \gamma \sum_{\boldsymbol{v} \in \mathcal{X}} {}_{C^g} P_{\boldsymbol{zv}}^n V^g(\boldsymbol{v}) \leq (\gamma^g)^n b^g, \qquad \text{(By induction step)}$$

so (48) is shown. We collect these bounds later on for our result on Term 2.

We now simplify the summation in (45). Similar to previous arguments, we will use induction for $n \geq 1$ and show for all $\boldsymbol{x} \in \mathcal{X}$

$$\sum_{\boldsymbol{y} \in \mathcal{X}} {}_{C^g} P_{\boldsymbol{xy}}^n V^g(\boldsymbol{y}) \leq (\gamma^g)^{n-1} \left( \gamma^g V^g(\boldsymbol{x}) + b^g \right). \tag{49}$$

For $n = 1$, we have

$$\sum_{\boldsymbol{y} \in \mathcal{X}} {}_{C^g} P_{\boldsymbol{xy}} V^g(\boldsymbol{y}) \leq \sum_{\boldsymbol{y} \in \mathcal{X}} P_{\boldsymbol{xy}} V^g(\boldsymbol{y}) \leq \gamma^g V^g(\boldsymbol{x}) + b^g.$$

Assuming that (49) holds for $n$, for $n + 1$ we have

$$\sum_{\boldsymbol{y} \in \mathcal{X}} {}_{C^g} P_{\boldsymbol{xy}}^{n+1} V^g(\boldsymbol{y}) \leq \sum_{\boldsymbol{z} \notin C^g} {}_{C^g} P_{\boldsymbol{xz}}^n \sum_{\boldsymbol{y} \in \mathcal{X}} P_{\boldsymbol{zy}} V^g(\boldsymbol{y}) \leq \gamma^g \sum_{\boldsymbol{z} \notin C^g} {}_{C^g} P_{\boldsymbol{xz}}^n V^g(\boldsymbol{z})$$

$$\leq \gamma^g \sum_{\boldsymbol{z} \in \mathcal{X}} {}_{C^g} P_{\boldsymbol{xz}}^n V^g(\boldsymbol{z}) \leq (\gamma^g)^n \left( \gamma^g V^g(\boldsymbol{x}) + b^g \right),$$

where the first and second inequalities follow from the definition of taboo probabilities and (44). Thus, (49) is proved. Lastly, for Term 2 in (46), we note

$$\sum_{n=1}^{\infty} \sum_{\boldsymbol{z} \in C^g \setminus \{0^d\}} {}_{0^d} P_{\boldsymbol{xz}}^n V^g(\boldsymbol{z}) \leq \max_{\boldsymbol{y} \in C^g \setminus \{0^d\}} V^g(\boldsymbol{y}) \sum_{\boldsymbol{z} \in C^g \setminus \{0^d\}} \sum_{n=1}^{\infty} {}_{0^d} P_{\boldsymbol{xz}}^n$$

$$\leq b^g |C^g|^2 \max_{\boldsymbol{u} \in C^g \setminus \{0^d\}} \mathbb{E}_{\boldsymbol{u}}[\tau_{0^d}] \qquad \text{(From (47))}.$$

From the above equation, (47), (48), and (49), we bound $\tilde{V}^g(\boldsymbol{x})$ as follows:

$\tilde{V}^g(\boldsymbol{x})$

$$\leq V^g(\boldsymbol{x}) + (\gamma^g V^g(\boldsymbol{x}) + b^g) \sum_{n=1}^{\infty} (\gamma^g)^{n-1} + |C^g|^2 b^g \max_{\boldsymbol{u} \in C^g \setminus \{0^d\}} \mathbb{E}_{\boldsymbol{u}}[\tau_{0^d}] \left( 1 + \sum_{n=1}^{\infty} (\gamma^g)^{n-1} \right)$$

$$\leq \frac{V^g(\boldsymbol{x})}{1 - \gamma^g} + \frac{3 |C^g|^2 b^g}{1 - \gamma^g} \max \left( 1, \max_{\boldsymbol{u} \in C^g \setminus \{0^d\}} \mathbb{E}_{\boldsymbol{u}}[\tau_{0^d}] \right)$$

$$\leq V^g(\boldsymbol{x}) \left( \frac{3 b^g + 1}{1 - \gamma^g} \left( |C^g|^2 \max \left( 1, \max_{\boldsymbol{u} \in C^g \setminus \{0^d\}} \mathbb{E}_{\boldsymbol{u}}[\tau_{0^d}] \right) \right) \right),$$

where the last line is due to $V^g(\boldsymbol{x}) \geq 1$. Taking

$$\tilde{b}^g := \frac{3 b^g + 1}{1 - \gamma^g} \left( |C^g|^2 \max \left( 1, \max_{\boldsymbol{u} \in C^g \setminus \{0^d\}} \mathbb{E}_{\boldsymbol{u}}[\tau_{0^d}] \right) \right) > 1,$$

we have shown that

$$\tilde{V}^g(\boldsymbol{x}) \leq \tilde{b}^g V^g(\boldsymbol{x}), \quad \boldsymbol{x} \in \mathcal{X}. \tag{50}$$

We now upper-bound $\mathbb{P}_{0^d}(\tau_{0^d} > n)$ for all $n \geq 1$ in an inductive manner, starting with $\mathbb{P}_{0^d}(\tau_{0^d} > 1)$. As a part of showing this, for every $\boldsymbol{x} \neq 0^d$ we argue that for all $n \geq 1$

$$\mathbb{P}_{\boldsymbol{x}}(\tau_{0^d} > n) \leq \tilde{V}^g(\boldsymbol{x}) \left(1 - \frac{1}{\tilde{b}^g}\right)^n. \tag{51}$$

First note that

$$\tilde{V}^g(\boldsymbol{x}) \geq V^g(\boldsymbol{x}) \geq 1. \tag{52}$$

Thus,

$$\mathbb{P}_{\boldsymbol{x}}(\tau_{0^d} > 1) = \sum_{\boldsymbol{y} \in \mathcal{X}} {}_{0^d} P_{\boldsymbol{xy}} \leq \sum_{\boldsymbol{y} \in \mathcal{X}} {}_{0^d} P_{\boldsymbol{xy}} \tilde{V}^g(\boldsymbol{y})$$

$$= \sum_{\boldsymbol{y} \in \mathcal{X}} {}_{0^d} P_{\boldsymbol{xy}} \sum_{n=0}^{\infty} \sum_{\boldsymbol{z} \in \mathcal{X}} {}_{0^d} P_{\boldsymbol{yz}}^n V^g(\boldsymbol{z}) = \sum_{\boldsymbol{z} \in \mathcal{X}} \sum_{n=1}^{\infty} {}_{0^d} P_{\boldsymbol{xz}}^n V^g(\boldsymbol{z}). \tag{53}$$

We now apply the bound in (50) to get

$$\mathbb{P}_{\boldsymbol{x}}(\tau_{0^d} > 1) \leq \sum_{\boldsymbol{z} \in \mathcal{X}} \sum_{n=1}^{\infty} {}_{0^d} P_{\boldsymbol{xz}}^n V^g(\boldsymbol{z}) = \tilde{V}^g(\boldsymbol{x}) - V^g(\boldsymbol{x}) \leq \tilde{V}^g(\boldsymbol{x}) \left(1 - \frac{1}{\tilde{b}^g}\right). \tag{54}$$

With the base of induction established, we assume the statement in (51) is true for $n$, and show that it continues to hold for $n + 1$ as follows:

$$\mathbb{P}_{\boldsymbol{x}}(\tau_{0^d} > n + 1) = \sum_{\boldsymbol{y} \neq 0^d} P_{\boldsymbol{xy}} \mathbb{P}_{\boldsymbol{y}}(\tau_{0^d} > n)$$

$$\leq \left(1 - \frac{1}{\tilde{b}^g}\right)^n \sum_{\boldsymbol{y} \neq 0^d} P_{\boldsymbol{xy}} \tilde{V}^g(\boldsymbol{y})$$

$$\leq \tilde{V}^g(\boldsymbol{x}) \left(1 - \frac{1}{\tilde{b}^g}\right)^{n+1},$$

where the final inequality uses the same arguments as in (53) and (54).

Finally, using the tail probabilities of hitting time of state $0^d$ from any state $\boldsymbol{x} \neq 0^d$, we bound the tail probability of the return time to state $0^d$ (starting from $0^d$) as follows

$$\mathbb{P}_{0^d}(\tau_{0^d} > n + 1) = \sum_{\boldsymbol{x} \neq 0^d} P_{0\boldsymbol{x}} \mathbb{P}_{\boldsymbol{x}}(\tau_{0^d} > n) \leq \left(1 - \frac{1}{\tilde{b}^g}\right)^n \sum_{\boldsymbol{x} \neq 0^d} P_{0\boldsymbol{x}} \tilde{V}^g(\boldsymbol{x})$$

$$\leq \tilde{b}^g \left(1 - \frac{1}{\tilde{b}^g}\right)^n \sum_{\boldsymbol{x} \neq 0^d} P_{0\boldsymbol{x}} V^g(\boldsymbol{x}) \leq b^g \tilde{b}^g \left(1 - \frac{1}{\tilde{b}^g}\right)^n,$$

where the final inequality follows from the definition of $b^g$, and we have

$$\tilde{\gamma}^g = 1 - \frac{1}{\tilde{b}^g}, \text{ and } c^g = \frac{b^g \left(\tilde{b}^g\right)^2}{\tilde{b}^g - 1},$$

and the proof is complete. $\qquad\square$

# E  Queueing model examples

## E.1  Model 1: Two-server queueing system with a common buffer

We consider a continuous-time queueing system with two heterogeneous servers with unknown service rate vector $\boldsymbol{\theta}^* = (\theta_1^*, \theta_2^*)$ and a common infinite buffer, shown in Figure 2a. Arrivals to the system are according to a Poisson process with rate $\lambda$ and service times are exponentially distributed

with parameter $\theta_i^*$, depending on the assigned server. The service rate vector $\boldsymbol{\theta}^*$ is sampled from the prior distribution $\nu_0$ defined on the space $\Theta$ given as

$$\Theta = \left\{ (\theta_1, \theta_2) \in \mathbb{R}_+^2 : \frac{\lambda}{\theta_1 + \theta_2} \leq \frac{1-\delta}{1+\delta}, 1 \leq \frac{\theta_1}{\theta_2} \leq R \right\}, \tag{55}$$

for fixed $\delta \in (0, 0.5)$ and $R \geq 1$. Note that for any $(\theta_1, \theta_2) \in \Theta$, we have $\theta_1 \geq \theta_2$ and the stability requirement $\lambda < \theta_1 + \theta_2$ holds. The countable state space $\mathcal{X}$ is defined as $\mathcal{X} = \{\boldsymbol{x} = (x_0, x_1, x_2) : x_0 \in \mathbb{N} \cup \{0\}, x_1, x_2 \in \{0, 1\}\}$, in which $x_0$ is the length of the queue, and $x_i, i = 1, 2$ is equal to 1 if server $i$ is busy serving a job. At each time instance $r \in \mathbb{R}_+$, the dispatcher can assign jobs from the (non-empty) buffer to an available server. Thus, the action space $\mathcal{A}$ is equal to

$$\mathcal{A} = \{h, b, 1, 2\},$$

where $h$ indicates no action, $b$ sends a job to both of the servers, and $i = 1, 2$ assigns a job to server $i$. The goal of the dispatcher is to minimize the expected sojourn time of customers, which by Little's law [52] is equivalent to minimizing the average number of customers in the system, or

$$\inf_{\pi \in \Pi} \limsup_{T \to \infty} \frac{1}{T} \int_0^T \|\boldsymbol{X}(r)\|_1 \, dr, \tag{56}$$

where $\boldsymbol{X}(r)$ is the state of the system at time $r \in \mathbb{R}_+$, immediately after the arrival/departure and just before the action is taken. In [38], it is argued that from uniformization [39] and sampling the continuous-time Markov process at a rate of $\lambda + \theta_1^* + \theta_2^*$, a discrete-time Markov chain is obtained, which converts the original continuous-time problem shown in (56) to an equivalent discrete-time problem as below:

$$\inf_{\pi \in \Pi} \limsup_{T \to \infty} \frac{1}{T} \int_0^T \|\boldsymbol{X}(r)\|_1 \, dr = \inf_{\pi \in \Pi} \limsup_{T \to \infty} \frac{1}{T} \sum_{i=0}^{T-1} \|\boldsymbol{X}(i)\|_1. \tag{57}$$

To obtain a uniform sampling rate of $\lambda + \theta_1^* + \theta_2^*$, the continuous-time system is sampled at arrivals, real and dummy customer departures. In [38], it is further shown that the optimal policy that achieves the infimum in (57) is a threshold policy $\pi_t$ with the optimal finite threshold $t(\theta) \in \mathbb{N}$, with the policy defined as below:

$$\pi_t(\boldsymbol{x}) = \begin{cases} h & \text{if } \{x_0 = 0\} \text{ or } \{\|\boldsymbol{x}\|_1 \leq t, x_1 = 1\} \text{ or } \{x_1 = x_2 = 1\} \\ 1 & \text{if } \{x_0 \geq 1, x_1 = 0\} \\ 2 & \text{if } \{x_0 \geq 1, \|\boldsymbol{x}\|_1 \geq t+1, x_1 = 1, x_2 = 0\}; \end{cases}$$

note that action $b$ is not used. Policy $\pi_t$ assigns a job to the faster (first) server whenever there is a job waiting in the queue and the first server is available. In contrast, $\pi_t$ dispatches a job to the second server only if the number of jobs in the system are greater than threshold $t$ and the second server is available. If neither of these conditions hold, no action or $h$ is taken. Consequently, we can restrict the set of all policies $\Pi$ in (57) to the set $\Pi_t$, which is the set of all possible threshold policies corresponding to some $t \in \mathbb{N}$.

In the rest of this subsection, our aim is to show that Assumptions 1-5 are satisfied for the discrete-time Markov process obtained by uniformization of the described queueing system and hence, conclude that Algorithm 1 can be used to learn the unknown service rate vector $\boldsymbol{\theta}^*$ with the expected regret of order $\tilde{O}(\sqrt{T})$.

**Assumption 1.** Cost function is given as $c(\boldsymbol{x}, a) = \|\boldsymbol{x}\|_1$, which satisfies Assumption 1 with $f_c(\boldsymbol{x}) = x_0 + x_1 + x_2$ and $K = r = 1$.

**Assumption 2.** For any state-action pair $(\boldsymbol{x}, a)$ and $\theta \in \Theta$, we have $P_\theta(A(\boldsymbol{x}); \boldsymbol{x}, a) = 0$ where $A(\boldsymbol{x}) = \{\boldsymbol{y} \in \mathcal{X} : |\|\boldsymbol{y}\|_1 - \|\boldsymbol{x}\|_1| > 1\}$; thus, Assumption 2 holds with $h = 1$.

**Assumption 3.** Consider a queueing system with parameter $\theta$ following threshold policy $\pi_t$ for some $t \in \mathbb{N}$. The uniformized discrete-time Markov chain is irreducible and aperiodic on a subset of state space given as $\mathcal{X}_t = \mathcal{X} \setminus (\{(i, 0, 0) : i \geq \min(t, 2)\} \cup \{(0, 1, 1)\})$. In [38], it is proved that for every $t$, the chain consists of a single positive recurrent class and the corresponding average number of customers, depicted by $J^t(\theta)$, is calculated. Moreover, it is shown that for every $\theta \in \Theta$ the optimal threshold $t(\theta)$ can be numerically found as the smallest $i \in \mathbb{N}$ for which $J^i(\theta) < J^{i+1}(\theta)$. Define the set $T^*$ as the set of all optimal thresholds corresponding to at least one $\theta \in \Theta$, or

$$T^* = \{t : t = t(\theta) \text{ for } \theta \in \Theta\}.$$

**Remark 7.** *There is a discrepancy between the class of MDPs defined in this section and in Section 2, as in the former the MDPs are not irreducible in the whole state space $\mathcal{X}$. Specifically, for every Markov process generated by a queueing system with parameter $\theta$ following threshold policy $\pi_t$, irreducibility holds on $\mathcal{X}_t \subset \mathcal{X}$. Nevertheless, the results of Section 4 are valid as starting from state $(0,0)$, the visited states are positive recurrent; see Remark 4.*

In the following proposition, we verify the geometric ergodicity of the discrete-time chain governed by any parameter $\theta \in \Theta$ and obtained by following any threshold policy $\pi_t$ for $t \in T^*$; proof is given in Appendix F.1.

**Proposition 1.** *The discrete-time Markov process obtained from the queueing system governed by parameter $\theta = (\theta_1, \theta_2) \in \Theta$ and following threshold policy $\pi_t$ for some $t \in T^*$ is geometrically ergodic. Equivalently, the following holds*

$$\Delta V_{\theta,t}^g(\boldsymbol{x}) \leq - \left(1 - \gamma_{\theta,t}^g\right) V_{\theta,t}^g(\boldsymbol{x}) + b_{\theta,t}^g \mathbb{I}_{C_{\theta,t}^g}(\boldsymbol{x}), \quad \boldsymbol{x} \in \mathcal{X}_t,$$

*for*

$$V_{\theta,t}^g(\boldsymbol{x}) = \exp(-\log(1-\delta)\|\boldsymbol{x}\|_1),$$
$$C_{\theta,t}^g = \{(x_0, x_1, 0) : x_0 < t\} \cup \{(0,0,1)\}, \tag{58}$$
$$b_{\theta,t}^g = \max_{\boldsymbol{x} \in C_{\theta,t}^g} \exp\left(-\log(1-\delta)\left(\|\boldsymbol{x}\|_1 + 1\right)\right), \tag{59}$$
$$\gamma_{\theta,t}^g = \frac{1}{2} - \frac{1}{2(\theta_1 + \theta_2 + \lambda)}\left((\theta_1 + \theta_2)(1-\delta) + \lambda(1-\delta)^{-1}\right). \tag{60}$$

Having described all the terms explicitly, we verify the rest of the conditions of Assumption 3, which lead to uniform (over model class) upper-bounds on the moments of hitting time to $0^d$ as follows:

1. From (60), $\sup_{\theta \in \Theta, t \in T^*} \gamma_{\theta,t}^g \leq 1/2 < 1$.

2. From (58), we can see that state $(0,0)$ belongs to $C_{\theta,t}^g$ for all $\theta \in \Theta$ and $t \in T^*$. In order for $C_*^g = \cup_{\theta \in \Theta, t \in T^*} C_{\theta,t}^g$ to be a finite set, the supremum of the optimal threshold $t(\theta)$ over $\Theta$ should be finite. In [37] with service rate vector $(\theta_1, \theta_2)$, it is shown that the optimal threshold is bounded above by $\sqrt{2}\theta_1/\theta_2$, which further gives

$$t(\theta) \leq \sqrt{2}\frac{\theta_1}{\theta_2} \leq \sqrt{2}R. \tag{61}$$

Thus, $\sup_{\theta \in \Theta} t(\theta) \leq \sqrt{2}R$, which is finite. To confirm a uniform upper bound for $b_{\theta,t}^g$, we note that from (59),

$$\sup_{\theta \in \Theta, t \in T^*} b_{\theta,t}^g = \frac{2-\delta}{1-\delta} \max_{x \in C_*^g} \exp(-\log(1-\delta)\|\boldsymbol{x}\|_1),$$

which is finite as $|C_*^g| < \infty$.

**Assumption 4.** To find an upper bound on the second moment of hitting times, we verify Assumption 4 and show that there exists a finite set $C_{\theta,t}^p$, constants $\beta_{\theta,t}^p, b_{\theta,t}^p > 0$, $r/(r+1) \leq \alpha_{\theta,t}^p < 1$, and a function $V_{\theta,t}^p : \mathcal{X}_t \to [1, +\infty)$ satisfying

$$\Delta V_{\theta,t}^p(\boldsymbol{x}) \leq -\beta_{\theta,t}^p \left(V_{\theta,t}^p(\boldsymbol{x})\right)^{\alpha_{\theta,t}^p} + b_{\theta,t}^p \mathbb{I}_{C_{\theta,t}^p}(\boldsymbol{x}), \quad \boldsymbol{x} \in \mathcal{X}_t. \tag{62}$$

**Proposition 2.** *The discrete-time Markov process obtained from the queueing system governed by parameter $\theta = (\theta_1, \theta_2) \in \Theta$ and following threshold policy $\pi_t$ for some $t \in T^*$ is polynomially*

*ergodic. This is true because* (62) *holds for*

$$V_{\theta,t}^p(\boldsymbol{x}) = \|\boldsymbol{x}\|_1^2, \tag{63}$$

$$C_{\theta,t}^p = \{(x_0, x_1, 0) : x_0 < t\} \cup \left\{(x_0, x_1, x_2) : x_0 < \frac{2\lambda}{\theta_1 + \theta_2 - \lambda}, x_1 + x_2 \geq 1\right\}, \tag{64}$$

$$b_{\theta,t}^p = \max_{\boldsymbol{x} \in C_{\theta,t}^p} (\|\boldsymbol{x}\|_1 + 1)^2), \tag{65}$$

$$\beta_{\theta,t}^p = 1 - \frac{2\lambda}{\theta_1 + \theta_2 + \lambda}, \tag{66}$$

$$\alpha_{\theta,t}^p = \frac{1}{2}. \tag{67}$$

Proof of Proposition 2 is given in Appendix F.2. We define the normalized rates as $\tilde{\lambda} = \frac{\lambda}{\lambda + \theta_1 + \theta_2}$ and $\tilde{\theta}_i = \frac{\theta_i}{\lambda + \theta_1 + \theta_2}$, for $i = 1, 2$. From the choice of parameter space $\Theta$, we have $\tilde{\lambda} \leq 0.5 - 0.5\delta$, $\tilde{\theta}_1 + \tilde{\theta}_2 \geq 0.5 + 0.5\delta$, and $\tilde{\theta}_1 \geq 0.25 + 0.25\delta$. We verify the remaining conditions of Assumption 4 as follows:

1. From (63), the first condition holds with $r_*^p = 2$ and $s_*^p = 2$.

2. From (64), we can see that state $(0,0)$ belongs to $C_{\theta,t}^p$ for all $\theta \in \Theta$ and $t \in T^*$. Furthermore,

$$\sup_{\theta \in \Theta, t \in T^*} \frac{2\lambda}{\theta_1 + \theta_2 - \lambda} \leq \frac{1 - \delta}{\delta},$$

   which follows from the stability condition $\tilde{\lambda} \leq 0.5 - 0.5\delta$. Thus, from the definition of $C_{\theta,t}^p$ in (64), and the fact that $\sup_{\theta \in \Theta} t(\theta) \leq \sqrt{2}R$ as argued in in (61), $C_*^p = \cup_{\theta \in \Theta, t \in T^*} C_{\theta,t}^p$ is a finite set. We also note that $\sup_{\theta \in \Theta, t \in T^*} b_{\theta,t}^p$ is finite as $|C_*^p| < \infty$. It remains to show that $\inf_{\theta \in \Theta, t \in T^*} \beta_{\theta,t}^p$ is positive, which is equivalent to verifying that $\sup_{\theta \in \Theta, t \in T^*} \tilde{\lambda} < 1/2$, which follows from the stability condition $\tilde{\lambda} \leq 0.5 - 0.5\delta$.

3. We need to show that $K_{\theta,t}(\boldsymbol{x}) := \sum_{n=0}^{\infty} 2^{-n-2} (P_\theta^t)^n (\boldsymbol{x}, 0^d)$ is strictly bounded away from zero. We notice that from any non-zero state $\boldsymbol{x}$, the queueing system hits $0^d$ in $\|\boldsymbol{x}\|_1$ transitions only if all transitions are real departures. Hence,

$$\begin{aligned}
K_{\theta,t}(\boldsymbol{x}) &\geq 2^{-\|\boldsymbol{x}\|_1 - 2} (P_\theta^t)^{\|\boldsymbol{x}\|_1} (\boldsymbol{x}, 0^d) \\
&\geq 2^{-\|\boldsymbol{x}\|_1 - 2} (\tilde{\theta}_1)^{\|\boldsymbol{x}\|_1} (\tilde{\theta}_2)^{\|\boldsymbol{x}\|_1} \\
&\geq 2^{-\|\boldsymbol{x}\|_1 - 2} R^{-\|\boldsymbol{x}\|_1} (\tilde{\theta}_1)^{2\|\boldsymbol{x}\|_1} \\
&\geq 2^{-\|\boldsymbol{x}\|_1 - 2} R^{-\|\boldsymbol{x}\|_1} \left(\frac{1}{4} + \frac{\delta}{4}\right)^{2\|\boldsymbol{x}\|_1},
\end{aligned}$$

   where the third and fourth inequalities follow from the definition of $\Theta$ in (55). Thus, the infimum of $K_{\theta,t}(\boldsymbol{x})$ over the finite set $C_*^p$ and sets $\Theta$ and $T^*$ is strictly greater than zero.

**Assumption 5.** We finally verify Assumption 5, which asserts that $\sup_{\theta \in \Theta} J(\theta)$ is finite. We have

$$J(\theta) = \mathbb{E}_{\boldsymbol{X} \sim \mu_{\theta,t(\theta)}} [c(\boldsymbol{X})] = \mathbb{E}_{\boldsymbol{X} \sim \mu_{\theta,t(\theta)}} [\|\boldsymbol{X}\|_1] = \mathbb{E}_{\boldsymbol{X} \sim \mu_{\theta,t(\theta)}} \left[\sqrt{V_{\theta,t(\theta)}^p(\boldsymbol{X})}\right],$$

where $\mu_{\theta,t(\theta)}$ is the stationary distribution of the discrete-time process governed by parameter $\theta$ and following the optimal policy according to $\theta$. From (62) and [43, Theorem 14.3.7],

$$\mu_{\theta,t(\theta)} \left(\sqrt{V_{\theta,t(\theta)}^p(\boldsymbol{X})}\right) \leq \frac{b_*^p}{\beta_*^p},$$

which is finite from the the previously verified assumption. Consequently,

$$\sup_{\theta \in \Theta} J(\theta) \leq \frac{b_*^p}{\beta_*^p} < \infty.$$

## E.2   Model 2: Two heterogeneous parallel queues

We consider two parallel queues with infinite buffers, each with its own single server, and unknown service rate vector $\boldsymbol{\theta}^* = (\theta_1^*, \theta_2^*)$, shown in Figure 2b. The service rate vector $\boldsymbol{\theta}^*$ is sampled from the prior distribution $\nu_0$ defined on the space $\Theta$ given as

$$\Theta = \left\{ (\theta_1, \theta_2) \in \mathbb{R}_+^2 : \frac{\lambda}{\theta_1 + \theta_2} \leq \frac{1-\delta}{1+\delta}, 1 \leq \frac{\theta_1}{\theta_2} \leq R \right\}, \tag{68}$$

for fixed $\delta \in (0, 0.5)$ and $R \geq 1$, which ensures the stability of the queueing system. Consider the discrete-time MDP $(\mathcal{X}, \mathcal{A}, P_{\boldsymbol{\theta}^*}, c)$ obtained by sampling the queueing system at the Poisson arrival sequence. The countably infinite state space $\mathcal{X}$ is defined as below

$$\mathcal{X} = \{ \boldsymbol{x} = (x_1, x_2) : x_i \in \mathbb{N} \cup \{0\} \},$$

where the state of the system is the number of jobs in the server-queue pair $i$ just before an arrival. Furthermore, the action space $\mathcal{A}$ is equal to

$$\mathcal{A} = \{1, 2\},$$

where action $i \in \mathcal{A}$ indicates the arrival dispatched to queue $i$. The unbounded cost function $c : \mathcal{X} \times \mathcal{A} \to \mathbb{N} \cup \{0\}$ is defined as the total number of jobs in the queueing system, i.e., $c(\boldsymbol{x}, a) = \|\boldsymbol{x}\|_1$. For every $\omega \in \mathbb{R}_+$, we define policy $\pi_\omega : \mathcal{X} \to \mathcal{A}$, which routes the arrival according to the weighted queue lengths, as

$$\pi_\omega(\boldsymbol{x}) = \arg\min \left( 1 + x_1, \omega \left( 1 + x_2 \right) \right),$$

where the tie is broken in favor of the first server. We also define policy class $\tilde{\Pi}$ as the set of policies $\pi_\omega$ such that $\omega$ belongs to a compact interval; in other words,

$$\tilde{\Pi} = \left\{ \pi_\omega; \, \omega \in \left[ \frac{1}{c_R R}, c_R R \right] \right\},$$

where $R$ is defined in (68) and $c_R \geq 1$. We aim to minimize the infinite-horizon average cost in the policy class $\tilde{\Pi}$, that is,

$$J(\theta) = \inf_{\pi \in \tilde{\Pi}} \limsup_{T \to \infty} \frac{1}{T} \mathbb{E} \left[ \sum_{t=1}^{T} c \left( \boldsymbol{X}(t), A(t) \right) \right], \tag{69}$$

where $\boldsymbol{X}(t) = (X_1(t), X_2(t))$ is the occupancy vector of the queueing system just before arrival $t$. Even with the controlled Markov process transition kernel fully-specified (by the values of the arrival rate and the two service rates), the optimal policy[1] that satisfies (69) in policy class $\tilde{\Pi}$ is not known except when $\theta_1 = \theta_2$ where the optimal value is $\omega = 1$, and so, to learn it, we will use Proximal Policy Optimization for countable state-space controlled Markov processes as developed in [18]. Note that [18] requires full knowledge of the controlled Markov process, which holds in our learning scheme since we use the parameters sampled from the posterior for determining the policy at the beginning of each episode. Furthermore, for each policy in the set of applicable policies $\tilde{\Pi}$, [18] also requires that the resulting Markov process be geometrically ergodic, which we will establish below.

**Proposition 3.** *The discrete-time Markov process obtained from the queueing system governed by parameter $\theta = (\theta_1, \theta_2) \in \Theta$ and following policy $\pi_\omega \in \tilde{\Pi}$ is geometrically ergodic. Equivalently, the following holds*

$$\Delta V_{\theta,\omega}^g(\boldsymbol{x}) \leq - \left( 1 - \gamma_{\theta,\omega}^g \right) V_{\theta,\omega}^g(\boldsymbol{x}) + b_{\theta,\omega}^g \mathbb{I}_{C_{\theta,\omega}^g}(\boldsymbol{x}), \quad \boldsymbol{x} \in \mathcal{X}, \tag{70}$$

---

[1]When $\theta_1 = \theta_2$, then the policy with $\omega = 1$ (Join-the-Shortest-Queue) is the optimal policy [19] for the underling MDP.

*for*

$$V_{\theta,\omega}^g(\boldsymbol{x}) = \frac{\omega}{\omega+1} \exp\left(a_{\theta,\omega}^g \frac{x_1+1}{\omega}\right) + \frac{1}{\omega+1} \exp\left(a_{\theta,\omega}^g (x_2+1)\right),$$

$$a_{\theta,\omega}^g = \min\left(\omega \log(1+\delta), \log(1+\delta), \omega \log \frac{1-0.5\delta}{1-\delta}, \log \frac{1-0.5\delta}{1-\delta}, \frac{\delta(1-\delta^2)}{4c_R R(1-0.5\delta)}\right), \tag{71}$$

$$C_{\theta,\omega}^g = \left\{(x_1,x_2) \in \mathcal{X} : x_i \le \max\left(x_{i,\theta,\omega}^{g_j}, 0\right), i, j = 1, 2\right\}, \tag{72}$$

$$b_{\theta,\omega}^g = \max_{\boldsymbol{x} \in C_{\theta,\omega}^g} \left(\frac{2\omega}{\omega+1} \exp\left(a_{\theta,\omega}^g \frac{x_1+2}{\omega}\right) + \frac{2}{\omega+1} \exp\left(a_{\theta,\omega}^g (x_2+2)\right)\right), \tag{73}$$

$$\gamma_{\theta,\omega}^g = \frac{1}{2} + \frac{1}{2} \max\left(\zeta_{1,\theta,\omega}, \zeta_{2,\theta,\omega}, \frac{\zeta_{1,\theta,\omega}\omega}{1+\omega} \exp\left(\frac{a_{\theta,\omega}^g}{\omega}\right) + \frac{\zeta_{2,\theta,\omega}}{1+\omega}, \frac{\zeta_{1,\theta,\omega}\omega}{1+\omega} + \frac{\zeta_{2,\theta,\omega}}{1+\omega} \exp\left(a_{\theta,\omega}^g\right)\right), \tag{74}$$

*and problem-dependent constants $x_{i,\theta,\omega}^{g_j}$ and $\zeta_{i,\theta,\omega}$ for $i,j = 1,2$.*

Proof of Proposition 3 is given in Appendix F.3. In the rest of this subsection, our aim is to show that Assumptions 1-5 are satisfied for the discrete-time MDP and conclude that Algorithm 1 can be used to learn the unknown service rate vector $\boldsymbol{\theta}^*$ with expected regret of order $\tilde{O}(\sqrt{T})$.

**Assumption 1.** Cost function is given as $c(\boldsymbol{x}, a) = \|\boldsymbol{x}\|_1$, which satisfies Assumption 1 with $f_c(\boldsymbol{x}) = x_0 + x_1 + x_2$ and $K = r = 1$.

**Assumption 2.** For any state-action pair $(\boldsymbol{x}, a)$ and $\theta \in \Theta$, we have $P_\theta(A(\boldsymbol{x}); \boldsymbol{x}, a) = 0$ where $A(\boldsymbol{x}) = \{\boldsymbol{y} \in \mathcal{X} : \|\boldsymbol{y}\|_1 - \|\boldsymbol{x}\|_1 > 1\}$; thus, the MDP is skip-free to the right with $h = 1$. Moreover, from any $(\boldsymbol{x}, a)$, the finite set $\{\boldsymbol{y} \in \mathcal{X} : \|\boldsymbol{y}\|_1 \le \|\boldsymbol{x}\|_1 + 1\}$ is only accessible in one step; thus, Assumption 2 holds.

**Assumption 3.** In Proposition 3, we verified the geometric ergodicity of the discrete-time chain governed by parameter $\theta = (\theta_1, \theta_2) \in \Theta$ and following policy $\pi_\omega \in \tilde{\Pi}$ and thus, it only remains to verify the uniform model conditions. We define the normalized rates as $\tilde{\lambda} = \frac{\lambda}{\lambda+\theta_1+\theta_2}$ and $\tilde{\theta}_i = \frac{\theta_i}{\lambda+\theta_1+\theta_2}$, for $i = 1, 2$. From the choice of parameter space $\Theta$, we have $\tilde{\lambda} \le 0.5 - 0.5\delta$, $\tilde{\theta}_1 + \tilde{\theta}_2 \ge 0.5 + 0.5\delta$, and $\tilde{\theta}_1 \ge 0.25 + 0.25\delta$.

1. We first argue that $\zeta_{1,\theta,\omega}$ is bounded away from 1 as follows

$$
\begin{aligned}
1 - \zeta_{1,\theta,\omega} &= 1 - \frac{\frac{\lambda}{\theta_1+\lambda}}{1 - \exp\left(-\frac{a_{\theta,\omega}^g}{\omega}\right)\frac{\theta_1}{\theta_1+\lambda}} = \frac{\frac{\theta_1}{\theta_1+\lambda}\left(1 - \exp\left(-\frac{a_{\theta,\omega}^g}{\omega}\right)\right)}{1 - \exp\left(-\frac{a_{\theta,\omega}^g}{\omega}\right)\frac{\theta_1}{\theta_1+\lambda}} \\
&\ge \frac{\theta_1}{\theta_1+\lambda}\left(1 - \exp\left(-\frac{a_{\theta,(c_R R)^{-1}}^g}{c_R R}\right)\right) > \tilde{\theta}_1\left(1 - \exp\left(-\frac{a_{\theta,(c_R R)^{-1}}^g}{c_R R}\right)\right) \\
&> (0.25 + 0.25\delta)\left(1 - \exp\left(-\frac{a_{\theta,(c_R R)^{-1}}^g}{c_R R}\right)\right),
\end{aligned}
$$

where the first line follows from the definition of $\zeta_{1,\theta,\omega}$ in Appendix F.3, the second line from (71) and the definition of policy class $\tilde{\Pi}$. As $a_{\theta,\omega}^g$ does not depend on $\theta$, $\sup_{\theta \in \Theta, \omega \in [\frac{1}{c_R R}, c_R R]} \zeta_{1,\theta,\omega} < 1$. Furthermore, by similar arguments it can be shown that $\zeta_{2,\theta,\omega}$ is bounded away from 1. We next argue that $\frac{\zeta_{1,\theta,\omega}\omega}{1+\omega} \exp\left(\frac{a_{\theta,\omega}^g}{\omega}\right) + \frac{\zeta_{2,\theta,\omega}}{1+\omega}$ is bounded away from 1 using an upper bound found in

Appendix F.3 as below,

$$1 - \frac{\zeta_{1,\theta,\omega}\omega}{1+\omega}\exp\left(\frac{a^g_{\theta,\omega}}{\omega}\right) - \frac{\zeta_{2,\theta,\omega}}{1+\omega}$$

$$\geq 1 - \frac{\frac{\lambda}{1+\omega}\left(\omega + a^g_{\theta,\omega}\zeta_4\right)}{\lambda + \frac{\theta_1 a^g_{\theta,\omega}\zeta_3}{\omega}} - \frac{\frac{\lambda}{1+\omega}}{\lambda + \theta_2 a^g_{\theta,\omega}\zeta_3}$$

$$= \frac{a^g_{\theta,\omega}\left(-a^g_{\theta,\omega}\zeta_3\theta_2\left(\lambda\zeta_4 - \frac{\zeta_3\theta_1(1+\omega)}{\omega}\right) + \lambda\zeta_3\left(\theta_1 + \theta_2\right) - \lambda^2\zeta_4\right)}{(1+\omega)(\lambda + \theta_1 a^g_{\theta,\omega}\zeta_3\omega^{-1})(\lambda + \theta_2 a^g_{\theta,\omega}\zeta_3)}$$

$$> \frac{(\zeta_3 a^g_{\theta,\omega})^2\,\tilde{\theta}_1\tilde{\theta}_2}{\omega(\tilde{\lambda} + \tilde{\theta}_1 a^g_{\theta,\omega}\zeta_3\omega^{-1})(\tilde{\lambda} + \tilde{\theta}_2 a^g_{\theta,\omega}\zeta_3)}$$

$$> \frac{(\zeta_3 a^g_{\theta,(c_R R)^{-1}})^2(0.25 + 0.25\delta)^2}{c_R R^2(1 + c_R R\zeta_3 a^g_{\theta,c_R R})^2}, \tag{75}$$

where $\zeta_3 = (1+\delta)^{-1}$, $\zeta_4 = \frac{1-0.5\delta}{1-\delta}$, and we have used the arguments of Appendix F.3 and the definition of $\Theta$. Using a similar argument, we can show that $\frac{\zeta_{1,\theta,\omega}\omega}{1+\omega} + \frac{\zeta_{2,\theta,\omega}}{1+\omega}\exp\left(a^g_{\theta,\omega}\right)$ is bounded away from one, and finally, we conclude that $\sup_{\theta\in\Theta,\omega\in[\frac{1}{c_R R},c_R R]}\gamma^g_{\theta,\omega} < 1$.

2. From (72), we can see that state $(0,0)$ belongs to $C^g_{\theta,\omega}$ for all $\theta \in \Theta$ and $\omega \in [\frac{1}{c_R R}, c_R R]$. In order for $C^g_*$ to be a finite set, the supremum of $x^{g_j}_{i,\theta,\omega}$ over $\Theta$ and $\tilde{\Pi}$ should be finite. From the definition of $x^{g_1}_{1,\theta,\omega}$ in Appendix F.3,

$$x^{g_1}_{1,\theta,\omega} = \frac{\omega}{a^g_{\theta,\omega}}\log\frac{(c_R R + 1)\exp(c_R R a^g_{\theta,\omega})}{(\omega+1)\gamma^g_{\theta,\omega} - \omega\zeta_{1,\theta,\omega}\exp\left(\frac{a^g_{\theta,\omega}}{\omega}\right) - \zeta_{2,\theta,\omega}}$$

$$\leq \frac{c_R R}{a^g_{\theta,(c_R R)^{-1}}}\log\frac{(c_R R + 1)\exp(c_R R a^g_{\theta,c_R R})}{(\omega+1)\gamma^g_{\theta,\omega} - \omega\zeta_{1,\theta,\omega}\exp\left(\frac{a^g_{\theta,\omega}}{\omega}\right) - \zeta_{2,\theta,\omega}},$$

and we can derive a lower bound for the denominator from (75). Similarly, we can show that $\sup_{\theta\in\Theta,\omega\in[\frac{1}{c_R R},c_R R]}x^{g_2}_{2,\theta,\omega}$ is finite. We next find a uniform upper bound for $x^{g_1}_{2,\theta,\omega}$ from Appendix F.3,

$$x^{g_1}_{2,\theta,\omega}$$

$$= \frac{1}{a^g_{\theta,\omega}}\log\frac{(c_R R + 1)\exp(c_R R a^g_{\theta,\omega}) + \omega\exp\left(a^g_{\theta,\omega}\frac{x^{g_1}_{1,\theta,\omega}+1}{\omega}\right)\left(\zeta_{1,\theta,\omega}\exp\left(\frac{a^g_{\theta,\omega}}{\omega}\right) - \gamma^g_{\theta,\omega}\right)}{\gamma^g_{\theta,\omega} - \zeta_{2,\theta,\omega}}$$

$$\leq \frac{1}{a^g_{\theta,(c_R R)^{-1}}}\log\frac{(2c_R R + 1)\exp\left(c_R R a^g_{\theta,c_R R}\left(x^{g_1}_{1,\theta,\omega} + 2\right)\right)}{1 - \gamma^g_{\theta,\omega}},$$

which is uniformly bounded as $\gamma^g_{\theta,\omega}$ is uniformly bounded away from 1 and the second line follows from (74) and the fact that $\gamma^g_{\theta,\omega} - \zeta_{2,\theta,\omega} \geq 1 - \gamma^g_{\theta,\omega}$. Arguments verifying the finiteness of the supremum of $x^{g_2}_{1,\theta,\omega}$ follow similarly, and we conclude that $|C^g_*| < \infty$. To confirm a uniform upper bound for $b^g_{\theta,\omega}$, we note that from (73),

$$\sup_{\theta\in\Theta,\omega\in[\frac{1}{c_R R},c_R R]}b^g_{\theta,\omega} \leq \max_{x\in C^g_*}\left(2\exp\left(c_R R a^g_{\theta,c_R R}(x_1 + 2)\right) + 2\exp\left(a^g_{\theta,c_R R}(x_2 + 2)\right)\right),$$

which is finite as $a^g_{\theta,c_R R}$ is independent of the choice of $\theta$ and $|C^g_*| < \infty$.

**Assumption 4.** We next verify Assumption 4 and show that there exists a finite set $C^p_{\theta,\omega}$, constants $\beta^p_{\theta,\omega}, b^p_{\theta,\omega} > 0$, $r/(r+1) \leq \alpha^p_{\theta,\omega} < 1$, and a function $V^p_{\theta,\omega} : \mathcal{X} \to [1,+\infty)$ satisfying

$$\Delta V^p_{\theta,\omega}(\boldsymbol{x}) \leq -\beta^p_{\theta,\omega}\left(V^p_{\theta,\omega}(\boldsymbol{x})\right)^{\alpha^p_{\theta,\omega}} + b^p_{\theta,t}\mathbb{I}_{C^p_{\theta,\omega}}(\boldsymbol{x}), \quad \boldsymbol{x}\in\mathcal{X}. \tag{76}$$

**Proposition 4.** *The discrete-time Markov process obtained from the queueing system governed by parameter $\theta = (\theta_1, \theta_2) \in \Theta$ and following policy $\pi_\omega \in \tilde{\Pi}$ is polynomially ergodic. This follow because (76) holds for*

$$V_{\theta,\omega}^p(\boldsymbol{x}) = \frac{x_1^2}{\omega} + x_2^2, \tag{77}$$

$$C_{\theta,\omega}^p = \left\{ (x_1, x_2) \in \mathcal{X} : x_i \leq \left( 16 c_R^2 R^{3-i} + 101 c_R R \right) \frac{\lambda + \theta_i}{\theta_i}, i = 1, 2 \right\}, \tag{78}$$

$$\beta_{\theta,\omega}^p = \min \left( \frac{\theta_2}{2(\theta_2 + \lambda)\sqrt{\omega + 1}}, \frac{\theta_1 + \theta_2 - \lambda}{(\theta_1 + \theta_2 + \lambda)\sqrt{\omega + 1}}, \frac{\theta_2}{2(\theta_2 + \lambda)}, \frac{\theta_1}{2(\theta_1 + \lambda)\sqrt{\omega}} \right), \tag{79}$$

$$b_{\theta,\omega}^p = (\beta_{\theta,\omega}^p + 1) \max_{\boldsymbol{x} \in C_{\theta,\omega}^p} \left( \frac{(x_1 + 1)^2}{\omega} + (x_2 + 1)^2 \right), \tag{80}$$

$$\alpha_{\theta,\omega}^p = \frac{1}{2}. \tag{81}$$

Proof of Proposition 4 is given in Appendix F.4. Next, we verify the remaining conditions of Assumption 4.

1. From (77) and the fact that $\omega \in [\frac{1}{c_R R}, c_R R]$, the first condition holds with $r_*^p = 2$ and $s_*^p = \sup_{\theta \in \Theta, \omega \in [\frac{1}{c_R R}, c_R R]} s_{\theta,\omega} = c_R R + 1$.

2. From (78), state $(0,0)$ belongs to $C_{\theta,\omega}^p$ for all $\theta \in \Theta$ and $\omega \in [\frac{1}{c_R R}, c_R R]$. Furthermore, for $i = 1, 2$,

$$\sup_{\theta \in \Theta, \omega \in [\frac{1}{c_R R}, c_R R]} \frac{\lambda + \theta_i}{\theta_i} \leq \sup_{\theta \in \Theta} \frac{1}{\theta_2} \leq \sup_{\theta \in \Theta} \frac{R}{\theta_1} \leq \frac{4R}{1 + \delta}, \tag{82}$$

which follows from the fact that $\theta_1 \leq R\theta_2$ and $\tilde{\theta}_1 \geq 0.25 + 0.25\delta$. Thus, from the definition of $C_{\theta,\omega}^p$ in (78), $C_*^p = \cup_{\theta \in \Theta, \omega \in [\frac{1}{c_R R}, c_R R]} C_{\theta,\omega}^p$ is a finite set. We next verify that the infimum of $\beta_{\theta,\omega}^p$, found in (79), is positive. In (82), we showed that infimum of $\frac{\lambda + \theta_i}{\theta_i}$ over $\Theta$ is lower bounded by $\frac{1+\delta}{4}$. From this, the fact that $\omega$ belongs to a compact set, and $\theta_1 + \theta_2 + \lambda \geq \delta$, it follows that $\inf_{\theta \in \Theta, \omega \in [\frac{1}{c_R R}, c_R R]} \beta_{\theta,\omega}^p > 0$. Furthermore, it is easy to see that $\beta_{\theta,\omega}^p \leq \sqrt{c_R R}$. Hence, from (80),

$$\sup_{\theta \in \Theta, \omega \in [\frac{1}{c_R R}, c_R R]} b_{\theta,\omega}^p = \sup_{\theta \in \Theta, \omega \in [\frac{1}{c_R R}, c_R R]} (\beta_{\theta,\omega}^p + 1) \max_{\boldsymbol{x} \in C_{\theta,\omega}^p} \left( \frac{(x_1 + 1)^2}{\omega} + (x_2 + 1)^2 \right)$$

$$\leq (\sqrt{c_R R} + 1) \max_{\boldsymbol{x} \in C_*^p} \left( c_R R (x_1 + 1)^2 + (x_2 + 1)^2 \right),$$

which is finite as $|C_*^p| < \infty$.

3. We need to show that $K_{\theta,\omega}(\boldsymbol{x}) := \sum_{n=0}^{\infty} 2^{-n-2} (P_\theta^{\pi_\omega})^n (\boldsymbol{x}, 0^d)$ is strictly bounded away from zero. We show this using the fact that from any state $\boldsymbol{x}$, the queueing system hits $(0,0)$ in one step with positive probability. Take $x_{i,\theta,\omega} = \max_{\boldsymbol{x} \in C_{\theta,\omega}} x_i$ for $i = 1, 2$. We have

$$\inf_{\theta \in \Theta, \omega \in [\frac{1}{c_R R}, c_R R]} \min_{\boldsymbol{x} \in C_{\theta,\omega}} K(\boldsymbol{x}) \geq \inf_{\theta \in \Theta, \omega \in [\frac{1}{c_R R}, c_R R]} \min_{\boldsymbol{x} \in C_{\theta,\omega}} P(\boldsymbol{x}, 0^d)$$

$$\geq \inf_{\theta \in \Theta, \omega \in [\frac{1}{c_R R}, c_R R]} P\left( (x_{1,\theta,\omega}, x_{2,\theta,\omega}), 0^d \right).$$

The infimum in the right-hand side of the above equation is attained for the minimum normalized service rates possible for each server, or $\tilde{\theta}_1 = \frac{1+\delta}{4}$ and $\tilde{\theta}_2 = \frac{1+\delta}{4R}$. Therefore, the infimum of $K_{\theta,\omega}(\boldsymbol{x})$ over the finite set $C_*^p$, $\Theta$, and interval $[\frac{1}{c_R R}, c_R R]$ is strictly greater than zero.

**Assumption 5.** We finally verify that $\sup_{\theta \in \Theta} J(\theta)$ is finite. We first note that for $\boldsymbol{x} = (x_1, x_2)$,

$$(x_1 + x_2)^2 \leq 2 \max(\omega^*(\theta), 1) \left( \frac{x_1^2}{\omega^*(\theta)} + x_2^2 \right) = 2 \max(\omega^*(\theta), 1) V_{\theta,\omega^*(\theta)}^p(\boldsymbol{x}).$$

From the above equation,

$$J(\theta) = \mathbb{E}_{\boldsymbol{X} \sim \mu_{\theta,\omega^*(\theta)}} \left[ c(\boldsymbol{X}) \right]$$

$$= \mathbb{E}_{\boldsymbol{X} \sim \mu_{\theta,\omega^*(\theta)}} \left[ \|\boldsymbol{X}\|_1 \right]$$

$$\leq \sqrt{2 \max(\omega^*(\theta), 1)} \mathbb{E}_{\boldsymbol{X} \sim \mu_{\theta,\omega^*(\theta)}} \left[ \sqrt{V^p_{\theta,\omega^*(\theta)}(\boldsymbol{X})} \right],$$

where $\mu_{\theta,\omega^*(\theta)}$ is the stationary distribution of the discrete-time process governed by parameter $\theta$ and following the best in-class policy according to $\theta$, shown by $\pi_{\omega^*(\theta)}$. From [43], Theorem 14.3.7,

$$\mu_{\theta,\omega^*(\theta)} \left( \sqrt{V^p_{\theta,\omega^*(\theta)}(\boldsymbol{X})} \right) \leq \frac{\sup_{\theta \in \Theta, \omega \in [\frac{1}{c_R R}, c_R R]} b^p_{\theta,\omega}}{\beta^p_*},$$

which is finite from the the previous verified assumption. Thus,

$$\sup_{\theta \in \Theta} J(\theta) \leq \frac{\sqrt{2 c_R R} \left( \sup_{\theta \in \Theta, \omega \in [\frac{1}{c_R R}, c_R R]} b^p_{\theta,\omega} \right)}{\beta^p_*} < \infty.$$

# F  Proofs related to the queueing model examples

## F.1  Proof of Proposition 1

*Proof.* We define the normalized rates as

$$\tilde{\lambda} = \frac{\lambda}{\lambda + \theta_1 + \theta_2}, \quad \tilde{\theta}_i = \frac{\theta_i}{\lambda + \theta_1 + \theta_2}, \tag{83}$$

for $i = 1, 2$. From the choice of parameter space $\Theta$, we have $\tilde{\lambda} \leq 0.5 - 0.5\delta$, $\theta_1 + \theta_2 \geq 0.5 + 0.5\delta$, and $\theta_1 \geq 0.25 + 0.25\delta$. To prove geometric ergodicity, from the discussions of Section 2, it suffices to show that there exists a finite set $C^g_{\theta,t}$, constants $b^g_{\theta,t} > 0$, $\gamma^g_{\theta,t} \in (0,1)$, and a function $V^g_{\theta,t} : \mathcal{X}_t \to [1, +\infty)$ satisfying

$$\Delta V^g_{\theta,t}(\boldsymbol{x}) \leq - \left( 1 - \gamma^g_{\theta,t} \right) V^g_{\theta,t}(\boldsymbol{x}) + b^g_{\theta,t} \mathbb{I}_{C^g_{\theta,t}}(\boldsymbol{x}), \quad \boldsymbol{x} \in \mathcal{X}_t. \tag{84}$$

Take $V^g_{\theta,t}(\boldsymbol{x}) = \exp(a^g_{\theta,t} \|\boldsymbol{x}\|_1)$ for some $a^g_{\theta,t} > 0$. For $i \geq 1$ and $\boldsymbol{x} = (i, 1, 1)$,

$$P^t_\theta V^g_{\theta,t}(i,1,1) = \tilde{\lambda} V^g_{\theta,t}(i+1,1,1) + \tilde{\theta}_1 V^g_{\theta,t}(i,0,1) + \tilde{\theta}_2 V^g_{\theta,t}(i,1,0),$$

where $P^t_\theta$ is the corresponding transition kernel. Thus,

$$P^t_\theta V^g_{\theta,t}(i,1,1) - (1 - \gamma^g_{\theta,t}) V^g_{\theta,t}(i,1,1)$$

$$= \tilde{\lambda} \exp\left( a^g_{\theta,t}(i+3) \right) + (\tilde{\theta}_1 + \tilde{\theta}_2) \exp\left( a^g_{\theta,t}(i+1) \right) - (1 - \gamma^g_{\theta,t}) \exp\left( a^g_{\theta,t}(i+2) \right)$$

$$= \exp\left( a^g_{\theta,t}(i+1) \right) \left( \tilde{\lambda} \exp(2 a^g_{\theta,t}) + \tilde{\theta}_1 + \tilde{\theta}_2 - (1 - \gamma^g_{\theta,t}) \exp(a^g_{\theta,t}) \right).$$

Take $\tilde{a}_{\theta,t} = \exp(a^g_{\theta,t})$. We need to find $\tilde{a}_{\theta,t} > 1$ and $0 < \gamma^g_{\theta,t} < 1$ such that

$$\tilde{\lambda} \tilde{a}^2_{\theta,t} - (1 - \gamma^g_{\theta,t}) \tilde{a}_{\theta,t} + \tilde{\theta}_1 + \tilde{\theta}_2 < 0. \tag{85}$$

Take $\tilde{a}_{\theta,t} = (1 - \delta)^{-1} > 1$ and

$$\tilde{\gamma}_{\theta,t} := 1 - \gamma^g_{\theta,t} = \frac{1}{2} \left( 1 + (1 - \tilde{\lambda})(1 - \delta) + \tilde{\lambda}(1 - \delta)^{-1} \right).$$

We need to have $\tilde{\gamma}_{\theta,t} < 1$ which follows from the stability condition $\tilde{\lambda} \leq 0.5 - 0.5\delta$ as below:

$$\tilde{\gamma}_{\theta,t} = \frac{1}{2} + \frac{1}{2} \left( (1 - \tilde{\lambda})(1 - \delta) + \frac{\tilde{\lambda}}{1 - \delta} \right) = \frac{1}{2} + \frac{1}{2} \left( 1 - \delta - \tilde{\lambda}(1 - \delta) + \frac{\tilde{\lambda}}{1 - \delta} \right)$$

$$= \frac{1}{2} + \frac{1}{2} \left( 1 - \delta + \tilde{\lambda} \frac{1 - (1 - \delta)^2}{1 - \delta} \right) = \frac{1}{2} + \frac{1}{2} \left( 1 - \delta + \tilde{\lambda} \frac{\delta(2 - \delta)}{1 - \delta} \right)$$

$$\leq \frac{1}{2} + \frac{1}{2} \left( 1 - \delta + \frac{\delta(2 - \delta)}{2} \right) = 1 - \frac{\delta^2}{4} < 1.$$

We now verify (85):

$$\tilde{\lambda}\tilde{a}_{\theta,t}^2 - (1-\gamma_{\theta,t}^g)\tilde{a}_{\theta,t} + \tilde{\theta}_1 + \tilde{\theta}_2 = \frac{\tilde{\lambda}}{(1-\delta)^2} - \frac{1}{2(1-\delta)} - \frac{1-\tilde{\lambda}}{2} - \frac{\tilde{\lambda}}{2(1-\delta)^2} + 1 - \tilde{\lambda}$$

$$= \frac{\tilde{\lambda}}{2(1-\delta)^2} + \frac{1-\tilde{\lambda}}{2} - \frac{1}{2(1-\delta)}$$

$$= \frac{1}{2(1-\delta)^2}\left(\tilde{\lambda} + (1-\tilde{\lambda})(1-\delta)^2 - (1-\delta)\right)$$

$$= \frac{\delta}{2(1-\delta)^2}\left(\delta - 1 - \tilde{\lambda}\delta + 2\tilde{\lambda}\right)$$

$$= \frac{\delta}{2(1-\delta)^2}\left(\tilde{\lambda}(2-\delta) + \delta - 1\right)$$

$$< 0,$$

where the last line follows from $\tilde{\lambda} \le 0.5 - 0.5\delta < (1-\delta)/(2-\delta)$.

For $\boldsymbol{x} = (i,0,1)$ and $i \ge 1$, we have

$$P_\theta^t V_{\theta,t}^g(i,0,1) = \tilde{\lambda}V_{\theta,t}^g(i,1,1) + \tilde{\theta}_1 V_{\theta,t}^g(i-1,0,1) + \tilde{\theta}_2 V_{\theta,t}^g(i-1,1,0),$$

and

$$P_\theta^t V_{\theta,t}^g(i,0,1) - (1-\gamma_{\theta,t}^g)V_{\theta,t}^g(i,0,1)$$
$$= \tilde{\lambda}\exp\left(a_{\theta,t}^g(i+2)\right) + (\tilde{\theta}_1 + \tilde{\theta}_2)\exp\left(a_{\theta,t}^g i\right) - (1-\gamma_{\theta,t}^g)\exp\left(a_{\theta,t}^g(i+1)\right)$$
$$= \exp\left(a_{\theta,t}^g i\right)\left(\tilde{\lambda}\exp(2a_{\theta,t}^g) + \tilde{\theta}_1 + \tilde{\theta}_2 - (1-\gamma_{\theta,t}^g)\exp(a_{\theta,t}^g)\right),$$

which results in the same conditions as previously discussed. When $\boldsymbol{x} = (i,1,0)$ and $i \ge t$ also same argument holds.

Finally, (84) holds for

$$C_{\theta,t}^g = \{(x_0, x_1, 0) : x_0 < t\} \cup \{(0,0,1)\},$$
$$a_{\theta,t}^g = -\log(1-\delta),$$
$$\gamma_{\theta,t}^g = \frac{1}{2} - \frac{1}{2}\left((1-\tilde{\lambda})(1-\delta) + \tilde{\lambda}(1-\delta)^{-1}\right),$$
$$V_{\theta,t}^g(\boldsymbol{x}) = \exp(a_{\theta,t}^g\|\boldsymbol{x}\|_1),$$
$$b_{\theta,t}^g = \max_{\boldsymbol{x} \in C_{\theta,t}^g} \exp(a_{\theta,t}^g\|\boldsymbol{x}\|_1)\left(\exp(a_{\theta,t}^g) + 1\right),$$

where the last line holds because $PV_{\theta,t}^g(\boldsymbol{x}) \le V_{\theta,t}^g(\boldsymbol{y})$ for $\boldsymbol{y}$ such that $\|\boldsymbol{y}\|_1 = \|\boldsymbol{x}\|_1 + 1$. □

### F.2 Proof of Proposition 2

*Proof.* In order to show polynomially ergodicity, we will verify (62). We define $V_{\theta,t}^p(\boldsymbol{x}) = \|\boldsymbol{x}\|_1^2$ and $\alpha_{\theta,t}^p = 1/2$, which is equal to $r/(r+1)$ for $r = 1$; $r$ is defined in Assumption 1. For $\boldsymbol{x} = (i,0,1)$ and $i \ge 1$,

$$P_\theta^t V_{\theta,t}^p(i,0,1) = \tilde{\lambda}V_{\theta,t}^p(i,1,1) + \tilde{\theta}_1 V_{\theta,t}^p(i-1,0,1) + \tilde{\theta}_2 V_{\theta,t}^p(i-1,1,0),$$

in which $\tilde{\lambda}$, $\tilde{\theta}_1$, and $\tilde{\theta}_2$ are the normalized rates defined in (83). Thus,

$$P_\theta^t V_{\theta,t}^p(i,0,1) - V_{\theta,t}^p(i,0,1) + \beta_{\theta,t}^p\sqrt{V_{\theta,t}^p(i,0,1)}$$
$$= \tilde{\lambda}(i+2)^2 + (\tilde{\theta}_1 + \tilde{\theta}_2)i^2 - (i+1)^2 + \beta_{\theta,t}^p(i+1)$$
$$= i(4\tilde{\lambda} - 2 + \beta_{\theta,t}^p) + 4\tilde{\lambda} - 1 + \beta_{\theta,t}^p.$$

For $\beta_{\theta,t}^p = 1 - 2\tilde{\lambda}$, the right-hand side of above equation is non-positive for $i \geq \frac{2\tilde{\lambda}}{1-2\tilde{\lambda}}$. For $\boldsymbol{x} = (i,1,0)$ and $i \geq t$,

$$P_\theta^t V_{\theta,t}^p(i,1,0) = \tilde{\lambda} V_{\theta,t}^p(i,1,1) + \tilde{\theta}_1 V_{\theta,t}^p(i-1,0,1) + \tilde{\theta}_2 V_{\theta,t}^p(i-1,1,0).$$

Thus,

$$\begin{aligned}
&P_\theta^t V_{\theta,t}^p(i,1,0) - V_{\theta,t}^p(i,1,0) + \beta_{\theta,t}^p \sqrt{V_{\theta,t}^p(i,1,0)} \\
&= \tilde{\lambda}(i+2)^2 + (\tilde{\theta}_1 + \tilde{\theta}_2)i^2 - (i+1)^2 + \beta_{\theta,t}^p(i+1) \\
&= i(4\tilde{\lambda} - 2 + \beta_{\theta,t}^p) + 4\tilde{\lambda} - 1 + \beta_{\theta,t}^p,
\end{aligned}$$

which is also non-positive under the same conditions as the previous case. For $i \geq 1$ and $\boldsymbol{x} = (i,1,1)$,

$$P_\theta^t V_{\theta,t}^p(i,1,1) = \tilde{\lambda} V_{\theta,t}^p(i+1,1,1) + \tilde{\theta}_1 V_{\theta,t}^p(i,0,1) + \tilde{\theta}_2 V_{\theta,t}^p(i,1,0).$$

Thus,

$$\begin{aligned}
&P_\theta^t V_{\theta,t}^p(i,1,1) - V_{\theta,t}^p(i,1,1) + \beta_{\theta,t}^p \sqrt{V_{\theta,t}^p(i,1,1)} \\
&= \tilde{\lambda}(i+3)^2 + (\tilde{\theta}_1 + \tilde{\theta}_2)(i+1)^2 - (i+2)^2 + \beta_{\theta,t}^p(i+2) \\
&= i(4\tilde{\lambda} - 2 + \beta_{\theta,t}^p) + 8\tilde{\lambda} - 3 + 2\beta_{\theta,t}^p,
\end{aligned}$$

which is non-positive under the same conditions as the first case. Finally, (62) holds for

$$\begin{aligned}
C_{\theta,t}^p &= \{(x_0, x_1, 0) : x_0 < t\} \cup \left\{ (x_0, x_1, x_2) : x_0 < \frac{2\tilde{\lambda}}{1 - 2\tilde{\lambda}}, x_1 + x_2 \geq 1 \right\}, \\
\beta_{\theta,t}^p &= 1 - 2\tilde{\lambda}, \\
\alpha_{\theta,t}^p &= \frac{1}{2}, \\
V_{\theta,t}^p(\boldsymbol{x}) &= \|\boldsymbol{x}\|_1^2, \\
b_{\theta,t}^p &= \max_{\boldsymbol{x} \in C_{\theta,t}^p} (\|\boldsymbol{x}\|_1 + 1)^2),
\end{aligned}$$

where the last line holds because $PV_{\theta,t}^p(\boldsymbol{x}) \leq V_{\theta,t}^p(\boldsymbol{y})$ for $\boldsymbol{y}$ such that $\|\boldsymbol{y}\|_1 = \|\boldsymbol{x}\|_1 + 1$. $\qquad\square$

### F.3 Proof of Proposition 3

*Proof.* To show geometric ergodicity of the chain that follows $\pi_\omega$, we verify (70). Take $a_{\theta,\omega}^g > 0$ and

$$V_{\theta,\omega}^g(\boldsymbol{x}) = \frac{\omega}{\omega + 1} \exp\left( a_{\theta,\omega}^g \frac{x_1 + 1}{\omega} \right) + \frac{1}{\omega + 1} \exp\left( a_{\theta,\omega}^g (x_2 + 1) \right). \tag{86}$$

First, we find $PV_{\theta,\omega}^g(\boldsymbol{x})$ for the function defined above. We have

$$PV_{\theta,\omega}^g(\boldsymbol{x}) = \mathbb{E}_{\boldsymbol{x}}^{\pi_\omega} \left[ \frac{\omega}{\omega + 1} \exp\left( a_{\theta,\omega}^g \frac{X_1(2) + 1}{\omega} \right) \right] + \mathbb{E}_{\boldsymbol{x}}^{\pi_\omega} \left[ \frac{1}{\omega + 1} \exp\left( a_{\theta,\omega}^g (X_2(2) + 1) \right) \right], \tag{87}$$

where $\boldsymbol{X}(2) = (X_1(2), X_2(2))$ is the state of the system at the second arrival, starting from state $\boldsymbol{x}$. To find the above expectations, we first find the corresponding transition probabilities. If the number of departures from server $i$ during a fixed interval with length $t$ is less than the total number of jobs in the queue of that server, the number of departures follows a Poisson distribution with parameter $\theta_i t$. Let $\mathbb{P}\left((x_1, x_2) \to (x_1', \mathcal{X})\right)$ be the probability of transitioning from a system with $x_i$ jobs in server-queue pair $i$ (just after the assignment of the arrival) to a queueing system with $x_1'$ jobs in the first server-queue pair (just before the upcoming arrival). For $1 \leq x_1' \leq x_1$, we have

$$\mathbb{P}\left((x_1, x_2) \to (x_1', \mathcal{X})\right) = \int_0^\infty \lambda \exp(-\lambda t) \frac{(\theta_1 t)^{x_1 - x_1'}}{(x_1 - x_1')!} \exp(-\theta_1 t)\, dt = \frac{\lambda}{\theta_1 + \lambda} \left( \frac{\theta_1}{\theta_1 + \lambda} \right)^{x_1 - x_1'}, \tag{88}$$

and

$$\mathbb{P}\left((x_1, x_2) \rightarrow (0, \mathcal{X})\right) = 1 - \sum_{i=1}^{x_1} \frac{\lambda}{\theta_1 + \lambda} \left(\frac{\theta_1}{\theta_1 + \lambda}\right)^{x_1 - i} = \left(\frac{\theta_1}{\theta_1 + \lambda}\right)^{x_1}. \qquad (89)$$

Assume $1 + x_1 \leq \omega(1 + x_2)$, which results in the new arrival being assigned to the first server. For the first term in (87), we have

$$\mathbb{E}_{\boldsymbol{x}}^{\pi_\omega}\left[\exp\left(a_{\theta,\omega}^g \frac{X_1(2)}{\omega}\right)\right]$$

$$= \sum_{i=0}^{x_1+1} \mathbb{P}\left((x_1 + 1, x_2) \rightarrow (i, \mathcal{X})\right) \exp\left(a_{\theta,\omega}^g \frac{i}{\omega}\right)$$

$$= \left(\frac{\theta_1}{\theta_1 + \lambda}\right)^{x_1+1} + \sum_{i=1}^{x_1+1} \exp\left(a_{\theta,\omega}^g \frac{i}{\omega}\right) \frac{\lambda}{\theta_1 + \lambda} \left(\frac{\theta_1}{\theta_1 + \lambda}\right)^{x_1+1-i}$$

$$= \left(\frac{\theta_1}{\theta_1 + \lambda}\right)^{x_1+1} + \frac{\lambda}{\theta_1 + \lambda} \exp\left(a_{\theta,\omega}^g \frac{x_1 + 1}{\omega}\right) \frac{1 - \exp\left(-a_{\theta,\omega}^g \frac{x_1+1}{\omega}\right)\left(\frac{\theta_1}{\theta_1+\lambda}\right)^{x_1+1}}{1 - \exp\left(-\frac{a_{\theta,\omega}^g}{\omega}\right) \frac{\theta_1}{\theta_1+\lambda}},$$

$$< \left(\frac{\theta_1}{\theta_1 + \lambda}\right)^{x_1+1} + \frac{\lambda}{\theta_1 + \lambda} \exp\left(a_{\theta,\omega}^g \frac{x_1 + 1}{\omega}\right) \frac{1}{1 - \exp\left(-\frac{a_{\theta,\omega}^g}{\omega}\right) \frac{\theta_1}{\theta_1+\lambda}}. \qquad (90)$$

Similarly, for the second term in (87), we have

$$\mathbb{E}_{\boldsymbol{x}}^{\pi_\omega}\left[\exp\left(a_{\theta,\omega}^g X_2(2)\right)\right] \leq \left(\frac{\theta_2}{\theta_2 + \lambda}\right)^{x_2} + \frac{\lambda}{\theta_2 + \lambda} \exp\left(a_{\theta,\omega}^g x_2\right) \frac{1}{1 - \exp\left(-a_{\theta,\omega}^g\right) \frac{\theta_2}{\theta_2+\lambda}}. \qquad (91)$$

To satisfy (70), for some $0 < \gamma_{\theta,\omega}^g < 1$ and all but finitely many $\boldsymbol{x}$, the following should hold,

$$PV_{\theta,\omega}^g(\boldsymbol{x}) \leq \gamma_{\theta,\omega}^g V_{\theta,\omega}^g(\boldsymbol{x}),$$

or from (86) and (87),

$$\mathbb{E}_{\boldsymbol{x}}^{\pi_\omega}\left[\omega \exp\left(a_{\theta,\omega}^g \frac{X_1(2)+1}{\omega}\right)\right] + \mathbb{E}_{\boldsymbol{x}}^{\pi_\omega}\left[\exp\left(a_{\theta,\omega}^g (X_2(2)+1)\right)\right]$$

$$\leq \gamma_{\theta,\omega}^g \left(\omega \exp\left(a_{\theta,\omega}^g \frac{x_1+1}{\omega}\right) + \exp\left(a_{\theta,\omega}^g (x_2+1)\right)\right).$$

Notice that

$$\omega\left(\frac{\theta_1}{\theta_1 + \lambda}\right)^{x_1+1} + \left(\frac{\theta_2}{\theta_2 + \lambda}\right)^{x_2} \leq c_R R + 1.$$

From (90) and (91), it suffices to have

$$(c_R R + 1)\exp(c_R R a_{\theta,\omega}^g) + \frac{\omega \frac{\lambda}{\theta_1+\lambda} \exp\left(a_{\theta,\omega}^g \frac{x_1+2}{\omega}\right)}{1 - \exp\left(-\frac{a_{\theta,\omega}^g}{\omega}\right) \frac{\theta_1}{\theta_1+\lambda}} + \frac{\frac{\lambda}{\theta_2+\lambda} \exp\left(a_{\theta,\omega}^g (x_2+1)\right)}{1 - \exp\left(-a_{\theta,\omega}^g\right) \frac{\theta_2}{\theta_2+\lambda}}$$

$$\leq \gamma_{\theta,\omega}^g \left(\omega \exp\left(a_{\theta,\omega}^g \frac{x_1+1}{\omega}\right) + \exp\left(a_{\theta,\omega}^g (x_2+1)\right)\right). \qquad (92)$$

Define

$$\zeta_{1,\theta,\omega} = \frac{\frac{\lambda}{\theta_1+\lambda}}{1 - \exp\left(-\frac{a_{\theta,\omega}^g}{\omega}\right) \frac{\theta_1}{\theta_1+\lambda}}, \qquad \zeta_{2,\theta,\omega} = \frac{\frac{\lambda}{\theta_2+\lambda}}{1 - \exp\left(-a_{\theta,\omega}^g\right) \frac{\theta_2}{\theta_2+\lambda}}.$$

Simplifying (92), we need the following to hold

$$(c_R R + 1)\exp(c_R R a_{\theta,\omega}^g) + \omega \exp\left(a_{\theta,\omega}^g \frac{x_1+1}{\omega}\right)\left(\zeta_{1,\theta,\omega} \exp\left(\frac{a_{\theta,\omega}^g}{\omega}\right) - \gamma_{\theta,\omega}^g\right)$$

$$+ \exp\left(a_{\theta,\omega}^g (x_2+1)\right)\left(\zeta_{2,\theta,\omega} - \gamma_{\theta,\omega}^g\right) \leq 0. \qquad (93)$$

As $\zeta_{i,\theta,\omega} < 1$, there exists $\gamma^g_{\theta,\omega}$ such that

$$\zeta_{2,\theta,\omega} < \gamma^g_{\theta,\omega} < 1.$$

From the assumption $1 + x_1 \leq \omega(1 + x_2)$ and the above equation, (93) can be further simplified as

$$(c_R R + 1)\exp(c_R R a^g_{\theta,\omega}) + \exp\left(a^g_{\theta,\omega}\frac{x_1 + 1}{\omega}\right)\left(\omega\zeta_{1,\theta,\omega}\exp\left(\frac{a^g_{\theta,\omega}}{\omega}\right) + \zeta_{2,\theta,\omega} - (\omega+1)\gamma^g_{\theta,\omega}\right) \leq 0. \tag{94}$$

For the above to hold outside a finite set, we need to have

$$\frac{\zeta_{1,\theta,\omega}\omega}{1+\omega}\exp\left(\frac{a^g_{\theta,\omega}}{\omega}\right) + \frac{\zeta_{2,\theta,\omega}}{1+\omega} < \gamma^g_{\theta,\omega}. \tag{95}$$

Define

$$\zeta_3 = \frac{1}{1+\delta}, \quad \zeta_4 = \frac{1 - 0.5\delta}{1 - \delta}. \tag{96}$$

Note that $\zeta_3 < 1$ and $\zeta_4 > 1$. Defining function $f(y) := 1 + \zeta_4 y - \exp(y)$, we note that for $y \leq \log\zeta_4$, $f(y) > 0$, where $\log\zeta_4$ is the maximizer of $f(y)$. Similarly, taking $g(y) := 1 - \zeta_3 y - \exp(-y)$, for $y \leq -\log\zeta_3$, $g(y) > 0$, where $-\log\zeta_3$ is the maximizer of $g(y)$. Thus, we conclude that for $a^g_{\theta,\omega} \leq \min\left(-\omega\log\zeta_3, -\log\zeta_3, \omega\log\zeta_4\right)$,

$$\exp(-y) \leq 1 - \zeta_3 y \quad \text{holds for} \quad y \leq \max\left(\frac{a^g_{\theta,\omega}}{\omega}, a^g_{\theta,\omega}\right), \tag{97}$$

$$\exp(y) \leq 1 + \zeta_4 y \quad \text{holds for} \quad y \leq \frac{a^g_{\theta,\omega}}{\omega}. \tag{98}$$

To guarantee the existence of $0 < \gamma^g_{\theta,\omega} < 1$ that satisfies (95), we need to ensure the left-hand side of (95) is strictly less than 1. Using the bounds found in (97) and (98) and the definition of $\zeta_{1,\theta,\omega}$ and $\zeta_{2,\theta,\omega}$, we simplify (95) to get

$$\frac{\frac{\lambda}{1+\omega}\left(\omega + a^g_{\theta,\omega}\zeta_4\right)}{\lambda + \frac{\theta_1 a^g_{\theta,\omega}\zeta_3}{\omega}} + \frac{\frac{\lambda}{1+\omega}}{\lambda + \theta_2 a^g_{\theta,\omega}\zeta_3} < 1,$$

which is equivalent to

$$a^g_{\theta,\omega}\zeta_3\theta_2\left(\lambda\zeta_4 - \frac{\zeta_3\theta_1(1+\omega)}{\omega}\right) < \lambda\zeta_3(\theta_1 + \theta_2) - \lambda^2\zeta_4. \tag{99}$$

To make sure there exists $a^g_{\theta,\omega} > 0$ that satisfies (99), the right-hand side of (99) needs to be positive, which follows as below:

$$\lambda\zeta_3(\theta_1 + \theta_2) - \lambda^2\zeta_4 = \lambda\left(\frac{\theta_1 + \theta_2}{1+\delta} - \lambda\frac{1 - 0.5\delta}{1 - \delta}\right)$$

$$= \lambda(\theta_1 + \theta_2 + \lambda)\left(\frac{1 - \tilde{\lambda}}{1+\delta} - \tilde{\lambda}\frac{1 - 0.5\delta}{1 - \delta}\right)$$

$$= \lambda(\theta_1 + \theta_2 + \lambda)\left(\frac{1}{1+\delta} - \tilde{\lambda}\left(\frac{1}{1+\delta} + \frac{1 - 0.5\delta}{1 - \delta}\right)\right)$$

$$\geq \lambda(\theta_1 + \theta_2 + \lambda)\left(\frac{1}{1+\delta} - \frac{1 - \delta}{2}\left(\frac{1}{1+\delta} + \frac{1 - 0.5\delta}{1 - \delta}\right)\right)$$

$$= \frac{\delta}{4}\lambda(\theta_1 + \theta_2 + \lambda) \tag{100}$$

where $\tilde{\lambda}$, $\tilde{\theta}_1$, and $\tilde{\theta}_2$ are the normalized rates defined in (83) and we have used the stability condition $\tilde{\lambda} \leq 0.5 - 0.5\delta$. We further simplify the left-hand side of (99) as

$$\zeta_3\theta_2\left(\lambda\zeta_4 - \frac{\zeta_3\theta_1(1+\omega)}{\omega}\right) < \theta_2\lambda\zeta_3\zeta_4 < \frac{1 - 0.5\delta}{1 - \delta^2}(\theta_1 + \theta_2 + \lambda)\lambda.$$

From the above equation and (100), $a_{\theta,\omega}^g$ needs to satisfy

$$a_{\theta,\omega}^g \le \frac{\delta(1 - \delta^2)}{8(1 - 0.5\delta)}.$$

Finally, we take $a_{\theta,\omega}^g$ as

$$a_{\theta,\omega}^g = \min\left(-\omega \log \zeta_3, -\log \zeta_3, \omega \log \zeta_4, \frac{\delta(1 - \delta^2)}{8(1 - 0.5\delta)}\right).$$

After finding an appropriate $a_{\theta,\omega}^g$, we can choose $0 < \gamma_{\theta,\omega}^g < 1$ such that (95) holds or

$$\gamma_{\theta,\omega}^g \ge \frac{1}{2}\left(1 + \frac{\zeta_{1,\theta,\omega}\omega}{1 + \omega} \exp\left(\frac{a_{\theta,\omega}^g}{\omega}\right) + \frac{\zeta_{2,\theta,\omega}}{1 + \omega}\right).$$

Moreover, from (94) a lower bound $x_{1,\theta,\omega}^{g_1}$ for $x_1$ is derived; In other words,(94) holds for $x_1 > x_{1,\theta,\omega}^{g_1}$. From (93), we can find the corresponding $x_{2,\theta,\omega}^{g_1}$ and take $\boldsymbol{x}_{\theta,\omega}^{g_1} = (x_{1,\theta,\omega}^{g_1}, x_{2,\theta,\omega}^{g_1})$. By repeating the same arguments when $1 + x_1 < \omega(1 + x_2)$, we finally conclude that

$$\Delta V_{\theta,\omega}^g(\boldsymbol{x}) \le -\left(1 - \gamma_{\theta,\omega}^g\right) V_{\theta,\omega}^g(\boldsymbol{x}) + b_{\theta,\omega}^g \mathbb{I}_{C_{\theta,\omega}^g}(\boldsymbol{x}), \quad \boldsymbol{x} \in \mathcal{X},$$

for

$$V_{\theta,\omega}^g(\boldsymbol{x}) = \frac{\omega}{\omega + 1} \exp\left(a_{\theta,\omega}^g \frac{x_1 + 1}{\omega}\right) + \frac{1}{\omega + 1} \exp\left(a_{\theta,\omega}^g(x_2 + 1)\right),$$

$$a_{\theta,\omega}^g = \min\left(\omega \log(1 + \delta), \log(1 + \delta), \omega \log \frac{1 - 0.5\delta}{1 - \delta}, \log \frac{1 - 0.5\delta}{1 - \delta}, \frac{\delta(1 - \delta^2)}{4c_R R(1 - 0.5\delta)}\right),$$

$$C_{\theta,\omega}^g = \{(x_1, x_2) \in \mathcal{X} : x_i \le \max\left(x_{i,\theta,\omega}^{g_j}, 0\right), i, j = 1, 2\},$$

$$\gamma_{\theta,\omega}^g = \frac{1}{2} + \frac{1}{2}\max\left(\zeta_{1,\theta,\omega}, \zeta_{2,\theta,\omega}, \frac{\zeta_{1,\theta,\omega}\omega}{1 + \omega}\exp\left(\frac{a_{\theta,\omega}^g}{\omega}\right) + \frac{\zeta_{2,\theta,\omega}}{1 + \omega}, \frac{\zeta_{1,\theta,\omega}\omega}{1 + \omega} + \frac{\zeta_{2,\theta,\omega}}{1 + \omega}\exp\left(a_{\theta,\omega}^g\right)\right),$$

$$b_{\theta,\omega}^g = \max_{\boldsymbol{x} \in C_{\theta,\omega}^g}\left(\frac{2\omega}{\omega + 1}\exp\left(a_{\theta,\omega}^g \frac{x_1 + 2}{\omega}\right) + \frac{2}{\omega + 1}\exp\left(a_{\theta,\omega}^g(x_2 + 2)\right)\right),$$

$$\zeta_{1,\theta,\omega} = \frac{\frac{\lambda}{\theta_1 + \lambda}}{1 - \exp\left(-\frac{a_{\theta,\omega}^g}{\omega}\right)\frac{\theta_1}{\theta_1 + \lambda}},$$

$$\zeta_{2,\theta,\omega} = \frac{\frac{\lambda}{\theta_2 + \lambda}}{1 - \exp\left(-a_{\theta,\omega}^g\right)\frac{\theta_2}{\theta_2 + \lambda}},$$

$$x_{1,\theta,\omega}^{g_1} = \frac{\omega}{a_{\theta,\omega}^g}\log\frac{(c_R R + 1)\exp(c_R R a_{\theta,\omega}^g)}{(\omega + 1)\gamma_{\theta,\omega}^g - \omega\zeta_{1,\theta,\omega}\exp\left(\frac{a_{\theta,\omega}^g}{\omega}\right) - \zeta_{2,\theta,\omega}},$$

$$x_{2,\theta,\omega}^{g_1} = \frac{1}{a_{\theta,\omega}^g}\log\frac{(c_R R + 1)\exp(c_R R a_{\theta,\omega}^g) + \omega\exp\left(a_{\theta,\omega}^g \frac{x_{1,\theta,\omega}^{g_1} + 1}{\omega}\right)\left(\zeta_{1,\theta,\omega}\exp\left(\frac{a_{\theta,\omega}^g}{\omega}\right) - \gamma_{\theta,\omega}^g\right)}{\gamma_{\theta,\omega}^g - \zeta_{2,\theta,\omega}},$$

$$x_{2,\theta,\omega}^{g_2} = \frac{1}{a_{\theta,\omega}^g}\log\frac{(c_R R + 1)\exp(c_R R a_{\theta,\omega}^g)}{(\omega + 1)\gamma_{\theta,\omega}^g - \omega\zeta_{1,\theta,\omega} - \zeta_{2,\theta,\omega}\exp\left(a_{\theta,\omega}^g\right)},$$

$$x_{1,\theta,\omega}^{g_2} = \frac{\omega}{a_{\theta,\omega}^g}\log\frac{(c_R R + 1)\exp(c_R R a_{\theta,\omega}^g) + \exp\left(a_{\theta,\omega}^g(x_{2,\theta,\omega}^{g_2} + 1)\right)\left(\zeta_{2,\theta,\omega}\exp\left(a_{\theta,\omega}^g\right) - \gamma_{\theta,\omega}^g\right)}{\omega\left(\gamma_{\theta,\omega}^g - \zeta_{1,\theta,\omega}\right)}.$$

$\square$

### F.4 Proof of Proposition 4

*Proof.* Define $V^p_{\theta,\omega}(x) = \frac{x_1^2}{\omega} + x_2^2$, and $\alpha^p_{\theta,\omega} = 1/2$. Assume that $x_1 = 0$ and $x_2 > (1-\omega)/\omega$; which means the new job will be assigned to the first server. The transition probabilities of the discrete-time chain sampled at Poisson arrivals is given in (88) and (89), and we calculate $PV^p_{\theta,\omega}(x)$ as

$$PV^p_{\theta,\omega}(x) = \frac{\lambda}{\omega(\lambda+\theta_1)} + \sum_{i=1}^{x_2} i^2 \frac{\lambda}{\lambda+\theta_2}\left(\frac{\theta_2}{\theta_2+\lambda}\right)^{x_2-i} < c_R R + \sum_{i=1}^{x_2} i^2 \frac{\lambda}{\lambda+\theta_2}\left(\frac{\theta_2}{\theta_2+\lambda}\right)^{x_2-i}.$$
(101)

We define $d_i := \theta_i/(\theta_i+\lambda)$ for $i = 1,2$ and

$$\sum_{i=1}^{x_2} i^2 \frac{\lambda}{\lambda+\theta_2}\left(\frac{\theta_2}{\theta_2+\lambda}\right)^{x_2-i}$$

$$= \frac{1}{(1-d_2)^2}\left(-d_2^{x_2}\left(d_2+d_2^2\right) + d_2^2\left(x_2^2+2x_2+1\right) + d_2\left(-2x_2^2-2x_2+1\right) + x_2^2\right)$$

$$= \frac{1}{(1-d_2)^2}\left((1-d_2^{x_2})\left(d_2+d_2^2\right) + x_2^2\left(d_2^2-2d_2+1\right) + x_2\left(2d_2^2-2d_2\right)\right)$$

$$= x_2^2 - \frac{2d_2}{1-d_2}x_2 + \frac{(1-d_2^{x_2})\left(d_2+d_2^2\right)}{(1-d_2)^2}.$$
(102)

From (101),

$$PV^p_{\theta,\omega}(x) - V^p_{\theta,\omega}(x) + \beta^p_{\theta,\omega}x_2 < \left(-\frac{2d_2}{1-d_2}+\beta^p_{\theta,\omega}\right)x_2 + \frac{(1-d_2^{x_2})\left(d_2+d_2^2\right)}{(1-d_2)^2} + c_R R.$$

Outside a finite set, we need the above equation to be non-positive; which is equivalent to

$$\left(-2 + \beta^p_{\theta,\omega}\frac{1-d_2}{d_2}\right)x_2 + \frac{(1-d_2^{x_2})(1+d_2)}{1-d_2} + c_R R\frac{1-d_2}{d_2} \leq 0.$$

As $d_2 < 1$,

$$\frac{1-d_2^y}{1-d_2} = 1 + d_2 + \ldots + d_2^{y-1} \leq y \qquad \text{for } y \geq 1.$$
(103)

Thus,

$$\left(-2 + \beta^p_{\theta,\omega}\frac{1-d_2}{d_2}\right)x_2 + \frac{(1-d_2^{x_2})(1+d_2)}{1-d_2} + c_R R\frac{1-d_2}{d_2}$$

$$\leq \left(d_2 - 1 + \beta^p_{\theta,\omega}\frac{1-d_2}{d_2}\right)x_2 + c_R R\frac{1-d_2}{d_2}.$$

By taking $\beta^p_{\theta,\omega} \leq d_2/2$, it suffices for the following to be non-positive,

$$-\frac{1-d_2}{2}x_2 + c_R R\frac{1-d_2}{d_2} \leq 0,$$

which holds for $x_2 \geq 2c_R R/d_2$. Thus, for $x_1 = 0$ and $x_2 \geq \max\left(2c_R R(\lambda+\theta_2)/\theta_2, (1-\omega)/\omega\right) = 2c_R R(\lambda+\theta_2)/\theta_2$, (76) holds. The case of $x_2 = 0$ and non-zero $x_1$ follows same arguments and (76) holds for $\beta^p_{\theta,\omega} \leq d_1/2\sqrt{\omega}$, $x_2 = 0$, and $x_1 \geq \max\left(2c_R R(\lambda+\theta_1)/\theta_1, \omega-1\right) = 2c_R R(\lambda+\theta_1)/\theta_1$. We now consider the case of $x_1, x_2 > 0$ and $x_1 + 1 \leq \omega(x_2+1)$, and note that

$$\sqrt{V^p_{\theta,\omega}(x)} = \sqrt{\frac{x_1^2}{\omega} + x_2^2} \leq \sqrt{\frac{(x_1+1)^2}{\omega} + (x_2+1)^2} \leq \sqrt{\omega+1}(x_2+1).$$

Hence, it suffices to find finite set $C^p_{\theta,\omega}$, constants $b^p_{\theta,\omega}$ and $\beta^p_{\theta,\omega} > 0$, such that the following holds for $V^p_{\theta,\omega}(x) = \frac{x_1^2}{\omega} + x_2^2$,

$$\Delta V^p_{\theta,\omega}(x) \leq -\sqrt{\omega+1}\beta^p_{\theta,\omega}(x_2+1) + b^p_{\theta,\omega}\mathbb{I}_{C^p_{\theta,\omega}}(x).$$

As $x_1 + 1 \leq \omega(x_2 + 1)$, the new arrival is assigned to the first queue and we find $\Delta V^p_{\theta,\omega}(\boldsymbol{x}) + \sqrt{\omega + 1}\beta^p_{\theta,\omega}(x_2 + 1)$ using the same calculations as (102).

$$
\begin{aligned}
&\Delta V^p_{\theta,\omega}(\boldsymbol{x}) + \sqrt{\omega + 1}\beta^p_{\theta,\omega}(x_2 + 1) \\
&= \frac{1}{\omega}\left((x_1 + 1)^2 - \frac{2d_1}{1 - d_1}(x_1 + 1) + \frac{\left(1 - d_1^{x_1+1}\right)\left(d_1 + d_1^2\right)}{(1 - d_1)^2} - x_1^2\right) \\
&\quad - \frac{2d_2}{1 - d_2}x_2 + \frac{\left(1 - d_2^{x_2}\right)\left(d_2 + d_2^2\right)}{(1 - d_2)^2} + \sqrt{\omega + 1}\beta^p_{\theta,\omega}(x_2 + 1) \\
&= \frac{x_1}{\omega}\left(2 - \frac{2d_1}{1 - d_1}\right) + \frac{1 - 3d_1}{\omega(1 - d_1)} + \frac{\left(1 - d_1^{x_1+1}\right)\left(d_1 + d_1^2\right)}{\omega(1 - d_1)^2} \quad\quad (104) \\
&\quad + (x_2 + 1)\left(-\frac{2d_2}{1 - d_2} + \sqrt{\omega + 1}\beta^p_{\theta,\omega}\right) + \frac{2d_2}{1 - d_2} + \frac{\left(1 - d_2^{x_2}\right)\left(d_2 + d_2^2\right)}{(1 - d_2)^2}. \quad (105)
\end{aligned}
$$

We next consider two different cases based on the value of $d_1$ and analyze them separately.

**One.** $0.8 \leq d_1 < 1$ : We first notice that the coefficient of $x_1$ in (104) is negative, as $d_1 > 1/2$. For $x_1 \geq 1$, (104) is equal to

$$
\begin{aligned}
&\frac{1}{\omega(1 - d_1)}\left((2 - 4d_1)x_1 + 1 - 3d_1 + (d_1 + d_1^2)\sum_{i=0}^{x_1} d_1^i\right) \\
&= \frac{1}{\omega(1 - d_1)}\left((2 - 4d_1)(x_1 - 1) + d_1^3(1 + d_1)\sum_{i=0}^{x_1-2} d_1^i + d_1(1 + d_1)^2 + 3 - 7d_1\right) \\
&\leq \frac{1}{\omega(1 - d_1)}\left((2 - 4d_1)(x_1 - 1) + d_1^3(1 + d_1)(x_1 - 1) + d_1(1 + d_1)^2 + 3 - 7d_1\right) \\
&= \frac{1}{\omega(1 - d_1)}\left((d_1^4 + d_1^3 - 4d_1 + 2)(x_1 - 1) + d_1(1 + d_1)^2 + 3 - 7d_1\right) \\
&= \frac{-d_1^3 - 2d_1^2 - 2d_1 + 2}{\omega}(x_1 - 1) + \frac{-d_1^2 - 3d_1 + 3}{\omega} \\
&< 0,
\end{aligned}
$$

where the third line follows from (103), and the last line from the fact that when $0.8 \leq d_1 < 1$, both terms $-d_1^3 - 2d_1^2 - 2d_1 + 2$ and $-d_1^2 - 3d_1 + 3$ are negative. Next, we notice that (105) is equal to

$$
\begin{aligned}
&x_2\left(-\frac{2d_2}{1 - d_2} + \sqrt{\omega + 1}\beta^p_{\theta,\omega}\right) + \sqrt{\omega + 1}\beta^p_{\theta,\omega} + \frac{\left(1 - d_2^{x_2}\right)\left(d_2 + d_2^2\right)}{(1 - d_2)^2} \\
&\leq x_2\left(-\frac{2d_2}{1 - d_2} + \sqrt{\omega + 1}\beta^p_{\theta,\omega}\right) + \frac{d_2 + d_2^2}{1 - d_2}x_2 + \sqrt{\omega + 1}\beta^p_{\theta,\omega} \\
&= x_2\left(-\frac{2d_2}{1 - d_2} + \frac{d_2 + d_2^2}{1 - d_2} + \sqrt{\omega + 1}\beta^p_{\theta,\omega}\right) + \sqrt{\omega + 1}\beta^p_{\theta,\omega} \\
&= x_2\left(-d_2 + \sqrt{\omega + 1}\beta^p_{\theta,\omega}\right) + \sqrt{\omega + 1}\beta^p_{\theta,\omega},
\end{aligned}
$$

where the second line follows from (103). Taking $\beta^p_{\theta,\omega} \leq d_2/2\sqrt{\omega + 1}$, we get

$$
x_2\left(-\frac{2d_2}{1 - d_2} + \sqrt{\omega + 1}\beta^p_{\theta,\omega}\right) + \sqrt{\omega + 1}\beta^p_{\theta,\omega} + \frac{\left(1 - d_2^{x_2}\right)\left(d_2 + d_2^2\right)}{(1 - d_2)^2} \leq -\frac{d_2}{2}x_2 + \frac{d_2}{2},
$$

which is non-positive for $x_2 \geq 1$. Finally, when $0.8 \leq d_1 < 1$, $x_1, x_2 > 0$, and $x_1 + 1 \leq \omega(x_2 + 1)$, (76) holds for $\beta^p_{\theta,\omega} \leq d_2/2\sqrt{\omega + 1}$.

**Two.** $d_1 < 0.8$ : Taking $\beta_{\theta,\omega}^p \leq \frac{d_2}{\sqrt{\omega+1}(1-d_2)}$, we note that the coefficient of $x_2$ in (105) is negative. Thus, from $x_1 + 1 \leq \omega(x_2 + 1)$, (104) and (105),

$$
\begin{aligned}
&\Delta V_{\theta,\omega}^p(\boldsymbol{x}) + \sqrt{\omega+1}\beta_{\theta,\omega}^p(x_2+1) \\
&\leq \frac{x_1+1}{\omega}\left(2 - \frac{2d_1}{1-d_1}\right) - \frac{1}{\omega} + \frac{\left(1-d_1^{x_1+1}\right)\left(d_1+d_1^2\right)}{\omega(1-d_1)^2} \\
&\quad + \frac{x_1+1}{\omega}\left(-\frac{2d_2}{1-d_2} + \sqrt{\omega+1}\beta_{\theta,\omega}^p\right) + \frac{2d_2}{1-d_2} + \frac{\left(1-d_2^{x_2}\right)\left(d_2+d_2^2\right)}{(1-d_2)^2} \\
&< \frac{x_1+1}{\omega}\left(2 - \frac{2d_1}{1-d_1} - \frac{2d_2}{1-d_2} + \sqrt{\omega+1}\beta_{\theta,\omega}^p\right) + \frac{2d_2}{1-d_2} + \frac{d_1+d_1^2}{\omega(1-d_1)^2} + \frac{d_2+d_2^2}{(1-d_2)^2}.
\end{aligned}
\tag{106}
$$

As $d_i = \tilde{\theta}_i/(\tilde{\theta}_i + \tilde{\lambda})$ in terms of the normalized rates, we get

$$
2 - \frac{2d_1}{1-d_1} - \frac{2d_2}{1-d_2} = 2 - \frac{2\tilde{\theta}_1}{\tilde{\lambda}} - \frac{2\tilde{\theta}_1}{\tilde{\lambda}} = \frac{-2(\tilde{\theta}_1 + \tilde{\theta}_2 - \tilde{\lambda})}{\tilde{\lambda}},
$$

which is negative from the stability condition. For $\beta_{\theta,\omega}^p \leq \frac{\tilde{\theta}_1 + \tilde{\theta}_2 - \tilde{\lambda}}{\tilde{\lambda}\sqrt{\omega+1}}$, from (106) we get

$$
\begin{aligned}
&\Delta V_{\theta,\omega}^p(\boldsymbol{x}) + \sqrt{\omega+1}\beta_{\theta,\omega}^p(x_2+1) \\
&< \frac{-(\tilde{\theta}_1 + \tilde{\theta}_2 - \tilde{\lambda})}{\omega\tilde{\lambda}}(x_1+1) + \frac{2d_2}{1-d_2} + \frac{d_1+d_1^2}{\omega(1-d_1)^2} + \frac{d_2+d_2^2}{(1-d_2)^2} \\
&= \frac{-(\tilde{\theta}_1 + \tilde{\theta}_2 - \tilde{\lambda})}{\omega\tilde{\lambda}}(x_1+1) + \frac{2\tilde{\theta}_2}{\tilde{\lambda}} + \frac{\tilde{\theta}_1(2\tilde{\theta}_1 + \tilde{\lambda})}{\omega\tilde{\lambda}^2} + \frac{\tilde{\theta}_2(2\tilde{\theta}_2 + \tilde{\lambda})}{\tilde{\lambda}^2},
\end{aligned}
$$

which is non-positive for

$$
x_1 + 1 \geq \frac{\tilde{\theta}_1(2\tilde{\theta}_1 + \tilde{\lambda}) + \omega\tilde{\theta}_2(2\tilde{\theta}_2 + 3\tilde{\lambda})}{\tilde{\lambda}(\tilde{\theta}_1 + \tilde{\theta}_2 - \tilde{\lambda})}.
$$

As $d_1 < 0.8$, we can see that $\tilde{\lambda} > \tilde{\theta}_1/4$; thus,

$$
\frac{\tilde{\theta}_1(2\tilde{\theta}_1 + \tilde{\lambda}) + \omega\tilde{\theta}_2(2\tilde{\theta}_2 + 3\tilde{\lambda})}{\tilde{\lambda}(\tilde{\theta}_1 + \tilde{\theta}_2 - \tilde{\lambda})} < \frac{4\tilde{\theta}_1(2\tilde{\theta}_1 + \tilde{\lambda}) + 4\omega\tilde{\theta}_2(2\tilde{\theta}_2 + 3\tilde{\lambda})}{\tilde{\theta}_1(\tilde{\theta}_1 + \tilde{\theta}_2 - \tilde{\lambda})} < \frac{4c_R R(1 + 2\tilde{\lambda})}{\delta} \leq 4c_R R,
$$

where we have used the fact that $\tilde{\theta}_1 \geq \tilde{\theta}_2$, $\omega \leq c_R R$, $\tilde{\theta}_1 + \tilde{\theta}_2 - \tilde{\lambda} \geq \delta$, and $\tilde{\lambda} \leq 0.5 - 0.5\delta$ and it suffices for $x_1$ to be greater than or equal to $4c_R R$. For $x_1 < 4c_R R$, (104) can be upper bounded as

$$
\frac{8c_R R}{\omega} + \frac{1 - 3d_1}{\omega(1-d_1)} + \frac{d_1+d_1^2}{\omega(1-d_1)^2} \leq \frac{8c_R R}{\omega} + \frac{2}{\omega(1-d_1)^2} < \frac{8c_R R + 50}{\omega},
$$

where in the last inequality we have used $d_1 < 0.8$. From (105) and taking $\beta_{\theta,\omega}^p \leq d_2/2\sqrt{\omega+1}$,

$$
\begin{aligned}
&\Delta V_{\theta,\omega}^p(\boldsymbol{x}) + \sqrt{\omega+1}\beta_{\theta,\omega}^p(x_2+1) \\
&\leq \frac{8c_R R + 50}{\omega} + \left(-\frac{2d_2}{1-d_2} + \frac{d_2}{2}\right)(x_2+1) + \frac{2d_2}{1-d_2} + \frac{\left(1-d_2^{x_2}\right)\left(d_2+d_2^2\right)}{(1-d_2)^2} \\
&\leq \left(-\frac{2d_2}{1-d_2} + \frac{d_2}{2} + \frac{d_2+d_2^2}{1-d_2}\right)x_2 + \frac{d_2}{2} + \frac{8c_R R + 50}{\omega} \\
&= -\frac{d_2}{2}x_2 + \frac{d_2}{2} + \frac{8c_R R + 50}{\omega},
\end{aligned}
$$

which is negative for

$$
x_2 \geq 1 + \frac{16c_R R + 100}{\omega d_2}.
$$

Finally, when $x_1 + 1 \leq \omega(x_2 + 1)$ and $x_1, x_2 > 0$, (76) holds for $\beta_{\theta,\omega}^p \leq \frac{1}{\sqrt{\omega+1}} \min\left(\frac{\tilde{\theta}_2}{2(\tilde{\theta}_2+\tilde{\lambda})}, \tilde{\theta}_1 + \tilde{\theta}_2 - \tilde{\lambda}\right)$, $x_1 \geq 4c_R R$, and $x_2 \geq 1 + \frac{16c_R R + 100}{\omega d_2}$. Repeating the same arguments when $x_1, x_2 > 0$ and $x_1 + 1 > \omega(x_2 + 1)$, (76) holds for $\beta_{\theta,\omega}^p \leq \frac{1}{\sqrt{\omega+1}} \min\left(\frac{\tilde{\theta}_1}{2(\tilde{\theta}_1+\tilde{\lambda})}, \tilde{\theta}_1 + \tilde{\theta}_2 - \tilde{\lambda}\right)$, $x_1 \geq 1 + \frac{\omega(16c_R R^2 + 100)}{d_1}$, and $x_2 \geq 4c_R R^2$. Finally, (76) holds with

$$V_{\theta,\omega}^p(\boldsymbol{x}) = \frac{x_1^2}{\omega} + x_2^2,$$

$$C_{\theta,\omega}^p = \left\{ (x_1, x_2) \in \mathcal{X} : x_i \leq \left(16c_R^2 R^{3-i} + 101c_R R\right) \frac{\lambda + \theta_i}{\theta_i}, i = 1, 2 \right\},$$

$$\beta_{\theta,\omega}^p = \min\left(\frac{\tilde{\theta}_2}{2(\tilde{\theta}_2+\tilde{\lambda})\sqrt{\omega+1}}, \frac{\tilde{\theta}_1 + \tilde{\theta}_2 - \tilde{\lambda}}{\sqrt{\omega+1}}, \frac{\tilde{\theta}_2}{2(\tilde{\theta}_2+\tilde{\lambda})}, \frac{\tilde{\theta}_1}{2(\tilde{\theta}_1+\tilde{\lambda})\sqrt{\omega}}\right),$$

$$b_{\theta,\omega}^p = (\beta_{\theta,\omega}^p + 1) \max_{\boldsymbol{x} \in C_{\theta,\omega}^p} \left(\frac{(x_1+1)^2}{\omega} + (x_2+1)^2\right),$$

$$\alpha_{\theta,\omega}^p = \frac{1}{2},$$

where the fourth line holds since $PV_{\theta,\omega}^p(\boldsymbol{x}) \leq V_{\theta,\omega}^p(\boldsymbol{y})$ for $\boldsymbol{y} = (y_1, y_2)$ such that $y_i = x_i + 1$ for $i = 1, 2$. $\qquad \square$

# G  Numerical results

## G.1  Comparison of Algorithm 1 with other learning algorithms

We first note that due to the countably infinite state-space setting of our problem, we are unable to directly compare our algorithm to other learning algorithms proposed in the literature. One potential candidate algorithm uses the reward biased maximum likelihood estimation (RBMLE) [33, 34, 11, 42], which estimates the unknown model parameter with the likelihood perturbed a vanishing bias towards parameters with a larger long-term average reward (i.e., optimal value). This scheme also uses the principle of "optimism in the face of uncertainty" in how it perturbs the maximum likelihood estimate. The naive version of the RMBLE algorithm does not apply to our examples due the following key assumption: over all parameters (and the control policies used for them), the transition probabilities are assumed to be mutually absolutely continuous; this is critical for the proofs and also allows the use of log-likelihood functions for computations. Similarly, naive use of the algorithms in [36] and [24] is not possible, again due to a similar absolutely continuity assumption which is critical for the proofs. Our posterior computations avoid such issues as the true parameter always has non-zero mass during the execution of the algorithm: episode $k$ always starts in state $0^d$ which is positive recurrent for the Markov chain with true parameter $\boldsymbol{\theta}^*$ and policy used $\pi_{\theta_k}^*$. The RBMLE algorithm has yet another issue in that it requires knowledge of the optimal value function, and hence, for our examples, it may only apply to Model 1 for which the value function is known analytically. Finally, whereas we do get to observe inter-arrival times for both model, we never directly observe completed service times owing to the sampling employed, and this precludes the direct use of Upper-Confidence-Bound based parameter estimation followed by certainty equivalent control algorithms. Owing to these issues, at this point in time, we're unable to perform empirical comparisons of Algorithm 1 to other candidate algorithms with theoretical performance guarantees in a countable state setting.

As discussed in the previous paragraph, learning algorithms with theoretical performance guarantees are established in the finite state setting. One such algorithm is the certainty equivalence control with forcing, which is proposed and discussed in detail in [4]. To assess the finite-time performance of our algorithm, in Figure 4, we compare the performance of our proposed learning algorithm, denoted as TSDE, with the algorithm introduced in [4], referred to as AgrawalTeneketzis. Reference [4] proposes a certainty equivalence control law with forced exploration, which operates in episodes with increasing lengths and a priori fixed sequences of forcing times. Specifically, at the beginning of each episode, all possible stationary control laws are explored for one recurrence interval of state $(0, 0)$. Subsequently, based on this exploration, an empirical estimate of the average collected reward

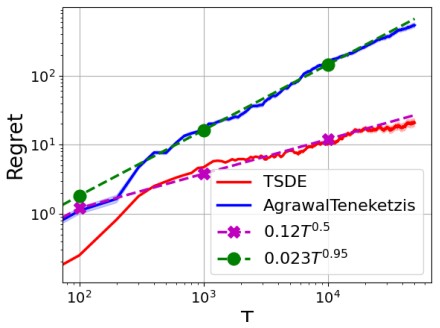
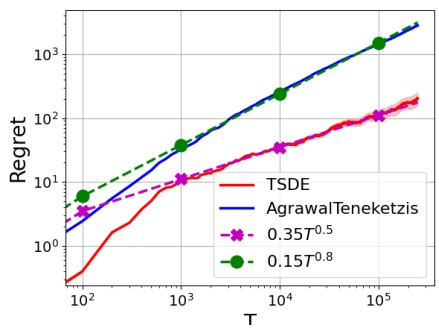

(a) Model 1: Queueing system of Figure 2a.    (b) Model 2: Queueing system of Figure 2b.

Figure 4: Comparison of the regret performance of Algorithm 1 (referred to as TSDE) with the algorithm proposed by [4] (denoted as AgrawalTeneketzis) for the queueing models of Figure 2.

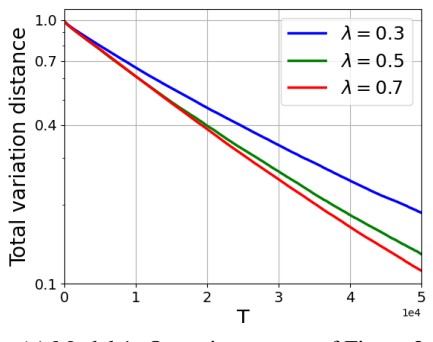
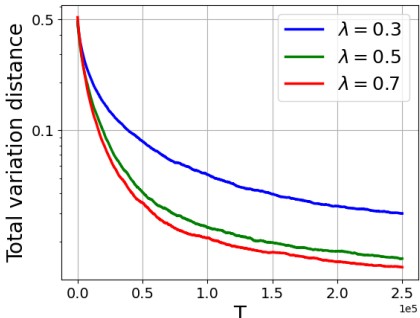

(a) Model 1: Queueing system of Figure 2a.    (b) Model 2: Queueing system of Figure 2b.

Figure 5: Total variation distance between the posterior and real distribution for $\lambda = 0.3, 0.5, 0.7$. The y axis is plotted on a logarithmic scale to display the differences clearly.

is formed, and the control law resulting in the maximum average reward is implemented for the remainder of the episode. The length of the episodes are determined according to sequence $\{a_i\}_{i=0}^{\infty}$ defined as following:

$$a_0 = 0,$$

$$a_i = \sum_{k=1}^{i} b_k + ip, \qquad \text{for } i \geq 1,$$

where $p$ is the number of possible stationary control laws and $b_i = \left\lfloor \exp\left(i^{\frac{1}{1+\delta}}\right) \right\rfloor$ for any $\delta > 0$. Specifically, episode $i$ terminates after completing additional $a_i - a_{i-1}$ recurrence intervals to state $(0,0)$. Both algorithms are implemented in the two queueing systems of Figure 2, where the arrival rate is $\lambda = 0.5$ and service rates are distributed according to a Dirichlet prior over $[0.5, 1.9]^2$. In Figures 4a and 4b, we set $\delta = 3.5$ and $\delta = 3$, respectively. Moreover, in Figure 4b, the goal is to find the optimal weight $w$ in the set $\{1.5, 2, 2.5, 3, 3.5\}$. The results in Figure 4 show that both algorithms exhibit a sublinear regret performance. Specifically, Algorithm 1, TSDE, achieves an $\tilde{O}(\sqrt{T})$ as predicted in our theoretical results of Theorem 1 and Corollary 1. Furthermore, in both queueing models, our proposed algorithm consistently outperforms the algorithm presented in [4] in terms of regret order.

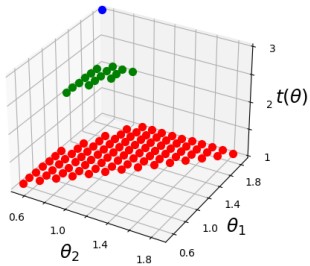
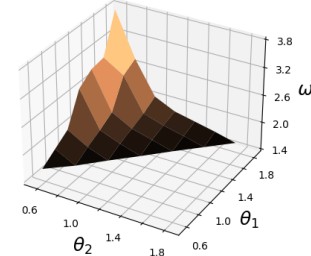

(a) Model 1: Queueing system of Figure 2a.      (b) Model 2: Queueing system of Figure 2b.

Figure 6: Optimal policy parameters for different service rate vectors in the two exemplary queuing systems in Model 1 and Model 2 with $\lambda = 0.5$.

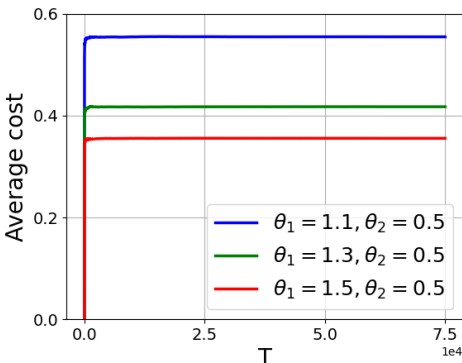

Figure 7: Estimated average cost of Model 2 for three different service rate vectors.

### G.2    Model 1: Two-server queueing system with a common buffer.

Figure 3b illustrates the behavior of the regret of Model 1 for three different arrival rate values and averaged over 2000 simulation runs. In these simulations, the parameter space is selected as

$$\Theta = \left\{ (\theta_1, \theta_2) \in [0.5, 0.6, \ldots, 1.9]^2 : \lambda < \theta_1 + \theta_2, \theta_2 < \theta_1 \right\},$$

which results in a prior size of 105. As depicted in Figure 3a, the regret has a sub-linear behavior and increases with the arrival rate. The total variation distance between the posterior and real distribution, a point-mass on the random $\theta^*$, are plotted in Figure 5a. As expected, the distance diminishes towards 0, indicating the learning of the true parameter. As mentioned in Appendix E.1, the optimal policy minimizing the average number of jobs in a system with parameter $\theta$, is a threshold policy $\pi_{t(\theta)}$ with optimal finite threshold $t(\theta) \in \mathbb{N}$, which can be numerically determined as the smallest $i \in \mathbb{N}$ for which $J^i(\theta) < J^{i+1}(\theta)$, calculated in [38]. We compute the optimal threshold $t(\theta)$ for every $\theta \in \Theta$ and present the results in Figure 6a. We can see that the threshold increases as the ratio of the service rates grows. Specifically, this is why in Appendix G, we imposed conditions on $\Theta$ to ensure that the ratio between the service rates is both upper and lower bounded.

### G.3    Model 2: Two heterogeneous parallel queues

Figure 3b illustrates the behavior of the regret of Model 2 for three different arrival rate values and averaged over 2000 simulation runs. We note that the regret is sub-linear and increases with higher arrival rates. In these simulations, the parameter space is selected as

$$\Theta = \left\{ (\theta_1, \theta_2) \in [0.5, 0.7, \ldots, 1.9]^2 : \lambda < \theta_1 + \theta_2, \theta_2 < \theta_1 \right\},$$

which results in a prior size of 28. As discussed earlier, our goal is to find the average cost minimizing policy within the class of policies $\Pi = \{\pi_\omega; \omega \in [(c_R R)^{-1}, c_R R]\}$, $c_R \geq 1$, where $\pi_\omega(\boldsymbol{x}) =$

$\arg\min\left(1+x_1, \omega\left(1+x_2\right)\right)$ with ties broken for 1. As discussed before, even with the transition kernel fully specified (by the values of arrival and service rates), the optimal policy in $\Pi$ is not known except when $\theta_1 = \theta_2$ where the optimal value is $\omega = 1$, and so, to learn it, we will use Proximal Policy Optimization with approximating martingale-process (AMP) method for countable state-space MDPs [18]. We run the algorithm for 200 policy iterations, using 20 actors for each iteration. We take the state $(0, 0)$ as a regeneration state and simulate 1500 independent regenerative cycles per actor in each algorithm iteration. To approximate the value function, we employ a fully connected feed-forward neural network with one hidden layer consisting of $10 \times 10$ units and ReLU activation functions. The AMP method is also employed for variance reduction in value function estimation. The optimal $\omega$ for every $\theta \in \Theta$ is shown in Figure 6b, indicating that $\omega$ increases as the ratio of the service rates grows. Therefore, it is necessary to ensure that the ratio between the service rates is bounded from above and below. Furthermore, to evaluate the regret numerically, the value of $J(\theta)$ is required for every $\theta \in \Theta$, which is not known. Thus, after finding the optimal $\omega$ using the PPO algorithm, we perform a separate simulation to approximate the optimal average cost. In Figure 7, we plot the estimated average cost for three different service rate vectors, demonstrating that the optimal average cost decreases as the service rates increase. In Figure 5b we also depict the total variation distance between the posterior and real distribution, which is a point-mass on the random $\boldsymbol{\theta}^*$, and observe that the distance is converging to zero.

