# OpenReview forum: "Bayesian Learning of Optimal Policies in Markov Decision Processes with Countably Infinite State-Space"
_NeurIPS.cc/2023/Conference — NeurIPS 2023 poster_

### Official Review · Reviewer_5dJh · 2023-06-26

**Soundness:** 3 good
**Presentation:** 3 good
**Contribution:** 3 good
**Rating:** 7
**Confidence:** 4

**Summary:**

This paper focuses on an online learning setting for Markov Decision Processes (MDPs) with a countably infinite number of states. It adopts a Bayesian learning perspective, assuming that the parameters of the MDP follow a prior distribution over a known parameter space. The paper proposes a Thompson-sampling-like approach to solve the MDP in an online fashion. This approach assumes access to an optimal policy oracle, where the parameters of the MDP are provided as inputs, and it also relies on specific assumptions about the features of the parameter space.

**Strengths:**

The model investigated in this paper exhibits a high degree of generality, and the results presented contribute significantly to the field of theoretical reinforcement learning by offering near-optimal algorithms for MDPs without a bounded state space. The inherent complexity of the problem necessitates intricate proofs, and although I haven't examined the complete proof in detail, I have confidence in the correctness of the underlying intuitions. The proof combines Lyapunov analysis with the proof presented in [38] for Bayesian learning in an MDP with a bounded state space, thus offering a potentially valuable contribution for future research. Additionally, the simulations conducted in the paper, which demonstrate the scaling of the algorithm's regret, are appreciated for providing empirical evidence supporting the algorithm's performance.

**Weaknesses:**

The results of the paper rely on a set of assumptions that may be difficult to verify. Of particular concern is Assumption 3, which assumes stability of the optimal algorithm under one set of parameters for the MDP, even when considering another set of parameters. Establishing this property for more general systems can be challenging and requires careful calibration of the parameter space and policy space.

The algorithm presented in the paper is heavily dependent on access to an oracle capable of solving the optimal policy, which itself is a complex problem for general queueing systems. This reliance on an oracle can limit the practical applicability of the algorithm.

The algorithm necessitates returning to state 0 (line 14) at the end of each episode. This requirement could result in an exponential dependence on the maximum queue length, potentially rendering the algorithm less relevant for practical implementation. As a result, the claim of practicality made in line 345 may not be adequately supported.

In relation to the previous point, the paper obscures many constants within the theoretical results. These constants, associated with the system's dimension and ergodicity, could play a crucial role in determining practical performance and should be given more attention and consideration.

**Questions:**

Is there a heuristic for designing the parameter space and policy space to ensure that Assumption 3 holds for a queueing system? What would happen to the algorithm if this assumption fails? It would be beneficial to test the algorithm in a more general system, such as the ones described in [41, 52].

How crucial is the requirement for an optimal policy oracle? Would the analysis still hold true if the policy space is restricted to, for example, the MaxWeight policy? It would be highly valuable if the results were presented in a form such as "the total queue lengths of this algorithm minus that of the MaxWeight policy is less than or equal to \sqrt{T}". This would also eliminate the need for the optimal oracle.

Could you discuss the dependence of the regret on the system size? Why does the dependence on T for the regret hold more significance than these constants?

Please maintain consistency by using either "queueing" or "queuing" throughout the paper, but not both.

In line 122, after "ergodicity)", there should be a period.

---

> ### Author Rebuttal · Authors · 2023-08-10
>
> We thank the reviewer for the suggestions on the presentation of the paper. We will make changes in the final version based on the suggestions.
>
> >Weakness 1. Necessity of assumptions.
>
> A. Given a specific parameter class $\Theta$, the assumptions can be verified for all MDPs corresponding to $\theta \in \Theta$, as  demonstrated in Appendix E for the two queuing models of Figure 2. While there isn't a programmatic/algorithmic way to find Lyapunov functions, queueing systems have accumulated a repository of Lyapunov functions (for classes of models and policies) over the years, making queueing systems a reasonable application domain to consider.  Assumption 3 is reasonable for queueing models since policies such as weighted Max-Weight are stabilizing for a large class of models. However, to the best of our knowledge, proving geometric ergodicity in such settings is an open problem; we prove it for the example of Figure 2b as we're using weighted Max-Weight.
>
> >Weakness 2. Requirement of an optimal policy oracle.
>
> A. For the requirement of an optimal policy oracle, please see Remark 2 in the global response.
>
> >Weakness 3. Bounds on hitting time of state $0^d$.
>
> A. Assumption 4 bounds the first $r+1$ moments of hitting time of state $0^d$ from any initial state $\boldsymbol{x}$ using the polynomial Lyapunov function (from Assumption 4); see Lemma 10. In contrast to the Lyapunov function used to prove geometric ergodicity (from Assumption 3) which is usually an exponential function of some norm of the state, the Lyapunov function used to show polynomial ergodicity (from Assumption 4) is often a polynomial function of the state, which then leads to the hitting time to $0^d$ being polynomially dependent on the initial state $\mathbf{x}$. Our queueing examples in Appendix E employ exponential Lyapunov functions to prove geometric ergodicity, and quadratic Lyapunov functions to establish polynomial ergodicity, with the latter leading to polynomial bounds for the hitting time of state $0^d$.
>
> >Weakness 4.  Dependence of regret bound on problem parameters.
>
> A. Please see Remark 3 in the global response.
>
> >Question 1. Requirement of Assumption 3.
>
> A. Assumption 3 needs to be verified on a case-by-case basis for the optimal policy of each model that can occur; see Figure 2a example. This is challenging since optimal policies have been characterized only for a small number of systems. However, within a given policy class, more general statements can be made-e.g., checking Assumption 3 for Max-Weight policies for a large class of queueing systems is feasible. However, even for MaxWeight policies establishing geometric ergodicity would be a new contribution to the literature, as the bulk of existing results establish ergodicity using a quadratic Lyapunov function. Note that the queueing model of Figure 2b (exemplifying Cor 1) uses MaxWeight policies a'la [52], and is illustrative of such settings. The geometric ergodicity characterization for this queueing model is new in the literature, so for the models in [52] too a geometric ergodicity result is needed (we expect this to hold based on their model assumptions).
>
> Whereas at present we don't know how to weaken Assumption 3 to an assumption similar to Assumption 4, we believe that stability assumptions across models are likely necessary for the countable space setting, and for the following reasons. First, in contrast to finite-state MDPs, where stability and existence of a stationary distribution are assured in a simple manner-via irreducibility or the existence of a single recurrent class, and aperiodicity-, the countable state-space setting needs additional conditions to ensure that the Markov process resulting from using any stationary policy $\pi \in \Pi$ is positive recurrent or ergodic. Furthermore, recovering after either using an unstable policy or starting in a transient state can be problematic, and with a countable set of transient states $S^{\mathrm{tr}}$, the expected time to exit $S^{\mathrm{tr}}$ can be infinite.
>
> >Question 2. Requirement of an optimal policy oracle.
>
> A. Please see Remark 2 in the global response. We also want to note that the queueing model of Figure 2b uses a weighted Max-Weight policy-we are selecting the best set of weights for each model within a class of weighted Max-Weight policies using the optimal oracle (the PPO algorithm used in simulations), and then regret is compared relative to the performance of the best weighted Max-Weight algorithm.
>
> >Question 3. Dependence of the regret on the problem parameters.
>
> A. Please see Remark 3 in the global response. We conclude by emphasizing that our sub-linear regret guarantee provides a rate of convergence characterization for asymptotic optimality results developed in past work, such as [18,27].

---

> > ### Comment · Reviewer_5dJh · 2023-08-10
> >
> > The authors' response addresses my questions, and I am particularly satisfied with the extension (Remark 2) that an optimal policy oracle may not be needed as long as the stability assumptions can be verified. I would like to raise my rating to 7.

---

> > > ### Author Response · Authors · 2023-08-20
> > >
> > > We thank the reviewer for the positive feedback.

---

### Official Review · Reviewer_5uHS · 2023-07-01

**Soundness:** 2 fair
**Presentation:** 2 fair
**Contribution:** 3 good
**Rating:** 5
**Confidence:** 2

**Summary:**

The authors study Bayesian learning of the problem of optimal control
of a family of discrete-time countable state-space MDPs governed by an unknown parameter $\theta$ from a general parameter space $\Theta$ with each MDP evolving on a common countably-infinite state space $X$ and finite action space $A$.
As the setting is Bayesian, they assume that the model is governed by an unknown
parameter $\theta_\star \in \Theta $ generated from a fixed and known prior distribution.
The learning goal: Bayesian regret minimization where the value function is the infinite-horizon average cost, and the regret is measured with respect to best policy in $\Pi$.
They prove a $\sqrt{TA}$ regret bound, up to poly-logarithmic factors, but the dependency in the complexity of the function class is unclear to me.

Disclaimer: I am not much familiar with this area of RL literature and hence might not understand the results correctly. Also, I could not fully verify the correctness of the presented results.
I gave that assessment  as the paper was very hard to follow for an unfamiliar reader.


**Strengths:**


1.	I think the results are a nice contribution to the Bayesian  RL community.
2.	The two examples presented contribute to the richness of the paper.
3.	The presented bounds seem reasonable.
4.	The adaption of Thompson sampling to countable state spaces might be useful in other RL settings.


**Weaknesses:**


1.	Abstract is quite long and parts of it seems like copy-paste of the introduction.
2.	Writing requires improvement: (1) lightening the contribution compared to existing literature, (2) notation is hard to follow and makes proof reading hard, (3) complexity of $\Pi$ or the dimension $d$ should appear in the presented bounds, even if the dependency on them is logarithmic, those parameters are not negligible.
3.	Need to mention the dependency in complexity of the function class is in the regret bound. Is it logarithmic for finite $\Pi$?  Is the regret dependent on the covering number? I would be happy if the authors could clarify that point.


**Questions:**

See weakness.

**Limitations:**

N/A.

---

> ### Author Rebuttal · Authors · 2023-08-10
>
>
> We thank the reviewer for the suggestions on the presentation of the paper and other remarks, and we will make changes in the final version. We note that the references marked with a letter are listed at the end of the response and are not included in the submission.
>
> For discussion on the dependence of our regret bound on problem parameters, please see Remark 3 in the global response.
>
> Regarding dependence of the regret on the complexity of $\Pi$, using our method and assumptions, it is not easy to characterize the regret in terms of complexity and associate Lyapunov-type conditions with covering numbers. The complexity implicitly determines the uniformity conditions of Assumptions 3, 4, and 5, and it is challenging to characterize the regret separately in terms of complexity of $\Pi$. In essence, the bounds are akin to instance bounds. This is easily understood when $\Pi$ is finite, as the quantities $J^*$ and $r_*^p$ above are the result of the maximization of similar parameters over a finite number of policies and models.
>
> To address the dependency on the covering number, we will need to explore model mismatch/perturbations more carefully, but such results are not available currently for countable state-space MDPs (to the best of our knowledge). Our assumptions result in $\Pi$ being compact, and so it is totally bounded, and given any error terms, we can get a finite cover of the parameter space. By using the centers of these covers---sampling from posterior and projecting to the closer center that covers the sample---a theoretical analysis may be feasible for the general problem if the transition kernels depend smoothly on the parameters. To make the analysis work, we will need results that can compare the performance of optimal policies of two close models (close as per metrics suggested in [A] based on the closeness of the parameters). While such perturbation-related results are available in general state-space problems for finite-horizon and discounted cost problems, they only exist in the finite state-space setting for average cost optimal problems:  [A] discusses the results for finite-horizon and discount cost problems; and
> the results of [B] can be extended to the average cost problem in the finite state-space setting by the vanishing discount method (taking a limit as the discount factor converges to $1$).
>
> ___
> [A] Müller, A. ``How does the value function of a Markov decision process depend on the transition probabilities?" Mathematics of Operations Research 22.4 (1997): 872-885.
>
> [B] Subramanian, J., Sinha, A., & Mahajan, A. ``Robustness and sample complexity of model-based MARL for general-sum Markov games." Dynamic Games and Applications 13.1 (2023): 56-88.

---

> > ### Comment · Reviewer_5uHS · 2023-08-10
> >
> > I thank the authors for their response and have no further questions.
> > I would positively consider to raise my rating.

---

> > > ### Author Response · Authors · 2023-08-20
> > >
> > > We thank the reviewer for the positive feedback.

---

### Official Review · Reviewer_Raj1 · 2023-07-06

**Soundness:** 3 good
**Presentation:** 3 good
**Contribution:** 3 good
**Rating:** 6
**Confidence:** 4

**Summary:**

This paper present an adaptation of TSDE to parametric MDPs with unbounded state space.

The regret is sqrt{T} which is good but that is  under strong ergodic assumptions and lower order terms can harm the behavior of the algorithm for small values of T.

**Strengths:**

- The paper is sound technically. (I quickly checked the proofs)

- The problem of learning unbounded MDP is a natural and important question to address.


**Weaknesses:**

1. The strong conditions (especially  Ass 4) are not  necessary and sufficient conditions for existence of an optimal policy, nor for the existence of a solution of the Bellman equation.

2. The queueing examples are not really convincing: The optimal service rates can be computed efficiently by just estimating the arrival rate and solving the optimal control problem for the expected sojourn time. Also, numerically,  the growing rate of the regret gets worse as the arrival rate increases.



**Questions:**

1. The assumption on the unicity of the optimal policy is not related with all the other assumptions and is not discussed. This is actually a strong assumption restricting the class of MDPs that can be learned. This deserves some discussion why it is needed here.


2. The authors should discuss the practical aspect of their approach. In particular, can one check whether all the assumptions are satisfied while the MDP is unknown?

3. The Q-learning approach seems more natural than a model based approach under unbounded space spaces, where only the visited states are used for learning. Several papers have already taken this option successfully and are not discussed here (see for example "Stable reinforcement learning with unbounded state space" by  D Shah, Q Xie, Z Xu )


**Limitations:**

I cannot see any limitations

---

> ### Author Rebuttal · Authors · 2023-08-10
>
> We thank the reviewer for their constructive review, and reference suggestions (which we will cite). The references depicted with a number indicate a reference in the submission and the references marked with a letter are listed at the end of the response.
>
> >Weakness 1.  Necessity of assumptions.
>
> A. Assumptions 1 & 2, combined with the positive recurrence from Assumption 3, ensure the existence of an optimal policy and a solution to the average cost optimality equation (ACOE) & Poisson equation; see Section 2. Assumptions 3 & 4 are required for our analysis, as we need bounds on the moments of the maximum state norm reached by time $T$ as well as the hitting time to state $0^d$, which we found using the Lyapunov functions in Assumptions 3 & 4. For the average cost problem in countable spaces, establishing necessary and sufficient results for the existence of stationary optimal policies, or for solutions of the average cost optimality equation are open problems. We used conditions from [9] (that are satisfied using our assumptions). The weakest conditions are in [A], and using them is for future work, but we note the importance of the existence of stabilizing policies with finite average cost even in this paper; see [B] for a comparison of different conditions in the literature. Bounded costs can be problematic in the countable setting as they do not cover practical examples from queueing systems. Thus, we impose Assumption 1: our cost function is unbounded.
>
> >Weakness 2. Queueing examples.
>
> A. Whereas it may be possible to establish asymptotic optimality for a scheme employing rate estimates (using a modified MLE a'la adaptive control), to the best of our knowledge, there are no finite-time regret guarantees provided for such schemes. Existing related results-[18] (finite number of policies) and [27] (countable or uncountable number of policies)-prove asymptotic optimality but assume further structure on the transition kernels, which our examples do not satisfy.
>
> Let $\rho$ be the normalized load. Then, the gap from the capacity boundary is a linear function of 1-$\rho$. Thereafter, the regret growing as the normalized load goes to 1, i.e., gap going to 0, is expected, since the system gets closer to the stability boundary as the arrival rate increases. Based on Remark 3 in the global response, our regret bound depends on $J^*$ and, thus, on the gap from the capacity boundary and will increase as $\rho$ goes to 1, which reinforces our earlier comment.
>
> >Question 1. Unicity of the optimal policy.
>
> A. Uniqueness of the optimal policy is not essential for the validity of our results, provided that all optimal policies satisfy our assumptions. When this condition is not met, we need to select an optimal policy that is geometrically ergodic for all parameters. This could entail searching over all optimal policies when non-uniqueness holds. This issue can be avoided by using a smaller subset of policies, such as Max-Weight policies, for which the ergodicity can be established for all possible models. We will clarify this in the final version.
>
> >Question 2. Practicality of our approach.
>
> A. For a given parameter set $\Theta$, the assumptions can be verified for all MDPs corresponding to any $\theta\in\Theta$, as in Appendix E for queuing models of Figure 2. Whereas there isn't an algorithmic way to find Lyapunov functions, queueing systems have accumulated a repository of Lyapunov functions over the years, making queueing systems a reasonable application domain to consider.
>
> >Question 3. Q-learning based on Shah et al. [E]
>
> A. The differences with our work are as follows: 1. [E] ignores optimality and focuses on finding a stable policy, which contrasts with our work that evaluates performance relative to the optimal policy. 2.[E] considers a discounted reward problem, essentially a finite-time horizon problem (given the geometrically distributed lifetime). Average cost problems (as studied by us) are infinite-time horizon problems, so connections to discounted problems can only be made in the limit of the discount parameter going to 1. 3. Moreover, [E] considers a bounded reward function which simplifies their analysis, but is not a practical assumption for many queueing examples. Further, for bounded reward settings with discounting, the assumption of a stable optimal policy with a Lyapunov function (as in [E]) is extremely restrictive: e.g., if the rewards increase to a bounded value as the state goes to infinity, then the stationary discount-cost optimal policy will likely be unstable as the goal will be to increase the state as much as possible. Additionally, bounded costs for average cost problems need strong state-independent recurrence conditions for the existence of (stationary) optimal solutions, which many queueing examples don't satisfy: see [C] for necessary conditions, plus [D] shows that a stationary average cost optimal policy may not exist. Finally, we are not aware of any other RL algorithms with provable low regret for the average cost problem in the countably infinite setting with an unknown model.
> ___
> [A] Sennott, L l. Average cost optimal stationary policies in infinite state Markov decision processes with unbounded costs. Operations Research 37.4 (1989): 626-633.
>
> [B] Cavazos-Cadena, R., Sennott, L. I. Comparing recent assumptions for the existence of average optimal stationary policies. Operations Research Letters 11.1 (1992): 33-37.
>
> [C] Cavazos-Cadena, R. Necessary conditions for the optimality equation in average-reward Markov decision processes. Applied Mathematics and Optimization 19.1 (1989): 97-112.
>
> [D] Fisher, L., Ross, S. M. An example in denumerable decision processes. The Annals of Mathematical Statistics 39.2 (1968): 674-675.
>
> [E] Shah, D., Xie, Q., Xu, Z. Stable reinforcement learning with unbounded state space. arXiv preprint arXiv:2006.04353 (2020).

---

### Official Review · Reviewer_MJCc · 2023-07-24

**Soundness:** 3 good
**Presentation:** 2 fair
**Contribution:** 3 good
**Rating:** 7
**Confidence:** 3

**Summary:**

The authors consider the average reward Markov decision process framework with countable state spaces. The considered objective is to perform closed-loop optimal control for a family of MDPs parameterized in a compact space; this is a particularly interesting setting as the cost function is not assumed to be bounded. The authors propose a Thompson Sampling based algorithm and analyze its regret under suitable assumptions. They are able to show a finite time $\sqrt{|\mathcal{A}| T}$ bayesian regret bound. The authors also show the practical significance of their algorithm by an empirical application to queuing models.


**Strengths:**

- This paper is very relevant to the community as it opens the way towards studying RL in continuous spaces without the assumptions of bounded reward functions. Therefore tightening the connection of reinforcement learning theory and optimal control.

- Although certain assumptions are somehow stringent, it is nice that the paper is able to convert an infinite horizon setting to a somewhat episodic setting. The latter usually allow for simpler analyses.

- The application to a queuing model is also appreciated, a nice change from the standard RL benchmarks.

- I would like to clarify that I did not go through the theoretical proofs and can therefore not comment on their correctness apart from the general intuition that such results should be possible given appropriate assumptions on stability and dynamics.


**Weaknesses:**

- The assumptions seem somewhat stringent. For example, the finite support of transitions can be challenged even in the specific example of queuing models when large amounts of arrivals are possible. More importantly, assumption 3 seems quite limiting, can the authors elaborate on what this implies for stability? I am not very knowledgeable in optimal control but it seems that you are assuming stability of all policies?

- The paper is sometimes very dense and not straightforward to follow. For example, lines 173 to 192 requires prior knowledge of several papers and definitions, e.g. Poisson equation’s link to the problem at hand/ forcing function and other similar passages in the text. I would advise the authors to add the relevant definitions and Lemmas at least in the appendix to make the paper self contained.

- The paper fails to cite many RL works in the continuous spaces setting. Namely, the entire line of function approximation in RL, for example: 1) “Frequentist Regret Bounds for Randomized Least-Squares Value Iteration” by Zanette et al. which provides a algorithm based on Thompson Sampling as well, and many other works (see references therein) in the model-free paradigm 2) “Bilinear Exponential Family of MDPs: Frequentist Regret Bound with Tractable Exploration and Planning” which also provides similar algorithms for MDPs that seem to include the queuing models presented here, see references therein for model-based approaches to RL in continuous spaces.


**Questions:**

- In line 203, where is $\pi_{\theta_k}^*$ defined? How is this implemented in the examples?

- From your understanding of your assumptions, is the stability ensured by default? Or does Thompson sampling just ensure it by some magic in the proof? If the latter, and since you have a bound on the maximum $\ell_\infty$ norm of the states, is it not more straightforward to use existing RL algorithms? I guess what I’m asking is how necessary the Lyapunov-type analysis is in this case?

- Why aren’t there comparisons with other RL algorithms in the experiments? especially since you implement PPO in the second model for your experiment, isn’t it possible to compare directly against it and other deep RL methods?



**Limitations:**

The proposed work and algorithms is theoretical and does direct societal impacts. The possible theoretical limitations are detailed to some extent in the paper and to be addressed in future work.

---

> ### Author Rebuttal · Authors · 2023-08-09
>
>
> We thank the reviewer for their constructive review and references (which we will cite). We will add related definitions in the appendix of the final version. The references depicted with a number indicate a reference in the submission, whereas the references marked with a letter are listed at the end of the response.
>
> >Weakness 1. Necessity of assumptions.
>
> A: The state-dependent support of the transition kernel is finite for every state-action pair, but we do not assume that the supremum of the number of possible transitions is finite over the state-space. Many queueing models fit this setting-see our examples at Section 5 and reference [52] (arrivals have an explicit bound). Generalizing to allow for an arbitrarily large number of arrivals at any instance will necessitate adjustments to our proof, but with reasonable assumptions (like finite moment generating function), we expect the results to continue to hold. As noted in Appendix A.1, Assumption 3 imposes stability criterion---geometric ergodicity---that applies uniformly across policies, models, and states: for any $\theta_1,\theta_2 \in \Theta$, sequence $\{(P_{\theta_1}^{\pi^*_{\theta_2}})^n\}$ converges geometrically fast to the stationary distribution $\mu_{\theta_1,\theta_2}$, where $P_{\theta_1}^{\pi^*_{\theta_2}}$ is the transition kernel of the Markov process obtained from the MDP $\left(\mathcal{X},\mathcal{A},c,P_{\theta_1}\right)$ by following policy $\pi^*_{\theta_2}$. This is a stronger compared to positive recurrence-beyond the existence of a stationary distribution, it enforces a geometric convergence rate to the stationary distribution. In our context, we assume this convergence for all optimal policies $\pi^*_{\theta}$ corresponding to some $\theta \in \Theta$. We need this assumption to upper bound the first $r+1$ moments of the maximum state norms and hitting times of state $0^d$. At present, we don't know if we can weaken the requirement from geometric ergodicity and to polynomial ergodicity (see Assumption 4). Please see Remark 1 in the global response for more discussion on the importance of stability.
>
> > Weakness 3. Citation of RL works Zanette et al. [A] & Ouhamma et al. B].
>
> A: We differentiate our work from [A,B] as follows. Both works [A,B] consider a finite-horizon problem. In contrast, our work considers an average cost problem which is an infinite-horizon setting, and provides finite-time performance guarantees; asymptotic optimality of our algorithm is then immediate. In addition, [A] studies an MDP with a bounded reward function. Our focus, however, is learning in MDPs with unbounded rewards with the goal of covering practical examples from queueing systems. The reviewer is correct that parameterization of transition kernels used in [B] and the prior work [C], can be used within the framework of our problem. However, similar to our work, additional assumptions, importantly stability conditions proposed in our problem, are necessary to guarantee asymptotic learning and sub-linear regret. As there aren't general necessary and sufficient conditions on the parameters to ensure stability, posterior updates can be complicated with this parameterization. Another issue with exponential families of transition kernels is that they do not allow for $0$ entries (except through parameters increasing without bound), and so will not be directly applicable to queueing models (like our examples).
>
> >Question 1. Definition of $\pi^*_{\theta_k}$.
>
> A: In Assumption 3, we have defined $\pi^*_{\theta_k}$ as the unique optimal policy that minimizes the infinite-horizon average cost for MDP $\left(\mathcal{X},\mathcal{A},c,P_{\theta_k}\right)$ within the policy class $\Pi$; Thm 1 considers all policies, and Cor 1 uses a subset of all policies. Here, $\theta_k$ is the sample generated from the posterior distribution at the beginning of episode $k$. In the queueing model of Figure 2a (illustrating Thm 1), $\Pi$ is the set of all policies, and the optimal policy corresponding to any parameter $\theta$ in $\Pi$ is explicitly characterized in [29], which is a threshold policy with an explicitly determined (finite) threshold. In the queueing model of Figure 2b (illustrating Cor 1), we find the average cost-minimizing policy within a subset of weighted Max-Weight policies, which route arrivals based on weighted queue lengths. The optimal policy even in this set is not known except when $\theta_1=\theta_2$ where the optimal value is ${\omega}=1$, and so, to learn it, we use Proximal Policy Optimization (PPO) for countable state-space MDPs [13]; see Figure 5b in Appendix. Hence, in both cases, the optimal oracle can be implemented.
>
> >Question 2. Necessity of stability.
>
> A: Please see Remark 1 in the global response.
>
> >Question 3. Comparisons with other RL algorithms.
>
> A: The PPO algorithm of [13] \textbf{requires} model knowledge and finds the optimal average-cost policy using PPO for an MDP with known transition kernels. In our second queueing model in Figure 2b, we have used the PPO algorithm to find the best in-class policy (illustrating Cor 1), utilized in Line 7 of Algorithm 1 after realizing parameter $\theta^*_k$ from the posterior distribution. We are not aware of any RL algorithms with provable low regret in the setting of countably infinite spaces with an unknown model.
> ___
> [A] Zanette, A., et al. Frequentist regret bounds for randomized least-squares value iteration. International Conference on Artificial Intelligence and Statistics. PMLR, 2020.
>
> [B] Ouhamma, R., Debabrota B., and Odalric M. Bilinear Exponential Family of MDPs: Frequentist Regret Bound with Tractable Exploration & Planning. Proceedings of the AAAI Conference on Artificial Intelligence. Vol. 37. No. 8. 2023.
>
> [C] Chowdhury, S. R., Gopalan, A., & Maillard, O. A. Reinforcement learning in parametric MDPs with exponential families. In International Conference on Artificial Intelligence and Statistics, pages 1855–1863. PMLR, 2021.

---

> > ### Comment · Reviewer_MJCc · 2023-08-18
> >
> > Thanks to the authors for their clear answer. I understand the necessity of the stability assumptions, but I still do not see how, given these assumptions, a standard RL algorithm with a high probability analysis wouldn't work just as fine. You seem to have an explicit bound on the maximum norm after $T$ time steps and this can be directly injected into the traditional analyses, am I wrong? Also, I still think authors should make a significant addition to the presentation to address Weakness 2 that I raised. Finally, for the empirical evaluation, what I meant is not to compare necessarily to some theoretically studied algorithm, but rather just the standard deep-RL algorithms, otherwise there is no baseline to also judge the empirical relevance.
> >
> > I believe that with minor improvements, the paper could be a good addition to the conference. Therefore, I would like to keep my score.

---

> > > ### Author Response · Authors · 2023-08-20
> > >
> > > We thank the reviewer for the positive feedback.
> > >
> > > 1. The skip-free to the right property in Assumption 2 yields a polynomially-sized subset of the underlying state-space depending on the time-horizon $T$, specifically $S(T)=(hT)^d$. This polynomially-sized subset can be viewed as the effective finite-size of the system in the worst-case, and then, directly applying finite-state problem bounds (e.g., by using [38]) would result in a regret of order $\tilde O(S(T)
> > > \sqrt{ |\mathcal{A}| T})$, which is essentially  $\tilde O(T^{d+0.5})$; since $d\geq 1$, such a coarse bound is not helpful even for asserting asymptotic optimality. Thus, to achieve a regret of $\tilde O(\sqrt{T})$, it is essential to carefully understand and characterize the distribution of $M^T_{\boldsymbol \theta^*}$ and then its moments; see Remark 1 in Appendix B. Furthermore, for the truncation plus standard RL algorithm, we need to add the error/regret due to the truncation of the state space to the regret term  $\tilde O(S(T)
> > > \sqrt{ |\mathcal{A}| T})$. As the stationary distribution of the optimal policy (in most examples) does ascribe non-zero probability to states outside the truncated finite space $S(T)$, the error here will depend on ergodicity properties—most likely decreasing to zero fast in $T$ with geometric ergodicity and much slower with polynomial ergodicity. Further, such a scheme works with a fixed $T$, so either the horizon needs to be fixed ahead of time or the doubling-trick needs to be used. In essence, more care is likely needed to use a truncation scheme.
> > >
> > > 2. We will address weakness 2 in our final version and add related definitions to the appendix.
> > >
> > >
> > > 3. The comparison with a standard deep RL scheme is a future direction, but it would be challenging to determine uniformly suitable hyperparameters throughout the parameter space.

---

### Author Rebuttal · Authors · 2023-08-10

Below we address questions and remarks to common questions.

>Remark 1. The necessity of stability assumptions.

Stability needs to be imposed separately, so we use Assumption 3. This is due to the following reasons. In contrast to finite-state MDPs, where stability and existence of a stationary distribution are easily assured-via irreducibility or the existence of a single recurrent class, and aperiodicity-, the countable state-space setting needs additional conditions to ensure that the Markov process resulting from using any stationary policy $\pi \in \Pi$ is positive recurrent or ergodic. Furthermore, recovering after either using an unstable policy or starting in a transient state can be problematic, and with a countable set of transient states $S^{\mathrm{tr}}$, the expected time to exit $S^{\mathrm{tr}}$ can be infinite. See [A,B] for more discussion on importance of stability.

To analyze the regret, we use the independence structure resulting from recurrent visits to state $0^d$ (with inter-visit durations being well-behaved), which may not occur without stability assumptions. Furthermore, to characterize the regret, we also require bounds on the moments of the (random) maximum state norm $l_\infty$ reached by time $T$ and on the hitting time to state $0^d$, for which we used the Lyapunov functions of Assumptions 3 & 4. Reference [13] also uses Lyapunov function-based arguments for finding the average cost optimal policy in countable-state MDPs with \textbf{known} transition kernels. The authors impose geometric ergodicity and utilize Lyapunov function-based arguments to analyze their proposed PPO policy's performance.

To further clarify the necessity of stability assumptions,  consider the queueing model in Figure 2a (illustrating Thm 1), which has a set of countable transient states $S^{\mathrm{tr}}$-all states where the second server is occupied above the threshold. Our algorithm avoids this set (or other transient states) by always remaining within the (policy-dependent) reachable set of states from $0^d$ (which are positive recurrent by the stability assumptions).

Finally, the stability assumption gives probabilistic control on the random $l_\infty$ and number of episodes, both of which are crucial to the result. Just using the skip-free to right property yields a state-space (plus memory use, and regret) bound, i.e., $l_\infty$, that is exponential in the time-horizon $T$, so directly using RL algorithms and state-of-the-art results for such algorithms will yield too high a regret bound (one that scales exponentially in $T$).

>Remark 2. Requirement of an optimal policy oracle.

To implement our algorithm, when we determine regret with-respect-to an optimal policy, we necessarily need to find the optimal policy for each model sampled by the algorithm-optimal policy for Thm 1 and optimal within policy class for Cor 1; this has also been used in past work [17,18,27]. In the finite state-space setting, [38] provide a schedule of $\epsilon$ values to select $\epsilon$-optimal policies such that $\tilde{O}(\sqrt{T})$ regret results. The issue with extending the analysis of [38] to the countable state-space setting is that we need to formulate (and verify) ergodicity assumptions for a potentially large set of close-to-optimal algorithms whose structure is undetermined. Another issue is that, to the best of our knowledge, there isn't a general structural characterization of all $\epsilon$-optimal stationary policies for countable state-space MDPs, or even a characterization of the policy within this set that is selected by any computational procedure in the literature; current results only discuss existence and characterization of the stationary optimal policy. In the absence of such results, stability assumptions with the same uniformity across models as in our submission, will be needed. At present we don't know how to verify such an assumption for any example, but nevertheless the conditions required are likely to be too strong to be useful.

If we could verify the stability requirements of Assumptions 3 & 4 for a subset of policies, the optimal oracle is not needed, and instead by choosing approximately optimal policies within this subset, we can follows the same proof steps as [38] to guarantee regret performance similar to Corollary 1 (without knowledge of model parameters). For instance, in the queueing model of Figure 2b, if instead of the optimal weight, we have access to an (performance-wise) $\epsilon$-optimal weight policy, we can easily see that the approximate policy satisfies the stability assumptions and the sub-linear regret bounds carry through following arguments similar to those in [38], section 3.2.  We will add this extension to the final version.

>Remark 3. Dependence of expected regret on problem parameters.

We would like to clarify that our expected regret depends on the skip-free parameter defined in Assumption 2, $h$, and the dimension of the state space, $d$, the cost function parameters defined in Assumption 1, $K$ and $r$ , supremum on the optimal cost $J^*$, and  $r^p_*$ that is defined in Assumption 4 as $\tilde O(Kr\ d\ J^*\ h^{d+2r+r_*^p}\ \sqrt{|\mathcal A| T })$, where $\tilde O$ hides logarithmic factors in problem parameters-one of which is $\log^{d+r+r_*^p+2}(T)$. For simplicity, we have not included the Lyapunov functions related parameters in the regret, but we will add the order accounting for the parameters associated with the problem structure to our final version.
___

[A] Sennott, Linn I. Average cost optimal stationary policies in infinite state Markov decision processes with unbounded costs. Operations Research 37.4 (1989): 626-633.

[B] Cavazos-Cadena, Rolando, and Linn I. Sennott. Comparing recent assumptions for the existence of average optimal stationary policies. Operations Research Letters 11.1 (1992): 33-37.

---

### Decision · Program_Chairs · 2023-09-21

**Decision:**

Accept (poster)

**Comment:**

The paper considers the problem of online RL in MDPs with countably infinite state-space. The authors provided sufficient support the interest in expanding existing theory to this setting. While the core algorithmic idea and some of the technical tools clearly derive from previous literature, the final result requires non-trivial technical steps and it is interesting. Overall, there is consensus that the paper is worth accepting.

I would still suggest the authors to consider improving the original submission by integrating the discussion with the reviewers, in particular the following aspects:

While the queuing model is interesting and it is important on its own, it would be helpful to have other examples of real-world problems that can be modeled as MDPs with countably infinite state-space.
Clarify the intuition of why the assumptions are needed.
Provide more details about the experiments.